# Stable Deep Reinforcement Learning via Isotropic Gaussian Representations

Ali Saheb Pasand [* 1 2]   Johan Obando-Ceron [* 1 3]   Aaron Courville [1 3]
Pouya Bashivan [† 1 2]   Pablo Samuel Castro [† 1 3 4]

## Abstract

Deep reinforcement learning systems often suffer from unstable training dynamics due to non-stationarity, where learning objectives and data distributions evolve over time. We show that under non-stationary targets, isotropic Gaussian embeddings are provably advantageous. In particular, they induce stable tracking of time-varying targets for linear readouts, achieve maximal entropy under a fixed variance budget, and encourage a balanced use of all representational dimensions– all of which enable agents to be more adaptive and stable. Building on this insight, we propose the use of Sketched Isotropic Gaussian Regularization for shaping representations toward an isotropic Gaussian distribution during training. We demonstrate empirically, over a variety of domains, that this simple and computationally inexpensive method improves performance under non-stationarity while reducing representation collapse, neuron dormancy, and training instability. **Our code is available at [IsoGaussian-DRL](IsoGaussian-DRL)**

## 1. Introduction

Deep reinforcement learning (deep RL) has achieved striking successes across a wide range of domains, including robotics, control, game playing, and large-scale decision-making (Vinyals et al., 2019; Bellemare et al., 2020; Fawzi et al., 2022; Schwarzer et al., 2023). Despite this progress, optimization pathologies such as instability, plasticity loss (Nikishin et al., 2022), neuron dormancy (Sokar et al., 2023), and representation collapse (Moalla et al., 2024; Mayor et al., 2025) remain persistent and often limit both performance and scalability. A defining challenge of deep RL is its inherently non-stationary nature. Unlike supervised or stan-

dard self-supervised learning, the data distribution, learning targets, and optimization landscape evolve continually as the agent updates its policy and interacts with the environment (Sutton & Barto, 1988; van Hasselt et al., 2018). This non-stationarity has been repeatedly linked to degraded representation quality, unstable gradients, and brittle learning dynamics, ultimately reducing effective capacity over long training horizons (Kumar et al., 2021; Obando-Ceron et al., 2026; Liu et al., 2026; Castanyer et al., 2026). Most prior work addressing instability in deep RL focuses on algorithmic or architectural interventions, including modified update rules (Hessel et al., 2018), auxiliary objectives (Castro et al., 2021; Schwarzer et al., 2021), target networks, or carefully designed architectural heuristics (Ceron et al., 2024c;b; Lee et al., 2025; Castanyer et al., 2026). While often effective, these approaches typically address downstream symptoms of instability and may introduce additional complexity or tuning requirements. In contrast, the role of representation geometry, specifically, how the statistical structure of learned representations interacts with non-stationary targets, has received comparatively limited attention in deep RL.

Recent advances in self-supervised learning suggest that representation geometry plays a central role in stability and generalization. In particular, LeJEPA (Balestriero & LeCun, 2025) demonstrates that enforcing simple statistical structure, namely isotropy and approximate Gaussianity, yields representations that generalize robustly across diverse downstream tasks. This setting closely mirrors deep RL, where the effective task and target distribution evolve over time and are not known a priori. Motivated by this parallel, we revisit instability in deep RL through the lens of representation geometry and ask: *what properties should representations satisfy to remain stable under continuously evolving targets?*

We show that isotropic Gaussian representations are particularly well suited to this regime. For linear readouts tracking non-stationary targets, such representations minimize sensitivity to drift, maximize entropy under a fixed variance budget, and promote balanced utilization of representational dimensions. Together, these properties mitigate collapse, reduce neuron dormancy, and stabilize learning dynamics under distributional shift. These guarantees highlight isotropic Gaussian structure as a principled target for representations in non-stationary learning systems.

---

[*]Equal contribution [†]Equal advising [1]Mila – Quebec AI Institute [2]McGill [3]Université de Montréal [4]Google DeepMind. Correspondence to: Ali Saheb Pasand <ali.sahebpasand@mila.quebec>, Johan Obando-Ceron <jobando0730@gmail.com>.

*Proceedings of the 43rd International Conference on Machine Learning*, Seoul, South Korea. PMLR 306, 2026. Copyright 2026 by the author(s).

To empirically study this hypothesis, we evaluate methods that encourage isotropic Gaussian structure in learned representations within standard deep RL pipelines. Our experiments span discrete and continuous control benchmarks, including the Arcade Learning Environment (Bellemare et al., 2013) and Isaac Gym (Makoviychuk et al., 2021), and cover both value-based and policy-gradient methods such as Parallelized Q-Networks (PQN) (Gallici et al., 2025) and Proximal Policy Optimization (PPO) (Schulman et al., 2017). Across settings, we observe consistent improvements in training stability, representation quality, and performance under non-stationarity. These results suggest that representation geometry, rather than algorithmic complexity or curvature estimation alone, plays a central role in stabilizing learning under non-stationarity such as deep RL.

## 2. Preliminaries

**Deep Reinforcement Learning.** Deep RL addresses sequential decision-making problems in which an agent learns a policy mapping states to actions through interaction with an environment, typically using neural networks as function approximators (Mnih et al., 2015). This interaction is formalized as a Markov decision process (MDP) $(\mathcal{S}, \mathcal{A}, P, r, \gamma)$ (Puterman, 1994). A policy $\pi_\theta(a \mid s)$ is optimized via gradient-based methods, inducing a discounted state–action occupancy measure $d_{\pi_\theta}(s, a)$. As policy parameters are updated, both the policy and the induced data distribution evolve, resulting in a continuously shifting training distribution even under fixed environment dynamics. Most deep RL algorithms rely on bootstrapped learning targets. Value-based and actor–critic methods, including DQN (Mnih et al., 2015), PQN (Gallici et al., 2025), SAC (Haarnoja et al., 2018), and TD3 (Fujimoto et al., 2018), optimize objectives of the form $\mathcal{L}_{\text{TD}}(\phi) = \mathbb{E}_{(s,a,r,s') \sim d_{\pi_\theta}} [(Q_\phi(s, a) - y_{\theta,\phi}(s, a, r, s'))^2]$; where the target $y_{\theta,\phi}$ depends on learned quantities. As a result, both the data distribution and learning targets evolve throughout training, introducing inherent non-stationarity and contributing to optimization instability (Vincent et al., 2025; Hendawy et al., 2025). Empirically, this non-stationarity leads to representation drift, neuron dormancy, and loss of effective capacity, often manifested as rank collapse and degraded adaptability (Lyle et al., 2022; Nikishin et al., 2022; Moalla et al., 2024; Tang et al., 2025).

**Representation Learning in Deep RL.** Deep RL agents typically decompose their networks into a shared representation $h_\eta(s) \in \mathbb{R}^d$ and task-specific prediction heads, e.g., $Q_\phi(s, a) = g_\omega(h_\eta(s), a)$ (Fujimoto et al., 2023; Echchahed & Castro, 2025). Although objectives are defined over scalar signals such as returns or advantages, gradients propagate through the prediction head into the representation. Consequently, deep RL simultaneously performs control and representation learning under indirect, noisy, and non-stationary

supervision (Igl et al., 2021). Sampling $s \sim d_{\pi_\theta}$ induces a distribution over representations $z = h_\eta(s)$, whose geometry evolves throughout training.

## 3. Isotropic Gaussian Representations Promote Stability Under Non-Stationarity

We study the role of isotropic Gaussian representations by analyzing learning dynamics under drifting targets. We show that when embeddings are constrained to be isotropic and Gaussian, the resulting optimization dynamics has a stable fixed point, and the tracking error remains bounded and decreases over time. This provides a theoretical explanation for the empirical robustness of isotropic Gaussian representations under non-stationary training and motivates explicitly enforcing this type of representation in deep RL. Our analysis is based on the formal model in App. B, where representation learning is viewed as a tracking problem under drifting supervision. This analysis is inspired by Lemmas 1 and 2 in (Balestriero & LeCun, 2025), which show that, without knowledge of the downstream task, an isotropic representation is optimal for learning a wide range of downstream tasks. Building on this insight, we model non-stationarity as learning a sequence of tasks that evolve over time, and we seek conditions that ensure task changes induced by non-stationarity remain learnable.

### 3.1. Linear Tracking Under Non-Stationary Targets

We consider a linear critic on top of a learned embedding, $Q_\theta(s, a) = w^\top \phi(s, a)$, where $\phi(s, a) \in \mathbb{R}^d$ is the penultimate-layer embedding and $w \in \mathbb{R}^d$ is the last-layer weight vector. The TD target is time-varying, $y_t = r + \gamma Q_{\theta^-}(s', a')$, which induces non-stationarity in the learning problem. To characterize the resulting dynamics, define the second-order statistics $\Sigma_\phi(t) := \mathbb{E}[\phi\phi^\top]$, $b_t := \mathbb{E}[\phi y_t]$, and the expected critic loss as $\mathcal{L}_t(w) = w^\top \Sigma_\phi(t)w - 2w^\top b_t + \mathbb{E}[y_t^2]$ (obtained by taking expectation over critic loss shown in Eq. 6).

Under continuous-time gradient flow, the dynamics are given by $\dot{w}(t) = -\nabla_w \mathcal{L}_t(w) = -2\Sigma_\phi(t)w + 2b_t$. At each time $t$, the instantaneous minimizer is $w_t^* = \Sigma_\phi(t)^{-1}b_t$, which varies over time due to non-stationarity in the targets, policy, and representation distribution. We define the tracking error as $e(t) := w(t) - w_t^*$, and study stability using the Lyapunov function $\Gamma := \|e(t)\|_2^2$.

**Theorem 3.1** (Tracking error dynamics)**.** *Assume that* $\Sigma_\phi(t) \succ 0$ *is constant over time (e.g., enforced by regularization) and that $b_t$ is differentiable. Under gradient flow, the time derivative of* $\Gamma$ *satisfies*

$$\dot{\Gamma} = -4\, e(t)^\top \Sigma_\phi(t)\, e(t) \;-\; 2\, e(t)^\top \Sigma_\phi(t)^{-1} \dot{b}_t. \quad (1)$$

*Proof sketch.* Writing $\dot{e}(t) = \dot{w} - \dot{w}_t^*$ and using $\dot{w} =$

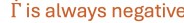

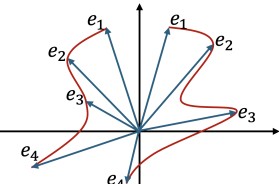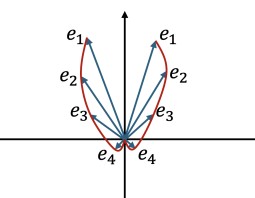

*Figure 1.* **Illustration of two tracking regimes. Left:** the norm of the tracking error exhibits non-monotonic behavior and fails to converge, indicating unstable tracking. **Right:** the tracking error decreases monotonically and converges to zero, corresponding to stable tracking dynamics.

$-2\Sigma_\phi w + 2b_t$ together with $w_t^* = \Sigma_\phi^{-1} b_t$ (and fixed $\Sigma_\phi$), we obtain $\dot{e}(t) = -2\Sigma_\phi e(t) - \Sigma_\phi^{-1} \dot{b}_t$. The claim follows from $\dot{\Gamma} = 2e(t)^\top \dot{e}(t)$. See Appendix B for the full proof. □

**Geometric Implications for Stability.** The decomposition in Eq. 1 highlights competing effects governing stability under non-stationary targets. The first term (in blue) is a strict contraction whose strength depends on the spectrum of $\Sigma_\phi$, while the second term (in red) captures drift induced by target non-stationarity through $\dot{b}_t$. The geometry of the learned representation plays a central role in determining whether the tracking error decays or grows over time.

1. **Tracking-error contraction.** Stability requires $\dot{\Gamma} < 0$ for all error directions, ensuring monotonic decay of the tracking error (see Fig. 1). While the contraction term $-4\,e^\top \Sigma_\phi e$ is always negative, its magnitude depends on the conditioning of $\Sigma_\phi$ and can vary significantly across directions.
2. **Effect of isotropy under non-stationarity.** The drift term $-2\,e^\top \Sigma_\phi^{-1} \dot{b}_t$ is amplified along poorly conditioned directions. Under a fixed total variance budget, anisotropic representations increase the likelihood that some directions exhibit weak contraction and large drift. Enforcing isotropy equalizes contraction across directions and reduces the probability that $\dot{\Gamma}$ becomes positive.
3. **Gaussian representations and tail behavior.** Among distributions with fixed second-order moments, Gaussian distributions maximize entropy and exhibit tail decay. This limits the occurrence of rare but harmful error directions that could dominate the drift term, while preserving maximal information content in the representation.

### 3.2. Why Isotropy Stabilizes Tracking Dynamics

To ensure that $\dot{\Gamma}$ is negative for all tracking error vectors, the contraction induced by the first term in Eq. 1 must dominate

the second term, whose sign may vary. The strength of this contraction is governed by the geometry of the covariance matrix $\Sigma_\phi$. Restricting the tracking error to the unit sphere, the weakest contraction occurs along the eigenvector corresponding to the smallest eigenvalue of $\Sigma_\phi$. Equalizing the contraction across all directions therefore requires distributing the total variance uniformly across dimensions, which implies that all eigenvalues of $\Sigma_\phi$ are equal. The second term in Eq. 1 can be either positive or negative, and large magnitudes may destabilize the dynamics. As shown in Appendix B.1.4, its absolute value is upper bounded by the condition number of $\Sigma_\phi$. Minimizing this bound requires the smallest possible condition number, which is achieved precisely when the covariance matrix is isotropic.

### 3.3. Why Gaussianity Controls Drift Variance

Among all isotropic distributions, the Gaussian is particularly well suited for stabilizing the dynamics induced by Eq. 1. The second term in this equation can fluctuate in sign and may occasionally dominate the contraction term, leading to positive values of $\dot{\Gamma}$. As shown in Appendix B.1.5, a Gaussian distribution minimizes the variance of this term, thereby reducing the likelihood of large destabilizing deviations. Distributions with heavier tails introduce higher-order moment contributions that increase this risk. This effect can be understood through moment concentration. The embedding vectors enter the dynamics through $\dot{b}_t$, and for any random variable $X$, $\mathbb{P}(|X - \mu| \geq t) \leq \mathbb{E}[|X - \mu|^p]/t^p$.

Large higher-order moments therefore translate directly into rare but severe spikes in the magnitude of the second term. By Isserlis' theorem (Isserlis, 1918), all higher-order moments of a Gaussian distribution are fully determined by its covariance, preventing such uncontrolled fluctuations. In other words, while different distributions may share the same covariance, non-Gaussian ones can differ in higher-order moments, which are not controlled by eigenvalues of the covariance matrix. From an information-theoretic perspective, a Gaussian distribution further maximizes entropy among all continuous distributions with a fixed covariance. As a result, it achieves the least-structured, most unbiased representation compatible with isotropy, balancing expressiveness with stability.

### 3.4. Sketched Isotropic Gaussian Regularization

In Sections 3.2 and 3.3, we argued that isotropy and Gaussian tail behavior are desirable properties for stabilizing learning under non-stationarity. This raises two practical questions: *(i)* how to measure deviations from an isotropic Gaussian representation distribution, and *(ii)* how to enforce such structure efficiently during training. To this end, we adopt *Sketched Isotropic Gaussian Regularization* (SI-GReg) (Balestriero & LeCun, 2025), a lightweight regular-

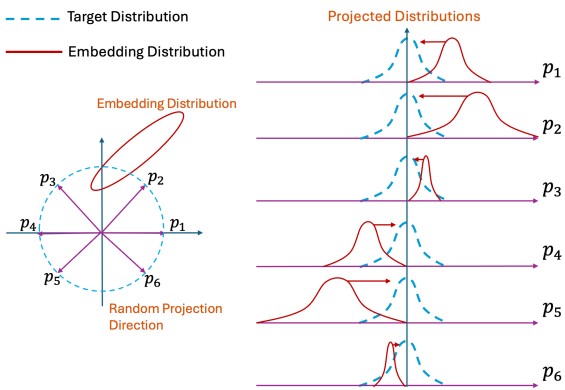

*Figure 2.* **Directly shaping a multivariate distribution.** SIGReg first projects the embeddings onto a small set of random directions ($p_i$: sketching), producing multiple univariate distributions. Each projection is then matched to the corresponding univariate target distribution.

izer recently introduced in self-supervised learning. SIGReg avoids directly matching high-dimensional distributions by projecting embeddings onto a small number of random directions and applying a univariate distribution-matching loss to each projection. By resampling directions across iterations, SIGReg enforces isotropy and Gaussianity in expectation while remaining computationally efficient.

Formally, given an embedding $\phi \in \mathbb{R}^d$ and random unit vectors $\{v_k\}_{k=1}^K$, SIGReg operates on the projected variables $z_k = v_k^\top \phi$. The regularization loss matches the empirical distribution of each $z_k$ to a zero-mean Gaussian with variance $\sigma^2$ using a characteristic-function-based objective (Balestriero & LeCun, 2025) (see Fig. 2). This construction yields a fully differentiable regularizer that simultaneously controls isotropy and tail behavior. In our experiments, we study the impact of each component through targeted ablations.

---

**Key Insight: Isotropy and Stability**

When the feature covariance satisfies $\Sigma_\phi = \sigma^2 I$, the contraction term becomes isotropic and the tracking error is uniformly damped across all directions. In contrast, ill-conditioned representations induce anisotropic contraction, amplifying the effect of target drift through $\Sigma_\phi^{-1} \dot{b}_t$. As a consequence, non-isotropic embeddings can lead to unstable or directionally biased tracking behavior under non-stationarity.

---

### 3.5. Empirical Validation

To further validate the theoretical motivation for isotropic Gaussian representations, we study a controlled non-stationary linear regression setting in which SGD must con-

tinuously track a time-varying optimal solution (see App. F). Across a wide range of feature distributions, we find that isotropic Gaussian features consistently yield the most stable optimization dynamics, characterized by rapid and nearly monotonic contraction of the tracking error following task switches. In contrast, increasing anisotropy slows convergence and introduces instability, while heavy-tailed Laplace features produce frequent transient increases in tracking error even when the covariance remains isotropic.

These trends persist across multiple condition numbers, random seeds, and higher-dimensional settings, indicating that both isotropy and distributional shape play a fundamental role in optimization stability. Furthermore, we identify specific low-variance directions induced by anisotropic covariances along which SGD can exhibit persistent drift and even divergence. These results provide empirical evidence that isotropy and Gaussianity improve tracking stability under non-stationarity, motivating the use of an isotropic Gaussian target embedding distribution in our approach.

## 4. Empirical Results

### 4.1. CIFAR-10 Under Distribution Shift

To isolate the effect of non-stationarity independently of control, exploration, and environment dynamics, we first study a controlled supervised setting based on CIFAR-10 (Krizhevsky et al., 2009). Following prior work (Igl et al., 2021; Sokar et al., 2023; Lyle et al., 2024; Castanyer et al., 2026; Obando-Ceron et al., 2026), we introduce non-stationary targets via a shuffled-label protocol, in which label assignments are periodically permuted during training. This induces a sequence of related but non-identical prediction problems, serving as a proxy for the drifting targets arising from bootstrapped updates in deep RL. This setup is widely used to study representation drift and neuron dormancy under non-stationarity, as it exposes similar representational stresses while avoiding confounding effects from policy learning and exploration (Sokar et al., 2023; Castanyer et al., 2026). Operating in a fully supervised regime allows us to directly analyze how representation geometry evolves under target drift. All experimental hyperparameters and training details are reported in App. E.

We compare standard training against models encouraged toward isotropic representations, keeping the architecture, optimizer, and learning rate fixed. We track classification performance, effective rank, and the fraction of dormant neurons under distribution shift (see Fig. 3)[1]. Under standard training, representations quickly become anisotropic, with variance concentrating along few directions, accompanied by rank collapse and reduced adaptability. By contrast, encouraging isotropy preserves balanced variance, reduces

---

[1] Unless otherwise specified, results are averaged over 5 seeds.

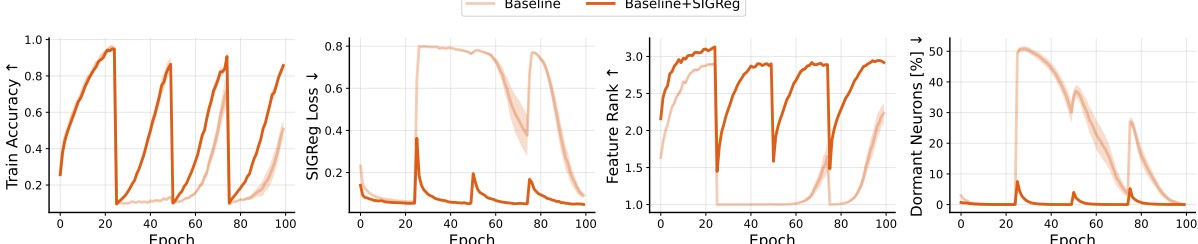

*Figure 3.* **Non-stationary CIFAR-10.** Training under repeated label shuffling. The baseline shows poor recovery after each shift, with SIGReg loss spikes, rank collapse, and increased dormancy. Enforcing isotropic Gaussian representations stabilizes training, accelerates recovery, preserves rank, and reduces dormancy.

neuron dormancy, and leads to more stable performance and faster recovery after shifts. These effects closely mirror behaviors observed in deep RL under policy-induced distributional drift.

## 4.2. Deep Reinforcement Learning

We evaluate isotropic Gaussian representations in deep RL, where non-stationarity arises naturally from policy improvement and bootstrapped learning targets. We focus on online deep RL agents trained in non-stationary regimes, where the distribution of visited states evolves continuously as the policy improves. This setting is particularly sensitive to representation collapse, gradient interference, and plasticity loss, which can silently degrade effective capacity even when training appears numerically stable.

**Setup.** We evaluate the effect of enforcing isotropic Gaussian representations in Parallelized Q-Networks (PQN) (Gallici et al., 2025). Experiments are conducted on the Atari-10 subset (Aitchison et al., 2023) of the Arcade Learning Environment (ALE) (Bellemare et al., 2013), following standard evaluation protocols (Agarwal et al., 2021). For each experiment, we compare a baseline implementation against an identical version in which representations are regularized toward an isotropic Gaussian distribution. All deep RL agents use the same convolutional encoders, optimizer hyperparameters, and training schedules (see App. E). The SIGReg regularizer is applied only to the learned representations and does not modify the policy or value objectives.

We empirically analyze the effect of encouraging isotropic Gaussian representations via explicit minimization of the SIGReg loss. We observe consistent improvements in representation quality, reduced neuron dormancy, and improved asymptotic performance. These results indicate that promoting isotropic Gaussian structure in representations mitigates common optimization pathologies in deep RL.

**Effect on Representation Rank.** Encouraging isotropic Gaussian structure in the learned representations leads to a systematic increase in embedding rank throughout training.

This behavior is consistent with the objective of maintaining well-conditioned covariance structure, preventing collapse along low-variance directions. For PQN, the evolution of feature rank shows that representations remain high-rank over time, unlike the baseline, which exhibits progressive rank degradation. This indicates that enforcing isotropy stabilizes representation geometry under non-stationary training, preserving expressive capacity as learning progresses (see Figures 4 and 5).

**Reduction of Neuron Dormancy.** Alongside improved rank, isotropic Gaussian representations substantially reduce neuron dormancy in PQN. Maintaining high-entropy, well-spread representations prevents activations from concentrating on a small subset of units, thereby sustaining gradient flow across the network. As a result, fewer neurons become inactive over time, mitigating a common form of plasticity loss in deep RL. This suggests that representation geometry plays a direct role in preserving network capacity under continual bootstrapping (see Fig. 4).

**Temporal Evolution of Representation Distributions.** To better understand how isotropic Gaussian structure manifests during training, we analyze the temporal evolution of the learned representation distributions under PQN. Fig. 5 visualizes this evolution using a two-dimensional PCA projection of the embedding covariance at successive training stages. Without explicit constraints on representation geometry, the embeddings progressively collapse onto a small number of dominant principal directions, resulting in highly anisotropic distributions with most variance concentrated in a few components. In contrast, when representations are encouraged to follow an isotropic Gaussian structure, the projected covariance becomes approximately circular and centered at the origin. This indicates both isotropy and a more uniform allocation of variance across dimensions, corresponding to a higher effective dimensionality. These observations provide direct empirical evidence that isotropic Gaussian representations counteract representational collapse and maintain well-conditioned feature spaces throughout training. See App. H for further discussion.

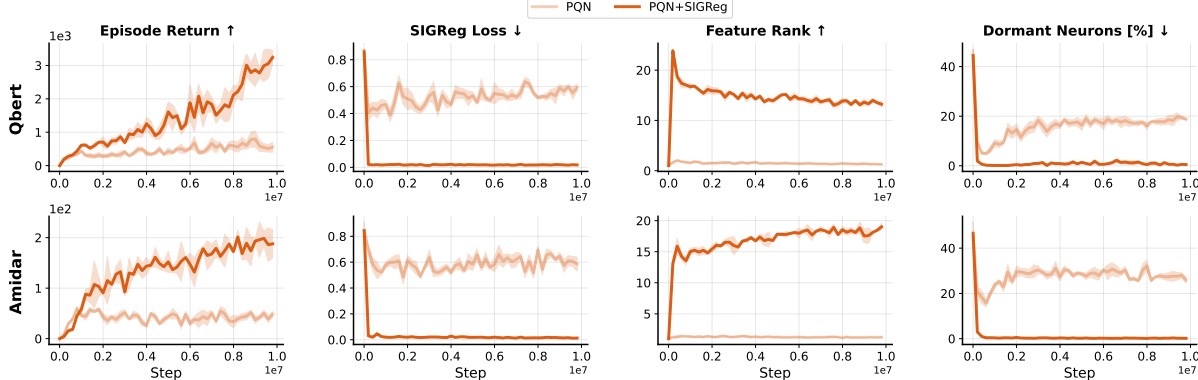

*Figure 4.* **Two Atari-10 games (PQN).** Without isotropy regularization, representations exhibit rank collapse, increased neuron dormancy, and early performance saturation. Encouraging isotropic geometry leads to improved representation quality and higher, more stable performance.

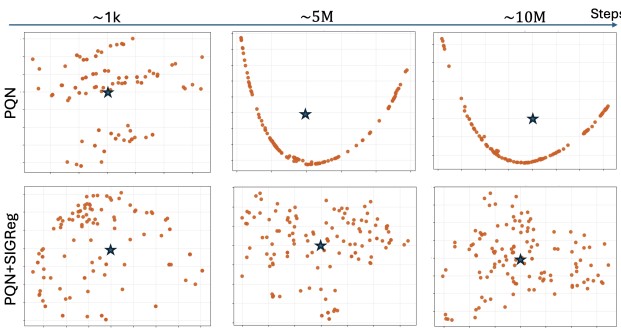

*Figure 5.* **2D PCA of embedding covariance over training.** Without constraints, representations collapse onto a few dominant principal components. Encouraging isotropic Gaussian structure yields more evenly distributed variance and higher effective dimensionality, reflected in reduced concentration on the leading components (PQN: $[0.4, 0.2] \rightarrow [0.9, 0.8]$ vs. PQN+SIGReg: $[0.3, 0.2] \rightarrow [0.1, 0.1]$).

**Performance Implications in PQN.** We next examine whether these representational improvements translate into gains in learning performance. Using Area Under the Curve (AUC) as our primary metric, which captures both learning speed and final performance, we observe consistent improvements over the baseline PQN agent (Gallici et al., 2025). These gains indicate that stabilizing representation geometry through isotropic Gaussian structure not only improves internal metrics such as rank and dormancy, but also yields tangible benefits in control performance. Importantly, these improvements are achieved without introducing additional architectural complexity or second-order optimization, highlighting representation regularization as a lightweight and effective mechanism for stabilizing PQN training. Fig. 4 shows improved sample efficiency and final performance on two Atari games when promoting isotropic Gaussian representations. See Fig. 6 for additional results across more ALE games (Bellemare et al., 2013; Aitchison et al., 2023).

## 4.3. Analysis of Design Choices

In this section, we analyze how different design choices for promoting isotropic representations affect learning under PQN. Specifically, we study the impact of (i) alternative isotropic target distributions with heavier tails or imposing isotropy alone through covariance whitening, as proposed in VICReg (Bardes et al., 2022) in the self-supervised learning literature, and (ii) enforcing only symmetry (minimizing the imaginary part) or only tail behavior (minimizing the real part). These ablations allow us to disentangle which statistical properties of isotropic Gaussian representations are most critical for stable and efficient deep RL.

**Alternative Isotropic Distributions.** We first compare isotropic Gaussian representations with alternative isotropic distributions exhibiting heavier tails, namely Laplacian and Logistic distributions. While heavier-tailed distributions may be sufficient in some regimes, our results indicate that they are consistently less effective under PQN. As shown in Table 2, isotropic Gaussian representations yield the strongest and most reliable gains, improving approximately 90% of games with large average relative improvements. In contrast, Laplacian and Logistic distributions achieve smaller gains despite improving a similar fraction of environments. It has also been observed that achieving isotropy via covariance whitening is not enough (last row in Table 2). This suggests that, beyond isotropy, fast tail decay plays a central role in stabilizing learning and preventing extreme activations under non-stationary targets.

**Role of Symmetry and Tail Decay.** To further isolate the role of different statistical properties, we evaluate variants that enforce only symmetry (imaginary part) or only tail behavior (real part). Table 2 shows that neither property alone matches the performance of full isotropic Gaussian representations. Enforcing tail behavior is generally more effective than enforcing symmetry alone, likely because

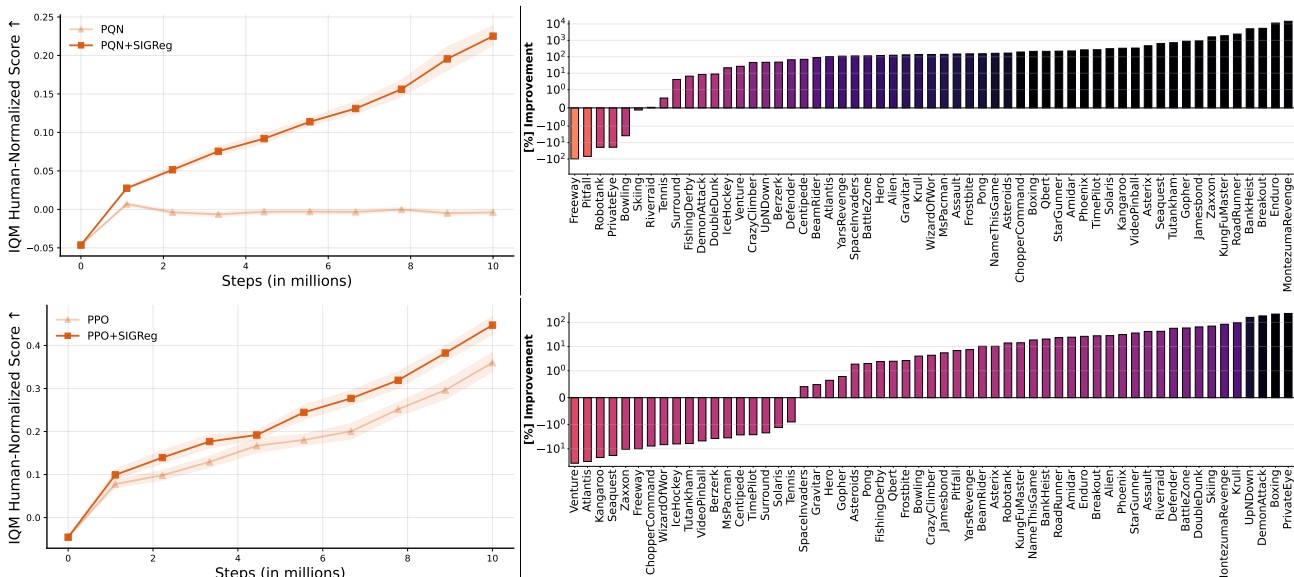

*Figure 6.* **Full Atari suite. Effect of isotropic Gaussian regularization. Left:** IQM human-normalized learning curves as a function of environment steps for PQN and PPO, with and without isotropic regularization. **Right:** Per-game improvement in AUC obtained by encouraging isotropic representation geometry. Across both algorithms, isotropic regularization improves final performance and sample efficiency.

intrinsic regularization already induces approximate symmetry. However, jointly enforcing both consistently gives the strongest and most stable improvements, showing that the benefit comes from their combination rather than from either property in isolation.

### 4.4. Implicit Isotropy in Stabilization Methods

We further investigate the connection between isotropic Gaussian representations and strong stabilization mechanisms introduced by Castanyer et al. (2026), namely Kronecker-factored optimization and multi-skip residual architectures, both of which were designed to mitigate gradient pathologies and improve stability at scale. Fig. 9 reports results for Parallelized Q-Networks (PQN) for Atari-10 games. Across nearly all games, Kronecker-factored optimization consistently induces representations that are substantially closer to isotropic Gaussian distribution (lower SIGReg loss) than those learned with first-order methods such as RAdam. Importantly, this effect emerges without any explicit representation regularization. Similarly, multi-skip architectures, which were introduced to stabilize gradient propagation, also implicitly promote better-conditioned and more isotropic representations (see Fig. 10).

Although the Kronecker-factored optimizer can be effective and provide significant improvements, it requires additional memory and computation due to gradient curvature estimation. Therefore, it is desirable to achieve similar performance with first-order optimizers. Based on observations from Fig. 9, we hypothesize that part of the stability and performance gains provided by these optimizers are the

effect of the way they shape the geometry of the learned representations. Therefore, if we explicitly shape the embeddings of a baseline model, we should expect to see a reduction in the performance gap. Consistent with this idea, Fig. 14 and Table 3 show that enforcing isotropic and Gaussian representations through the auxiliary SIGReg objective significantly narrows the gap between the baseline and Kronecker-factored optimizer. Importantly, this improvement is obtained without adding significant computational or memory overhead.

### 4.5. Comparison with Alternative Methods.

To determine whether the benefits of isotropic Gaussian representations arise from generic representation regularization or from the specific combination of isotropy and Gaussianity, we compare SIGReg against several alternative approaches that have previously been shown to improve representation quality. These include covariance whitening (VICReg), embedding norm regularization, simplicial embeddings, and weight orthogonalization. While many of these methods improve performance relative to the baseline, isotropic Gaussian regularization achieves the strongest overall gains, improving 90% of games and yielding the highest average improvement across Atari-10 (Table 10).

These results suggest that the observed gains cannot be fully explained by generic anti-collapse mechanisms, feature spreading, or rank maximization alone. For example, VICReg promotes isotropy but does not explicitly enforce Gaussianity, while weight orthogonalization encourages diverse features through constraints on the network weights.

Despite their benefits, neither matches the average improvement obtained by isotropic Gaussian representations. These findings provide further evidence that the combination of isotropy and Gaussianity constitutes a particularly effective inductive bias for learning robust representations under non-stationarity.

### 4.6. Full Atari Suite

We next evaluate isotropic Gaussian representations at scale on the full Atari benchmark using PQN. Fig. 6 reports per-game improvements in area under the learning curve (AUC) relative to a RAdam baseline. Encouraging isotropic Gaussian structure in the learned representations yields broad and consistent gains across the suite. Out of 57 games, 51 (89.5%) exhibit improved performance, with a mean AUC improvement of 889% and a median improvement of 138%. Crucially, these improvements are not driven by a small number of outlier environments, but are distributed across games with diverse dynamics, reward structures, and exploration challenges. These results demonstrate that isotropic Gaussian representations scale reliably to large and heterogeneous benchmarks, providing a simple and effective mechanism for improving both learning efficiency and final performance in deep RL.

### 4.7. Beyond Atari and PQN

**Policy Gradient Methods (PPO).** Motivated by the results that isotropic Gaussian representation leads to significant improvements in value-based methods (Section 4.6), we next evaluate whether similar geometric effects, and their benefits, extend to policy-gradient algorithms. We focus on Proximal Policy Optimization (PPO) (Schulman et al., 2017), which differs substantially from PQN in both optimization dynamics and update structure. Although PPO is often regarded as more stable, we observe that it still suffers from representation collapse and neuron dormancy under long training horizons and non-stationary data distributions (Moalla et al., 2024; Mayor et al., 2025).

Across the full Atari suite, encouraging isotropic representation geometry leads to consistent improvements in IQM human-normalized performance and higher AUC in the majority of games (see Fig. 6). Importantly, stabilization strategies that operate at the optimization or architectural level, such as Kronecker-factored optimization or multi-skip residual connections, do not reliably transfer to PPO and can even degrade performance (see Fig. 14 and Table 3). By contrast, modestly encouraging isotropy at the representation level yields improvements without altering the PPO update rule or introducing algorithm-specific modifications. These results reinforce the interpretation that isotropic representations effectively address optimization issues under non-stationarity and that this effect generalizes beyond value-based methods.

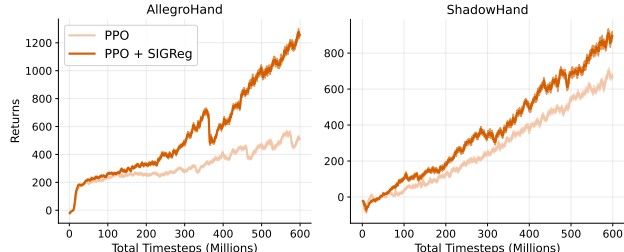

*Figure 7.* **Isaac Gym continuous control.** Learning curves on two representative locomotion tasks. Isotropic Gaussian representations improve stability and reduce variance. We report returns over 5 runs for each experiment. See Section D for additional results on four Isaac Gym control tasks.

We observe larger gains in PQN than in PPO in Fig. 6, which may be explained by differences in how the two algorithms interact with non-stationarity. PQN relies on bootstrapped value estimates and replayed experience, making learning more sensitive to evolving targets and representation quality. In contrast, PPO operates in an on-policy regime with fresher data and more tightly coupled policy and value updates, which can reduce sources of distribution shift and instability. Consequently, improvements arising from isotropic Gaussian representations may be more pronounced in PQN, where learning dynamics are particularly sensitive to the quality of the underlying features.

**Continuous Control in Isaac Gym.** To assess whether the benefits of isotropic Gaussian representations extend beyond discrete control and pixel-based domains, we evaluate continuous control tasks in Isaac Gym (Makoviychuk et al., 2021). These environments exhibit strong non-stationarity due to contact dynamics, evolving state distributions, and high-dimensional continuous action spaces. Across a range of locomotion and manipulation tasks, encouraging isotropic representation geometry leads to improved training stability and reduced variance across 5 random seeds as shown in Fig. 7.

## 5. Related Work

Deep RL agents operate under inherently non-stationary conditions, as both data distributions and learning targets evolve with the policy. This non-stationarity exacerbates representation drift and plasticity loss, leading to dormant neurons, rank collapse, and reduced effective capacity during training (Lyle et al., 2023; Obando-Ceron et al., 2026; Lyle et al., 2022; Nikishin et al., 2022; Moalla et al., 2024). Such degradation has been linked to performance collapse and diminished adaptability in long-horizon, continual, and online learning settings (Tang et al., 2025). Existing mitigation strategies typically rely on architectural changes, auxiliary losses, neuron reinitialization, or optimization-centric

techniques (Nikishin et al., 2022; Liu et al., 2026; Palenicek et al., 2026; Castanyer et al., 2026), as well as spectral and rank-based diagnostics (Lyle et al., 2023), but do not directly target the statistical structure of learned representations.

A complementary line of work improves representation stability through auxiliary objectives inspired by self-supervised learning (Echchahed & Castro, 2025). These include contrastive methods such as CURL (Laskin et al., 2020), predictive approaches like SPR (Schwarzer et al., 2021), and metric-based objectives such as MiCO (Castro et al., 2021), which mitigate representation collapse and neuron dormancy (Kumar et al., 2021; Lyle et al., 2021; Sokar et al., 2023; Liu et al., 2026; Obando-Ceron et al., 2026). See App. A for further discussion.

## 6. Conclusion

We studied deep RL through the lens of representation geometry and showed that isotropic Gaussian representations provide a principled and effective solution to instability under non-stationarity. Our analysis formalizes representation learning as a tracking problem with drifting targets and establishes that, under isotropic Gaussian embeddings, the zero tracking-error equilibrium is stable, with bounded and decreasing error dynamics. This theoretical result offers a clear explanation for why certain representation structures are intrinsically robust to evolving objectives. We evaluated SIGReg as a lightweight and practical mechanism for shaping representations toward this favorable geometry. Across controlled non-stationary supervised settings and large-scale deep RL benchmarks, SIGReg consistently improved training stability, reduced representation collapse and neuron dormancy, and led to substantial performance gains.

**Discussion.** A defining challenge of deep RL is that representation learning and control are inseparable: as policies improve, both data distributions and learning targets evolve, so representations are never trained against a fixed objective. Non-stationarity is therefore a structural property of deep RL, not a secondary source of noise. Most prior work addresses this challenge indirectly, for example through optimization stabilization or variance reduction, implicitly treating representations as passive byproducts of training. Our results suggest this view is incomplete: under non-stationary supervision, representation geometry plays a central role in determining learning stability. Viewing representation learning as a tracking problem with drifting targets clarifies why anisotropic or low-entropy embeddings are fragile. Such representations concentrate capacity along directions favored by transient targets, amplifying sensitivity to drift and accelerating the loss of effective capacity. From this perspective, representation collapse and neuron dormancy are predictable consequences of unconstrained geometry. This

motivates representation-level inductive biases that remain well-conditioned as objectives evolve. Simple statistical constraints, such as isotropy and controlled variance, provide a robust foundation for stable and scalable learning under continual change.

**Limitations.** This work focuses on shaping the marginal distribution of learned representations and does not explicitly enforce task-specific structure or semantic alignment. While isotropic Gaussian representations provide a strong default prior under uncertainty and non-stationarity, they may be suboptimal for tasks requiring highly structured features, and balancing isotropy with task-adaptive biases remains an open challenge. Our analysis assumes simplified linear readouts and approximate stationarity of the representation covariance; extending these results to nonlinear heads, fully coupled off-policy actor–critic algorithms (Seo et al., 2025), and continual deep RL settings (Tang et al., 2025) is an important direction for future work.

## Acknowledgments

The authors would like to thank Roger Creus and Walter Mayor Toro for valuable discussions during the preparation of this work. We would also like to give special thanks to Jesse Farebrother for providing valuable feedback on an early draft of the paper.

The research was enabled in part by computational resources provided by the Digital Research Alliance of Canada (`https://alliancecan.ca`) and Mila (`https://mila.quebec`). Pablo Samuel Castro acknowledges funding from NSERC Discovery Grant. We acknowledge funding support from Google and CIFAR AI. We would also like to thank the Python community (Van Rossum & Drake Jr, 1995; Oliphant, 2007) for developing tools that enabled this work, including NumPy (Harris et al., 2020), Matplotlib (Hunter, 2007), Jupyter (Kluyver et al., 2016), and Pandas (McKinney, 2013).

## Impact Statement

This paper presents work whose goal is to advance the field of Machine Learning. There are many potential societal consequences of our work, none of which we feel must be specifically highlighted here.

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

# Appendix Contents

# A. Related Work

**Non-Stationarity and Deep RL.** Deep RL agents are trained under inherently non-stationary conditions, as both the data distribution and learning targets evolve with the policy. This non-stationarity exacerbates representation drift and plasticity loss, often resulting in dormant neurons and degraded learning dynamics (Lyle et al., 2023; Obando-Ceron et al., 2026). Recent empirical studies have shown that deep RL agents progressively lose effective capacity during training, manifested as a reduction in representation rank and an increasing fraction of inactive units (Lyle et al., 2022; Nikishin et al., 2022; Ceron et al., 2023; Moalla et al., 2024; Liu et al., 2025). Such degradation has been linked to performance collapse and reduced adaptability in long-horizon, continual, and online learning settings (Tang et al., 2025).

Prior approaches to mitigate plasticity loss typically focus on architectural modifications, auxiliary losses, or explicit neuron recycling and reinitialization strategies (Ceron et al., 2024c; Sokar et al., 2025; Nikishin et al., 2022; Liu et al., 2026). Other work has analyzed representational collapse and capacity loss through spectral and rank-based diagnostics, highlighting their prevalence across algorithms and domains (Lyle et al., 2023). Castanyer et al. (2026) showed that mitigating gradient degradation through approximate second-order optimization and multi-skip information propagation can alleviate plasticity loss, enabling more stable scaling of deep RL architectures. More recently, (Obando-Ceron et al., 2026) evaluated simplicial embeddings as a geometric inductive bias on latent representations, showing that structured and sparse feature spaces can stabilize critic bootstrapping and improve sample efficiency in actor–critic methods, alleviating feature collapse as a consequence of non-stationarity. While effective, these approaches primarily target optimization dynamics or architectural design, rather than directly shaping the statistical structure of learned representations.

**Representation Learning in Deep RL.** Learning stable and expressive representations is a long-standing challenge in deep RL. A substantial body of work has introduced auxiliary objectives to regularize representations and improve stability (Echchahed & Castro, 2025), often inspired by self-supervised learning (Balestriero et al., 2023). Contrastive methods such as CURL (Laskin et al., 2020) encourage feature diversity, while predictive approaches such as SPR (Schwarzer et al., 2021) promote temporal consistency through future-state prediction. Metric-based objectives (Desharnais et al., 2004; Ferns et al., 2012; Castro, 2020; Zang et al., 2023) such as MiCO (Castro et al., 2021) aim to align representational geometry with behavioral similarity.

More broadly, auxiliary losses and representation regularization techniques have been shown to mitigate representation collapse and neuron dormancy in deep RL (Kumar et al., 2021; Lyle et al., 2021; Sokar et al., 2023; Liu et al., 2026; Obando-Ceron et al., 2026). While effective, these methods typically introduce additional prediction heads, task-specific objectives, or architectural complexity, and their benefits can be sensitive to hyperparameter choices and domain characteristics. For instance, Farebrother et al. (2023) propose *Proto-Value Networks*, which scale representation learning by training on a large family of auxiliary tasks derived from the successor measure. Similarly, Cetin et al. (2023) show that imposing an explicit *geometric prior* on the latent space, by learning representations in hyperbolic space, can improve performance and generalization. Fujimoto et al. (2025) pursue general-purpose model-free RL by leveraging learned representations inspired by model-based objectives that approximately linearize the value function, enabling competitive performance across diverse benchmarks with a single set of hyperparameters.

In contrast to auxiliary-task-based approaches (Gelada et al., 2019; Zhang et al., 2021; Fujimoto et al., 2023; Obando-Ceron et al., 2026), we focus on shaping representation geometry directly through a lightweight statistical regularizer. Our approach is inspired by recent advances in self-supervised learning, which demonstrate that enforcing simple statistical constraints, such as isotropy and Gaussianity, can be sufficient to yield stable representations under non-stationary targets. By encouraging isotropic Gaussian structure at the representation level, our method complements existing approaches while remaining simple, computationally efficient, and broadly applicable across deep RL algorithms.

**Stability Analysis of Linear Time-Variant Systems.** Stability analysis of linear time-varying systems has been extensively studied in the control systems literature. Classical approaches, such as Lyapunov stability analysis of Kalman-Bucy filters (Del Moral & Tugaut, 2018), establish convergence guarantees under fixed structural assumptions and well-specified dynamical models. More broadly, stochastic stability and convergence in adaptive systems have been studied in stochastic approximation theory (Benveniste et al., 2012; Kushner & Yin, 2003). Our work takes a complementary perspective: rather than assuming fixed embedding properties, we ask what regularization constraints lead to optimal embedding structures for stable task tracking. We characterize how specific properties of learned embedding distributions, such as isotropy and tail behavior, directly influence contraction rates and drift terms in non-stationary settings common in DRL. This geometric perspective, which explicitly optimizes representation properties for convergence rather than only analyzing convergence

under given representations, is novel in the DRL literature.

## B. Formal Analysis

### B.1. In the presence of non-stationary tasks, Isotropic Gaussian makes zero tracking error a stable equilibrium

To simplify the analysis, we consider a linear critic on top of the embedding:

$$Q_\theta(s, a) := w^\top \phi(s, a) \tag{2}$$

where

- $\phi(s, a) \in \mathbb{R}^d$ is the **penultimate-layer embedding**

- $w \in \mathbb{R}^d$ is the last-layer weight vector

**Probability space.** All expectations are taken with respect to an underlying probability space $(\Omega, \mathcal{F}, \mathbb{P})$ induced by interaction with the environment. In particular, the embedding $\phi(s, a)$ is a random variable induced by the state-action visitation distribution $(s, a) \sim d_t^\pi$, where $d_t^\pi$ denotes the (possibly time-varying) occupancy measure under policy $\pi$. The TD target $y_t = r + \gamma Q_{\theta^-}(s', a')$ is a random variable induced by trajectory sampling $\tau \sim \mathbb{P}_t^\pi(\tau)$, where trajectories are generated from the initial state distribution, policy $\pi$, and environment dynamics.

Define the TD target (the target is time-varying which is the source of non-stationarity):

$$y_t = r + \gamma Q_{\theta^-}(s', a') \tag{3}$$

The expected critic loss is

$$\mathcal{L}_t(w) = \mathbb{E}\left[\left(w^\top \phi - y_t\right)^2\right] \tag{4}$$

If we expand the loss term:

$$\mathcal{L}_t(w) = \mathbb{E}\left[\left(w^\top \phi - y_t\right)\left(\phi^\top w - y_t^\top\right)\right] \tag{5}$$

$$\mathcal{L}_t(w) = \mathbb{E}[w^\top \phi \phi^\top w] - 2\mathbb{E}[y_t \phi^\top w] + \mathbb{E}[y_t^2] \tag{6}$$

Define:

$$\Sigma_\phi(t) := \mathbb{E}_{(s,a)\sim d_t^\pi}[\phi \phi^\top], \qquad b_t := \mathbb{E}_{\tau \sim \mathbb{P}_t^\pi}[\phi y_t]. \tag{7}$$

where $\Sigma_\phi(t)$ is the covariance matrix of the embedding vectors and $b_t$ is the drift term caused by non-stationarity. Both $\Sigma_\phi(t)$ and $b_t$ are defined under the same trajectory distribution $\mathbb{P}_t^\pi$, which induces statistical dependence between representation geometry and the TD drift term. Then:

$$\boxed{\mathcal{L}_t(w) = w^\top \Sigma_\phi(t) w - 2w^\top b_t + \mathbb{E}[y_t^2]} \tag{8}$$

The gradient of the loss w.r.t the weight of the final layer:

$$\nabla_w \mathcal{L}_t(w) = 2\Sigma_\phi(t)w - 2b_t \tag{9}$$

the final term was eliminated due to assuming target weights to be independent of the actual weights. We analyze continuous-time gradient flow:

$$\boxed{\dot{w}(t) = -\nabla_w \mathcal{L}_t(w) = -2\Sigma_\phi(t)w + 2b_t} \tag{10}$$

At each time $t$, the instantaneous minimizer for the weight matrix of the final layer can be calculated as follows:

$$\nabla_w \mathcal{L}_t(w_t^*) = 0 \tag{11}$$

This gives:

$$\boxed{w_t^* = \Sigma_\phi(t)^{-1} b_t} \tag{12}$$

This optimal weight moves in time as a result of factors imposing non-stationarity. Our goal in this analysis is to:

1. Define the tracking error for the weights of last layer.

2. Derive the formula for how this tracking error changes.

3. Derive the energy, Lyapunov, function for the norm of tracking error and show that isotropic Gaussian structure makes the zero equilibrium stable, meaning that an increase in the norm of tracking error will be damped and converge to zero over time.

We define the tracking error at time $t$ as the difference between the weight matrix and the optimal unknown weights:

$$e(t) := w(t) - w_t^* \tag{13}$$

Our goal is to analyze under what conditions $||e||_2^2 = 0$ is contractive. We focus on this equilibrium because it shows whether, when $||e||_2$ increases due to non-stationarity, it eventually returns to zero. To perform this analysis, we adopt Lyapunov stability analysis common in the analysis of the stability of equilibria of dynamical systems (Massera, 1949; Lefschetz & LaSalle, 1961). We choose a quadratic function as the Lyapunov function, which represents the energy of the system:

$$\Gamma = ||e||_2^2. \tag{14}$$

The condition for stability of $||e||_2^2 = 0$ is that the derivative of the Lyapunov function be negative, meaning that an increase in $||e||_2$, and therefore the energy of the system, leads to contraction of the dynamics to the equilibrium $||e||_2^2 = 0$.

**Theorem B.1** (Tracking error dynamics). *Assume that $\Sigma_\phi(t) \succ 0$ is constant over time (e.g., enforced by regularization) and that $b_t$ is differentiable. Under gradient flow, the time derivative of $\Gamma$ satisfies*

$$\dot{\Gamma} = -4\, e(t)^\top \Sigma_\phi(t)\, e(t)\ -\ 2\, e(t)^\top \Sigma_\phi(t)^{-1} \dot{b}_t. \tag{15}$$

*Proof.* To analyze the stability of $||e||_2^2 = 0$, we need the time derivative of tracking error term:

$$\dot{e} = \dot{w} - \dot{w}_t^* \tag{16}$$

Substitute $\dot{w}$:

$$\dot{e} = (-2\Sigma_\phi w + 2b_t) - \dot{w}_t^* \tag{17}$$

$$\dot{e} = -2\Sigma_\phi(e + w_t^*) + 2b_t - \dot{w}_t^* \tag{18}$$

Since $\Sigma_\phi w_t^* = b_t$:

$$\dot{e} = -2\Sigma_\phi(t)e - \dot{w}_t^* \tag{19}$$

We are interested in writing this dynamic formula in terms of the embedding and drift. To write $\dot{w}_t^*$ in terms of the embedding and drift, recall:

$$w_t^* = \Sigma_\phi^{-1} b_t \tag{20}$$

From matrix calculus (Petersen et al., 2008):

$$\frac{d}{dt}\Sigma^{-1} = -\Sigma^{-1}\dot{\Sigma}\Sigma^{-1}. \tag{21}$$

Thus:

$$\dot{w}_t^* = \Sigma_\phi^{-1}\dot{b}_t - \Sigma_\phi^{-1}\dot{\Sigma}_\phi\Sigma_\phi^{-1}b_t \tag{22}$$

Substitute into error dynamics:

$$\dot{e} = -2\Sigma_\phi e - \Sigma_\phi^{-1}\dot{b}_t + \Sigma_\phi^{-1}\dot{\Sigma}_\phi w_t^* \tag{23}$$

Differentiating the Lyapunov function:

$$\dot{\Gamma} = 2e^\top \dot{e}. \tag{24}$$

Substitute $\dot{e}$:

$$\dot{\Gamma} = -4e^\top \Sigma_\phi e - 2e^\top \Sigma_\phi^{-1} \dot{b}_t + 2e^\top \Sigma_\phi^{-1} \dot{\Sigma}_\phi w_t^*. \tag{25}$$

Writing all terms in terms of embedding, tracking error, and drift:

$$\dot{\Gamma} = \underbrace{-4e^\top \Sigma_\phi e}_{\text{contraction}} \underbrace{-2e^\top \Sigma_\phi^{-1} \dot{b}_t}_{\text{target non-stationarity}} + \underbrace{2e^\top \Sigma_\phi^{-1} \dot{\Sigma}_\phi \Sigma_\phi^{-1} b_t}_{\text{representation drift}} \tag{26}$$

$\square$

Our goal is to show that by regularizing the embedding dimension, certain conditions lead to stability of $\|e\|_2^2 = 0$, or in other words, $\dot{\Gamma} < 0$. Since we assume regularizing the embedding distribution (leading to a fixed desired covariance matrix), we can remove the third term.

### B.1.1. FIRST TERM ANALYSIS

Since the covariance matrix is positive definite, the sign of the first term is always negative. Therefore, this term acts as a contractor, adding a negative component to the Lyapunov function's derivative. This term will not cause a problem as it always increases the stability of the equilibrium.

### B.1.2. SECOND TERM ANALYSIS

Although the sign in front of this term is negative, the whole term could be negative or positive. It depends on the angle between the tracking error and the change in the drift vector under the inverse of the covariance matrix as the metric for calculating the inner product. Therefore, it could be helpful (if negative) or harmful (if positive). Our analysis will focus mainly on limiting and bounding the norm of this term, since the sign is not under our control and we cannot exploit the cases in which this term is helpful.

### B.1.3. THIRD TERM ANALYSIS

The third term is due to the drift in the embedding. As explained before, we are looking for the distribution that we need to choose as the desired distribution in the SIGReg regularizer. Therefore, we can remove this term since we regularize the embedding to be a fixed desired distribution, which makes $\dot{\Sigma}_\phi \approx 0$. Although the sign in front of this term is positive, this term could also be negative in multiple scenarios, for example if $\text{Tr}(\Sigma_\phi)$ is being reduced, making $\dot{\Sigma}_\phi$ negative definite.

In the following subsections, we are going to consider only the first two terms, as we will regularize the distribution to be fixed and as a result covariance will not change, and assume that the $Tr(\Sigma_\phi) = c$. In other words, we have a fixed budget as the variance of the embedding.

### B.1.4. WHY ISOTROPY IS HELPFUL?

To achieve a stable equilibrium, we want to ensure $\dot{\Gamma} < 0$ during training. This condition causes any non-zero tracking error to tend toward zero. Since the first two terms operate on different and independent vectors—the first term only on the tracking error vector and the second term on both the tracking error and drift—our only solution is to control the magnitude of these terms separately.

The first term (contraction) is always negative and helpful for driving the derivative toward negative values, since the covariance matrix is a positive semi-definite matrix. However, we want this term to be large and negative for all possible tracking error vectors.

Assume $\Sigma_\phi \succ 0$ and fix the total variance

$$\text{Tr}(\Sigma_\phi) = \sum_{i=1}^d \lambda_i = c \tag{27}$$

where $\lambda_i$ are the eigenvalues of $\Sigma_\phi$. We always have the bound

$$e^\top \Sigma_\phi e \ \geq \ \lambda_{\min}(\Sigma_\phi) \, \|e\|_2^2 \tag{28}$$

Equality happens when the tracking error is aligned with the eigenvector corresponding to the smallest eigenvalue. We want to control this and ensure there is no direction for the tracking error in which the lower bound is small, as for that direction the contraction term becomes weak and small. Since the average of numbers is always greater than or equal to the smallest number:

$$\lambda_{\min}(\Sigma_\phi) \ \leq \ \frac{1}{d} \sum_{i=1}^{d} \lambda_i = \frac{c}{d} \tag{29}$$

it follows that

$$\min_{\|e\|_2=1} e^\top \Sigma_\phi e = \lambda_{\min}(\Sigma_\phi) \ \leq \ \frac{c}{d} \tag{30}$$

Therefore,

$$\max_{\Sigma_\phi : \operatorname{tr}(\Sigma_\phi)=c} \ \min_{\|e\|_2=1} e^\top \Sigma_\phi e = \frac{c}{d} \tag{31}$$

and equality holds if and only if:

$$\lambda_1 = \lambda_2 = \cdots = \lambda_d = \frac{c}{d} \tag{32}$$

In simple terms, we want to boost the direction with the smallest eigenvalue to ensure no direction will significantly dampen the contraction term. Since the average of eigenvalues is fixed and the smallest eigenvalue will always be less than or equal to the average, the best case is when the smallest eigenvalue equals the average, which occurs when all eigenvalues are the same.

For the second term, since two different vectors are involved, we cannot use the positive semi-definiteness of the covariance matrix. Therefore, this term could be positive or negative and could cause the whole derivative to be large and positive. To control the worst-case scenario, we consider the upper bound for $|e^\top \Sigma_\phi^{-1} \dot{b}_t|$. Since $b_t = \Sigma_t w_t^*$, assuming negligible shift in the covariance matrix:

$$|e^\top \Sigma_\phi^{-1} \dot{b}_t| \leq \lambda_{\max}(\Sigma_\phi^{-1}) \lambda_{\max}(\Sigma_\phi) \|e\|_2 = \frac{\lambda_{\max}(\Sigma_\phi)}{\lambda_{\min}(\Sigma_\phi)} \|e\|_2 = \kappa(\Sigma_\phi) \|e\|_2 \tag{33}$$

where $\kappa(\Sigma_\phi)$ is the condition number of the covariance matrix. This term clearly shows that to minimize this upper bound, we need to set the condition number to the minimum value possible, which is one. This will be achieved if and only if the distribution is isotropic: $\lambda_{\max} = \lambda_2 = \cdots = \lambda_{\min} = \frac{c}{d}$.

### B.1.5. Among Isotropic Distributions, Gaussian Leads to Minimum Fluctuations in the Drift Term

To justify the choice of Gaussian as the distribution instead of other isotropic distributions, we focus on the second term in which the time derivative of $b_t = \mathbb{E}[\phi y_t]$ is involved. We show that if the distribution is non-Gaussian, some extra terms appear that increase the variance of the second term. As a result, the uncertainty of the sign of the second term increases, which is not desirable. Before showing this, we need to establish some properties of general and Gaussian random variables.

**If embedding vectors $\phi$ follow Gaussian distribution (proof of stein's lemma).** Let $\phi \sim \mathcal{N}(0, \Sigma_\phi)$ with density

$$p(\phi) = \frac{1}{Z} \exp\left(-\tfrac{1}{2} \phi^\top \Sigma_\phi^{-1} \phi\right) \tag{34}$$

Then

$$\nabla_\phi p(\phi) = -\Sigma_\phi^{-1} \phi \, p(\phi) \tag{35}$$

$$\boxed{\phi \, p(\phi) = -\Sigma_\phi \nabla_\phi p(\phi)} \tag{36}$$

For any smooth function $f : \mathbb{R}^d \to \mathbb{R}$,

$$\mathbb{E}[\phi f(\phi)] = \int_{\mathbb{R}^d} \phi f(\phi) \, p(\phi) \, d\phi \tag{37}$$

Substitute Eq. 36 in Eq. 37:

$$\mathbb{E}[\phi f(\phi)] = -\Sigma_\phi \int_{\mathbb{R}^d} f(\phi) \, \nabla_\phi p(\phi) \, d\phi \tag{38}$$

If we write the integral dimension-wise and apply integration by part:

$$\int f(\phi) \, \nabla_\phi p(\phi) \, d\phi = \sum_{j=1}^{d} \int f(\phi) \, \frac{\partial p(\phi)}{\partial \phi_j} \, d\phi \tag{39}$$

$$\int f(\phi) \, \frac{\partial p(\phi)}{\partial \phi_j} \, d\phi = \left[ f(\phi) \, p(\phi) \right]_{\phi_j=-\infty}^{\phi_j=\infty} - \int p(\phi) \, \frac{\partial f(\phi)}{\partial \phi_j} \, d\phi \tag{40}$$

As we have assumed multivariate Gaussian distribution, each dimension will have marginal uni variate distribution which decays exponentially fast. If we assume that $f(\phi)$ increases at most polynomially fast, we can conclude that:

$$\lim_{\|\phi\|_2 \to \infty} f(\phi) p(\phi) = 0 \tag{41}$$

so the boundary term vanishes:

$$\left[ f(\phi) \, p(\phi) \right]_{\phi_j=-\infty}^{\phi_j=\infty} \approx 0 \tag{42}$$

Substituting back into Eq. 38:

$$\mathbb{E}[\phi f(\phi)] = \Sigma_\phi \int_{\mathbb{R}^d} p(\phi) \, \nabla_\phi f(\phi) \, d\phi \tag{43}$$

Therefore,

$$\boxed{\mathbb{E}[\phi f(\phi)] = \Sigma_\phi \, \mathbb{E}[\nabla_\phi f(\phi)]} \tag{44}$$

**General form for an arbitrary distribution.** Let $p(\phi)$ be any smooth density and define residual $r$ as:

$$r(\phi) := \phi + \Sigma_\phi \nabla_\phi \log p(\phi) \tag{45}$$

This residual function is equal zero if and only if the distribution is Gaussian as in that case:

$$r(\phi) := \phi + \Sigma_\phi \Sigma_\phi^{-1} \phi = 0 \tag{46}$$

If we multiply both sides by $f(\phi)$ and take expectation:

$$\mathbb{E}[r(\phi) f(\phi)] = \mathbb{E}[\phi f(\phi)] + \Sigma_\phi \mathbb{E}[f(\phi) \nabla_\phi \log p(\phi)] \tag{47}$$

To simplify the second term, recall that:

$$\nabla_\phi \log p(\phi) = \frac{\nabla_\phi p(\phi)}{p(\phi)}, \tag{48}$$

so that

$$\mathbb{E}[f(\phi) \, \nabla_\phi \log p(\phi)] = \int f(\phi) \, \nabla_\phi p(\phi) \, d\phi. \tag{49}$$

By combining Eq. 38, 47, and 49 and reordering terms:

$$\boxed{\mathbb{E}[\phi f(\phi)] = \Sigma_\phi \, \mathbb{E}[\nabla_\phi f(\phi)] + \mathbb{E}[f(\phi) \, r(\phi)]} \tag{50}$$

If the distribution is Gaussian, the second term in Eq. 50 becomes zero and we recover the stein's lemma for multivariate Gaussian distribution.

**Effect on the second term in the derivative of Lyapanouv formula.** Recall

$$\boxed{\dot{\Gamma} = \underbrace{-4e^\top \Sigma_\phi e}_{\text{contraction}} \underbrace{-2e^\top \Sigma_\phi^{-1}\dot{b}_t}_{\text{target non-stationarity}} + \underbrace{2e^\top \Sigma_\phi^{-1}\dot{\Sigma}_\phi \Sigma_\phi^{-1} b_t}_{\text{representation drift}}} \tag{51}$$

and

$$b_t = \mathbb{E}[\phi\, y_t(\phi)]. \tag{52}$$

Our goal is to ensure $\dot{\Gamma}$ is always, or most of the time, negative. Since we are regularizing the embedding to have a fixed distribution, the third term will be negligible. The first term is also always negative and therefore always helpful. Here we show that a non-Gaussian distribution causes the variance of the second term to increase. As a result, the sign of this term may fluctuate frequently, causing this term to be positive and problematic.

For a general embedding distribution,

$$b_t = \Sigma_\phi \mathbb{E}[\nabla_\phi y_t(\phi)] + \mathbb{E}[y_t(\phi)\, r(\phi)] \tag{53}$$

Differentiating with respect to time:

$$\dot{b}_t = \Sigma_\phi \frac{d}{dt}\mathbb{E}[\nabla_\phi y_t(\phi)] + \frac{d}{dt}\mathbb{E}[y_t(\phi)\, r(\phi)] \tag{54}$$

Substitute into the target non-stationarity Lyapunov term:

$$-2e^\top \Sigma_\phi^{-1}\dot{b}_t = -2e^\top \frac{d}{dt}\mathbb{E}[\nabla_\phi y_t(\phi)] - 2e^\top \Sigma_\phi^{-1}\frac{d}{dt}\mathbb{E}[y_t(\phi)\, r(\phi)] \tag{55}$$

Since we consider a linear readout for the target Q-network, the first term in Eq. 55 depends only on the task and is independent of the embedding distribution due to taking the derivative with respect to $\phi$. In contrast, the second term depends on the distribution and contributes to variance when the residual term $r(\phi)$ is non-zero. In the Gaussian case, the residual term vanishes (Eq. 46), removing this contribution and reducing the total variance. Thus, the distributional dependence in Eq. 55 enters only through the residual term, which is zero if and only if the distribution is Gaussian. In the case of a non-Gaussian distribution, the residual term has a variance that makes the sign of the second term in the Lyapunov equation fluctuate and causes instability.

The variance of the residual term due to usage of a non-Gaussian distribution is proportional to:

$$\text{Var}\left(e^\top \Sigma_\phi^{-1}\frac{d}{dt}\mathbb{E}[y_t(\phi)\, r(\phi)]\right) \propto \mathbb{E}\left[\left(e^\top \Sigma_\phi^{-1}y_t(\phi)\, r(\phi)\right)^2\right] \tag{56}$$

Recall that:

$$r(\phi) := \phi + \Sigma_\phi \nabla_\phi \log p(\phi) \tag{57}$$

Residual term is a non-linear function of $\phi$ due to $\nabla_\phi \log p(\phi)$. If we write the Taylor expansion of residual vector around $\bar{r} = \mathbb{E}[r]$:

$$\text{Var}\left(e^\top \Sigma_\phi^{-1}\frac{d}{dt}\mathbb{E}[y_t(\phi)\, r(\phi)]\right) \propto \mathbb{E}\left[\left(e^\top \Sigma_\phi^{-1}y_t(\phi)\right)^2\left(J_r(\bar{r})(\phi-\bar{r}) + \frac{1}{2}H_r(\bar{r})[\phi-\bar{r},\phi-\bar{r}] + \cdots\right)^2\right] \tag{58}$$

From this formula, high-order moments of $\phi$ emerge, which result in a positive value for the variance of the residual term, which in turn increases the total variance of the second term of the derivative of the Lyapunov function. Therefore, a non-Gaussian distribution will lead to a high-variance drift term that may lead to instability. The minimum possible variance can be achieved by a Gaussian distribution. One point which should be noted is that although sub-Gaussian distributions share similar tail properties, Gaussianity provides additional structure beyond light tails. First, the Gaussian is the maximum-entropy distribution under a fixed variance constraint, encouraging efficient use of representational capacity. Second, for Gaussian distributions, all higher-order moments are determined by the covariance (via Isserlis's theorem (Isserlis, 1918)), so the covariance fully characterizes the distribution. In contrast, for general sub-Gaussian distributions, higher-order structure is not determined by the covariance and may influence behavior beyond second-order statistics. Finally, the Gaussian provides a canonical and tractable target for distribution matching within our framework.

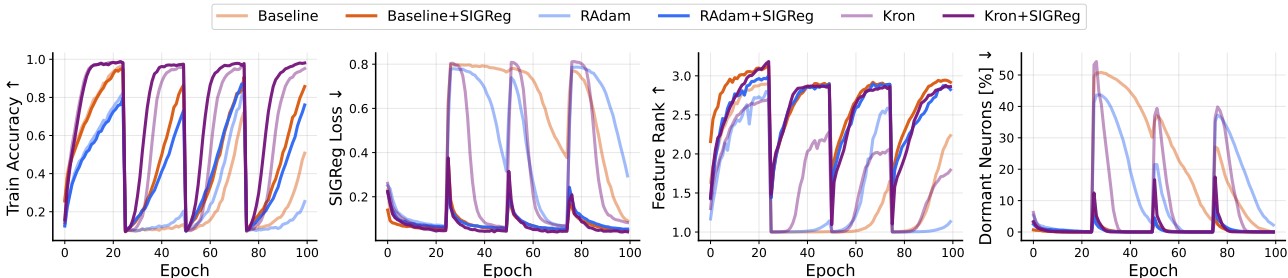

*Figure 8.* **Effect of non-stationarity and SIGReg loss minimization on representation stability and accelerating recovery in non-stationary CIFAR-10 experiment.** Non-stationarity induced by label shuffling causes a sharp drop in accuracy, increase in SIGReg loss, collapse of feature rank, and higher neuron dormancy. It can be observed that Kronecker-factored optimizer implicitly lowers the SIGReg loss compared to first-order methods. Explicitly minimizing SIGReg further improves all optimizers by accelerating recovery, preserving feature rank, and reducing neuron dormancy.

## C. Ablation Studies and Additional Results

### C.1. Non-Stationary CIFAR-10 Experiments with Different Optimizers

In this section, we extend the non-stationary CIFAR-10 experiments presented in subsection 4.1. We analyze the effect of different optimizers on training accuracy, feature rank, and neuron dormancy, and investigate how these quantities change after minimizing the SIGReg loss. The optimizers considered are:

- **Adam** (Kingma & Ba, 2017): a widely used first-order optimizer.

- **RAdam** (Liu et al., 2020): the default optimizer for PQN and used in some PPO implementations (Castanyer et al., 2026).

- **Kronecker-factored Optimizer (Kron)**(Martens & Grosse, 2015): a second-order optimizer shown to improve stability across various deep RL algorithms (Castanyer et al., 2026).

| Method | Train Accuracy(↑) | SIGReg Loss(↓) | Feature Rank (↑) | Dormant Neurons[%](↓) |
|---|---|---|---|---|
| Adam | 33.0 | 49.1 | 149.0 | 18.5 |
| Adam+SIGReg | 50.0 | 7.6 | 262.9 | 0.4 |
| RAdam | 29.8 | 42.3 | 148.8 | 12.6 |
| RAdam+SIGReg | 42.8 | 8.4 | 251.8 | 0.6 |
| Kron | 62.6 | 28.5 | 167.4 | 7.9 |
| Kron+SIGReg | 72.7 | 7.2 | 252.2 | 0.9 |

*Table 1.* **Non-stationary CIFAR-10.** Area Under the Curve (AUC) for different methods across evaluation metrics. SIGReg loss minimization improve results across all metrics and optimizers.

Figure 8 illustrates the effect of label shuffling on various metrics. When labels are shuffled, we observe: (1) significant drop in accuracy with poor recovery, (2) sharp increases in the SIGReg loss indicating loss of isotropic Gaussian structure, (3) collapse in feature rank, and (4) increased neuron dormancy. Notably, Kronecker-factored optimization implicitly reduces the SIGReg loss compared to first-order methods. However, explicitly minimizing the SIGReg objective further improves all optimizers by accelerating recovery, maintaining feature rank, and reducing dormancy.

Table 1 quantifies these observations through Area Under Curve measurements. Across all optimizers, adding SIGReg regularization leads to substantial improvements: training accuracy increases by 17–51%, SIGReg loss decreases by 78–85%, feature rank improves by 51–76%, and neuron dormancy drops by 89–98%. These results confirm that maintaining isotropic Gaussian structure is crucial for plasticity and recovery under non-stationary conditions.

### C.2. Effect of Different Target Distributions

In this section, we expand the results discussed in subsection 4.3. Results of this section are based on Atari-10 benchmark (Aitchison et al., 2023). Table 2 summarizes the impact of different representation regularization strategies on performance

| | PQN | | PPO | |
|---|---|---|---|---|
| Method | % of Games Improved (↑) | Avg. Improvement (↑) | % of Games (↑) | Avg. Improvement (↑) |
| Gaussian (Real and Imaginary parts) | 90% | 305% | 70% | 7.9% |
| Laplacian | 90% | 234% | 60% | 3.2% |
| Logistic | 90% | 209% | 50% | -0.8% |
| Real Part Only | 100% | 151% | 60% | 2.4% |
| Imaginary Part Only | 70% | 56% | 50% | 1.9% |
| *Covariance Whitening* | 90% | 144% | 70% | 4.4% |

*Table 2.* **Ablation study on Atari-10.** Percentage of games improved over baseline and average AUC improvement across all games. **Top:** different isotropic target distributions (Laplacian and Logistic). **Middle:** minimizing different SIGReg loss components. **Bottom:** whitening the covariance matrix without controlling tails. Gaussian with real and imaginary components performs best, while enforcing isotropy alone can improve some environments. In PQN, minimizing only the real part improves more games than the baseline Isotropic Gaussian, though with significantly smaller average improvement.

across Atari-10 games. We report both the fraction of games in which a method improves over the baseline and the average AUC improvement.

### C.2.1. ALTERNATIVE ISOTROPIC DISTRIBUTIONS

The goal of this subsection is to empirically evaluate whether Gaussianity of the target embedding distribution is an important factor for performance. To test this, we compare the Gaussian target to two alternative isotropic distributions with heavier tails, shown in the top section of Table 2. These distributions were chosen because, according to our formal analysis, heavier tails can increase instability. Across both PQN and PPO, the Gaussian target consistently outperforms the alternatives, both in terms of the number of games showing improvement and the average AUC increase across all 10 games. This indicates that matching the Gaussian distribution's characteristics, including tail behavior, contributes positively to performance.

To further isolate the effect of Gaussianity, we also evaluate a simple covariance whitening method, reported in the bottom section of Table 2. Whitening enforces isotropy by setting the covariance matrix to the identity, but it does not control tail characteristics. While this method achieves some improvements, particularly in PPO, it underperforms compared to the full Gaussian target in PQN and is generally less consistent. These results suggest that isotropy alone provides partial benefits, while Gaussianity, including its tail properties, plays a key role in stabilizing representations and achieving optimal performance.

### C.2.2. ROLE OF SYMMETRY AND TAIL DECAY

In this subsection, we analyze the effect of symmetry and tail characteristics of the representation separately. The SIGReg loss is based on projecting high-dimensional embedding points onto one-dimensional directions and matching the estimated characteristic function of the projections to a target distribution. Since the characteristic function is the Fourier transform of the probability distribution, the target or projected embeddings can be complex numbers. The imaginary part captures all odd-order moments of the distribution, so if it is zero, the distribution is symmetric. The real part correlates with the even-order moments, which correspond to the tail behavior.

For a Gaussian target, which is symmetric, the characteristic function is real. However, the estimated characteristic function of the embeddings may have a nonzero imaginary component. The middle section of Table 2 shows the effect of minimizing each part separately. The results indicate that minimizing the real component is significantly more important than minimizing the imaginary component. In PQN, focusing on the real part even increases the number of games improved compared to minimizing both components simultaneously. The underlying reason is not fully understood, but it may result from an implicit reduction of the imaginary component when minimizing the real part.

### C.3. More Results on Implicit Isotropy in Stabilization Methods

In this section, we expand on the discussion in subsection 4.4 by investigating the connection between isotropic Gaussian representations and the stabilization mechanisms introduced by Castanyer et al. (2026), namely Kronecker-factored optimization and multi-skip residual architectures, both designed to improve stability in deep RL at scale.

Fig. 9 presents results for Parallelized Q-Networks (PQN) across the Atari-10 benchmark. Across nearly all games,

Kronecker-factored optimization produces representations that are substantially closer to an isotropic Gaussian distribution, as indicated by lower SIGReg loss, compared to the baseline. Importantly, this effect emerges **without any explicit representation regularization**, suggesting that Kronecker-factored optimization implicitly encourages embeddings to adopt a geometry that is both Gaussian-like and approximately isotropic. Similarly, multi-skip residual architectures, originally designed to stabilize gradient propagation, also implicitly promote isotropic Gaussian embeddings, as shown in Fig. 10. These effects are not limited to PQN: analogous trends are observed in Proximal Policy Optimization (PPO), as illustrated in Fig. 11 and Fig. 12. Across both algorithms, the figures reveal a strong correlation between an increase in feature rank, a decrease in the percentage of dormant neurons, and an implicit reduction in SIGReg loss, highlighting that these stabilization mechanisms partially act by shaping the geometry of the learned representations. Another interesting observation from Figures 9 to 12 is that, across all games, PPO exhibits a lower SIGReg loss compared to PQN. Understanding the underlying reason for this difference remains an interesting research question.

Although Kronecker-factored optimization is effective and provides substantial improvements, it introduces additional memory and computational overhead due to the estimation of gradient curvature. This raises the question of whether similar benefits could be achieved using first-order optimizers if the geometry of the representations is explicitly controlled. Based on the patterns observed in Fig. 9, we hypothesize that a significant portion of the stability and performance gains from these methods arises from the implicit shaping of representation geometry. To test this hypothesis, we explicitly enforce isotropic Gaussian embeddings in baseline models using the auxiliary SIGReg objective. Fig. 14 and Table 3 show that this approach significantly narrows the performance gap relative to models trained with Kronecker-factored optimization, across both PQN and PPO. Notably, this improvement is achieved without introducing additional memory or computational burden compared to the baseline, highlighting the importance of embedding geometry rather than optimizer complexity.

Fig. 14 presents IQM human-normalized results for different methods. The upper panel shows PQN results, where explicit SIGReg loss minimization reduces the gap between the second-order Kronecker-factored optimizer and the first-order RAdam baseline. Adding multi-skip residual connections achieves nearly the same performance while being significantly faster than Kronecker-factored optimization (see the steps per second (SPS) column in Table 3). The bottom panel shows PPO results, indicating even more promising trends: SIGReg loss minimization on top of a first-order optimizer, with or without multi-skip connections, outperforms the second-order optimizer while remaining faster. Examining Table 3, we see that explicit SIGReg loss minimization successfully reduces the gap between Kronecker-factored optimization and the baseline, and can even surpass it, while being around ten times faster in PQN and three times faster in PPO, without requiring a memory bank for gradient curvature estimation.

Together, these observations suggest that representation isotropy and Gaussianity are key factors underlying the empirical stability gains seen with advanced stabilization mechanisms. Explicitly regularizing embeddings to be isotropic and Gaussian can capture much of the benefit of second-order optimizers while retaining the efficiency of first-order methods.

## C.4. Comprehensive Atari Benchmark Results

We evaluate our method on the Atari benchmark across 57 games using both PQN and PPO algorithms. This comprehensive evaluation allows us to assess the generalization of our approach across diverse environments and training paradigms. The second row of Table 3 summarizes the performance of SIGReg loss minimization across the full Atari suite. For PQN, applying SIGReg loss minimization improves performance in approximately 90% of games, with an average AUC improvement of 889%. For PPO, 68% of games show improvement, with an average AUC gain of 25%. This large difference between PQN and PPO can be attributed to two factors: (1) PPO exhibits a lower SIGReg loss compared to PQN in the baseline (see baseline results in Fig. 9 and Fig. 11), and (2) PPO already achieves strong performance, leaving limited room for further improvement. For per-game improvements and IQM human-normalized results, see Fig. 6.

In the following sections, we first demonstrate the effect of SIGReg loss minimization on several evaluation metrics that are useful for assessing the stability of deep RL methods. We then present per-game episode reward plots to illustrate the learning dynamics across all games.

### C.4.1. IMPACT ON REPRESENTATION QUALITY

Figure 13 extends Fig. 4 to all games in the Atari-10 benchmark and considers two regularization factors, 10 and 0.2. It shows how reward, SIGReg loss, feature rank, and the percentage of dormant neurons change over time, with and without SIGReg regularization. The orange line corresponds to the baseline without SIGReg, while the other lines show different regularization strengths, where a larger $\lambda$ means stronger regularization.

| | PQN | | | PPO | | |
|---|---|---|---|---|---|---|
| Method | % of Games Improved (↑) | Avg. Improvement (↑) | Avg. SPS (↑) | % of Games Improved (↑) | Avg. Improvement (↑) | Avg. SPS (↑) |
| Kron | 98.2% | 1404% | 1.6k | 45.6% | -2.6% | 1.8k |
| Baseline + SIGReg | 89.5% | 889% | 9.3k | 68.4% | 25.5% | 2.7k |
| Baseline + MultiSkip + SIGReg | 92.9% | 1695% | 11.4k | 87.7% | 88.6% | 3.2k |

*Table 3.* **PQN and PPO Performance Across Full Atari Suite.** We report the percentage of games in which each method improves over the baseline, the average percentage of AUC improvement across all games, and the average steps per second (Avg. SPS). In terms of average AUC improvement, the baseline augmented with multi-skip residual connections and SIGReg loss minimization outperforms all other methods, while also achieving higher SPS and avoiding the memory overhead associated with computing gradient curvature. In terms of the percentage of games improved, Kronecker-factored optimization achieves the best result by a small margin in PQN, and it is outperformed by both two other cases in PPO. Overall, these results indicate that the performance gains provided by Kron do not justify the additional memory and computational cost.

Across all games and settings, SIGReg loss minimization consistently reduces neuron dormancy, maintains a higher feature rank, and improves reward. The increase in rank is expected, since enforcing isotropy spreads variance more evenly across all principal components. However, the strong reduction in the percentage of dormant neurons is less straightforward and remains an interesting direction for future research.

### C.4.2. EPISODE REWARD FOR ALL INDIVIDUAL GAMES IN FULL ATARI SUITE

Figures 15 and 16 show the episode reward over time for individual games in PQN and PPO, respectively. Each plot includes three lines corresponding to different regularization factors. The line with $\lambda = 0$ represents the baseline method without SIGReg loss minimization. Improvements from SIGReg loss minimization are more pronounced in PQN. One possible reason is that, in PPO, the baseline SIGReg loss is already lower than in PQN, leaving less room for improvement. Additionally, the baseline performance in PPO is significantly higher than in PQN, further limiting the potential for noticeable gains.

Besides the difference in performance gains achieved by PQN compared to PPO, the magnitude of improvement also varies across different games. To further illustrate this variance, we highlight Freeway as a representative case where SigReg degrades performance for both PQN and PPO. Freeway has sparse and weak reward signals, and the additional regularization may introduce frequent but uninformative optimization signals that interfere with learning. The task also appears to require limited representational complexity, as reflected by its low effective rank. Although SigReg increases effective rank and reduces dormancy, indicating improved representation utilization, these changes do not translate into better performance, and in PQN lead to complete collapse. This gap suggests that the inductive bias introduced by SigReg may be misaligned with the task, and its impact on optimization dynamics can outweigh representational benefits. A similar pattern is observed in Pitfall and Private Eye, which have sparse rewards, as well as Robotank and Bowling, which exhibit low visual and state complexity. In these environments, SigReg also leads to degraded PQN performance, suggesting that the regularization may be misaligned with learning dynamics when reward signals are weak or the representational demands are limited. Uncovering the underlying causes for this variance in performance remains an open problem.

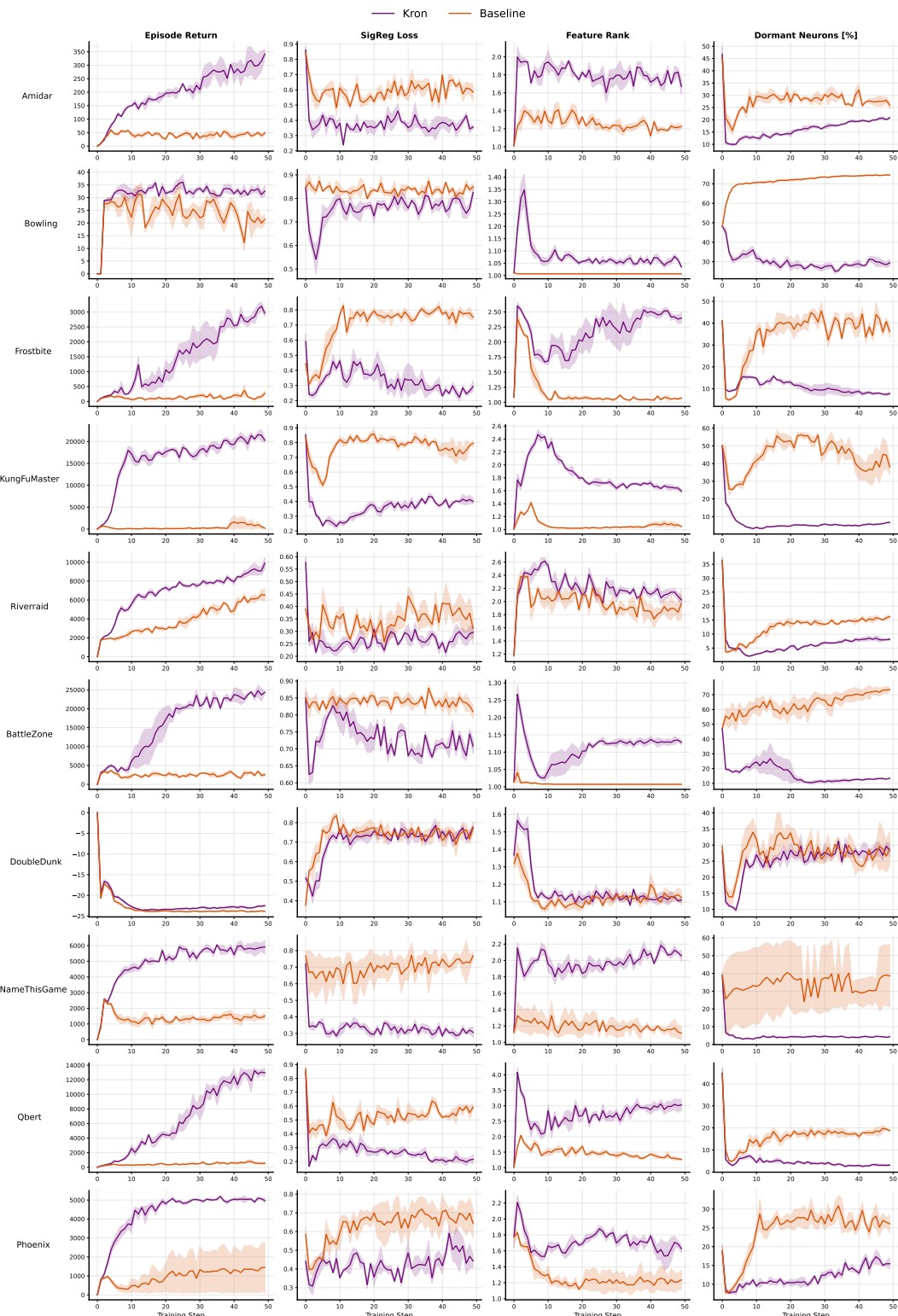

*Figure 9.* **Atari-10, PQN, Kron Optimizer.** The change of reward, SIGReg loss, feature rank, and percentage of dormant neurons. In almost all games, Kron leads to implicit minimization of SIGReg loss, higher rank, and lower dormancy.

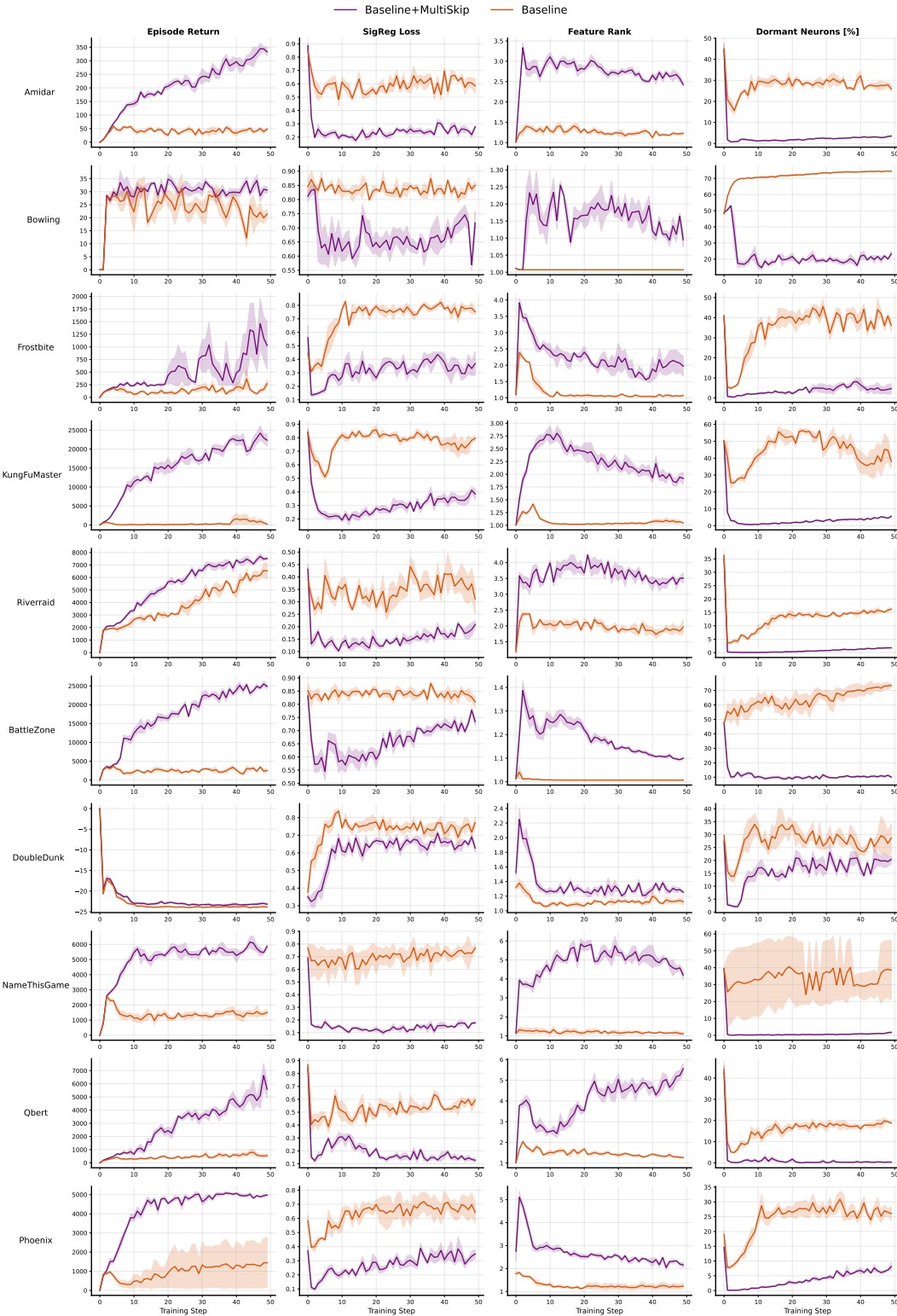

*Figure 10.* **Atari-10, PQN, Multi-skip Residual Architecture.** The change of reward, SIGReg loss, feature rank, and percentage of dormant neurons. In almost all games, using Multi-skip Residual Architecture leads to implicit minimization of SIGReg loss, higher rank, and lower dormancy.

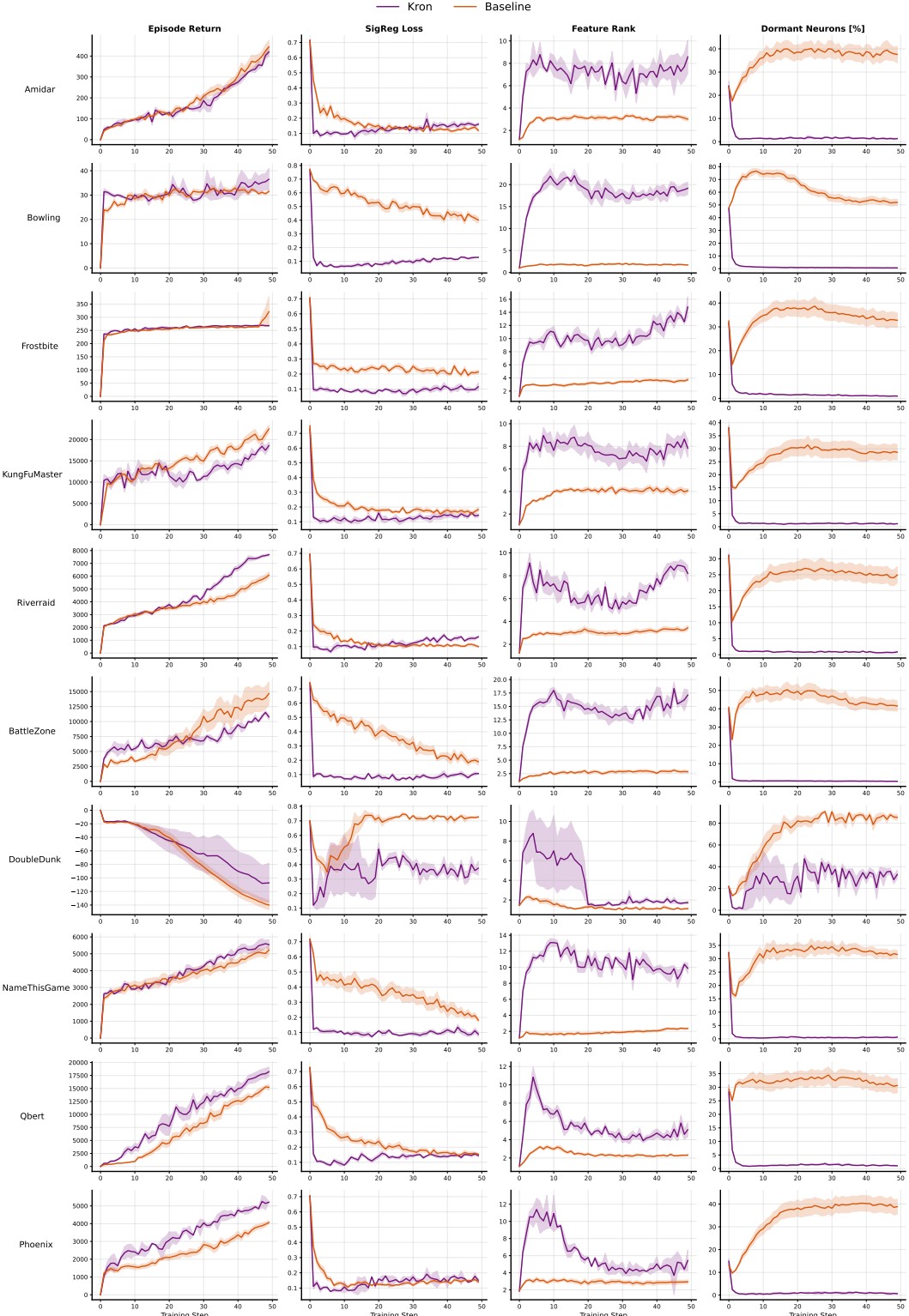

*Figure 11.* **Atari-10, PPO, Kron Optimizer.** The change of reward, SIGReg loss, feature rank, and percentage of dormant neurons. In almost all games, Kron leads to implicit minimization of SIGReg loss, higher rank, and lower dormancy.

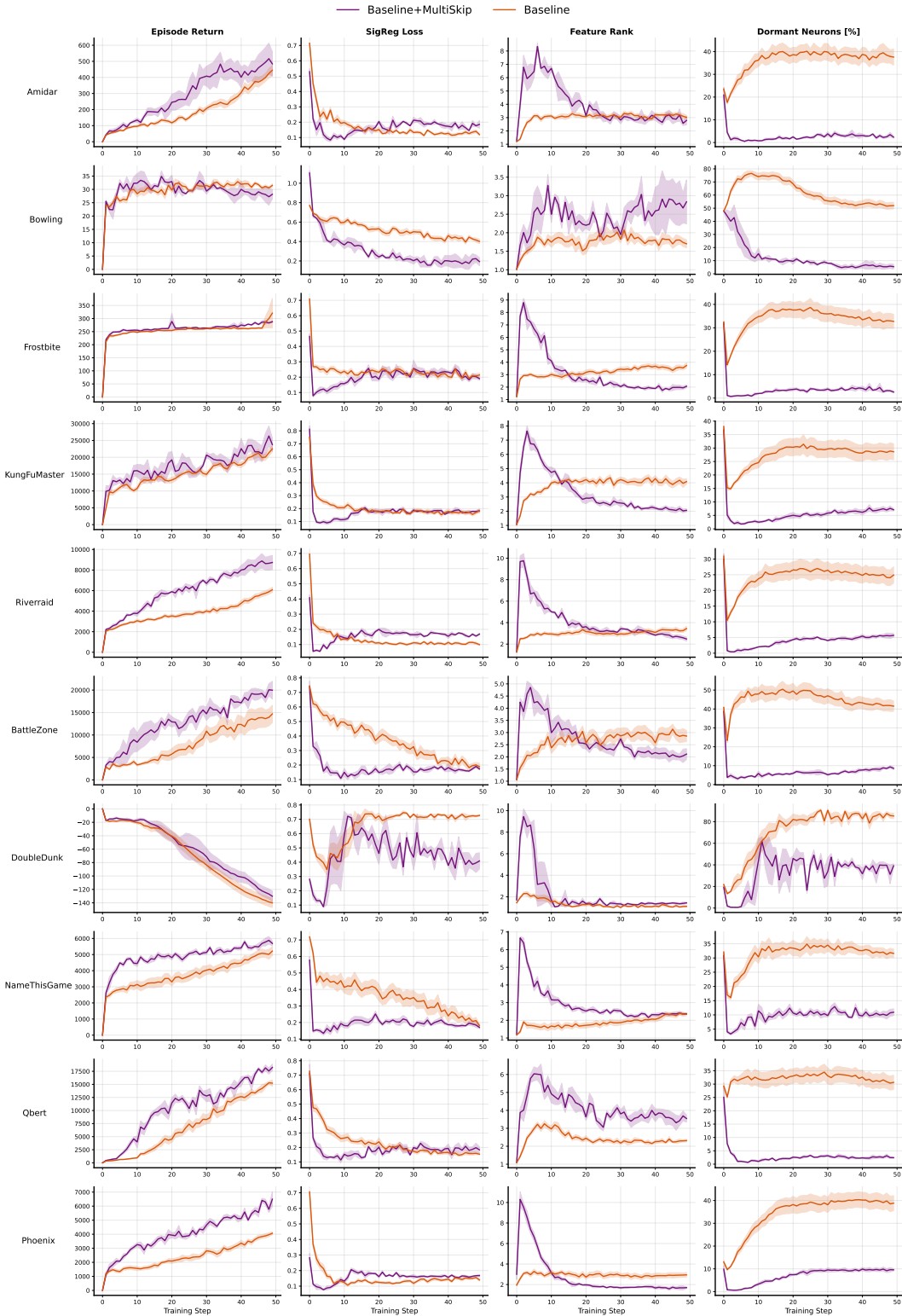

*Figure 12.* **Atari-10, PPO, Multi-skip Residual Architecture.** The change of reward, SIGReg loss, feature rank, and percentage of dormant neurons. In almost all games, using Multi-skip Residual Architecture leads to implicit minimization of SIGReg loss, higher rank, and lower dormancy.

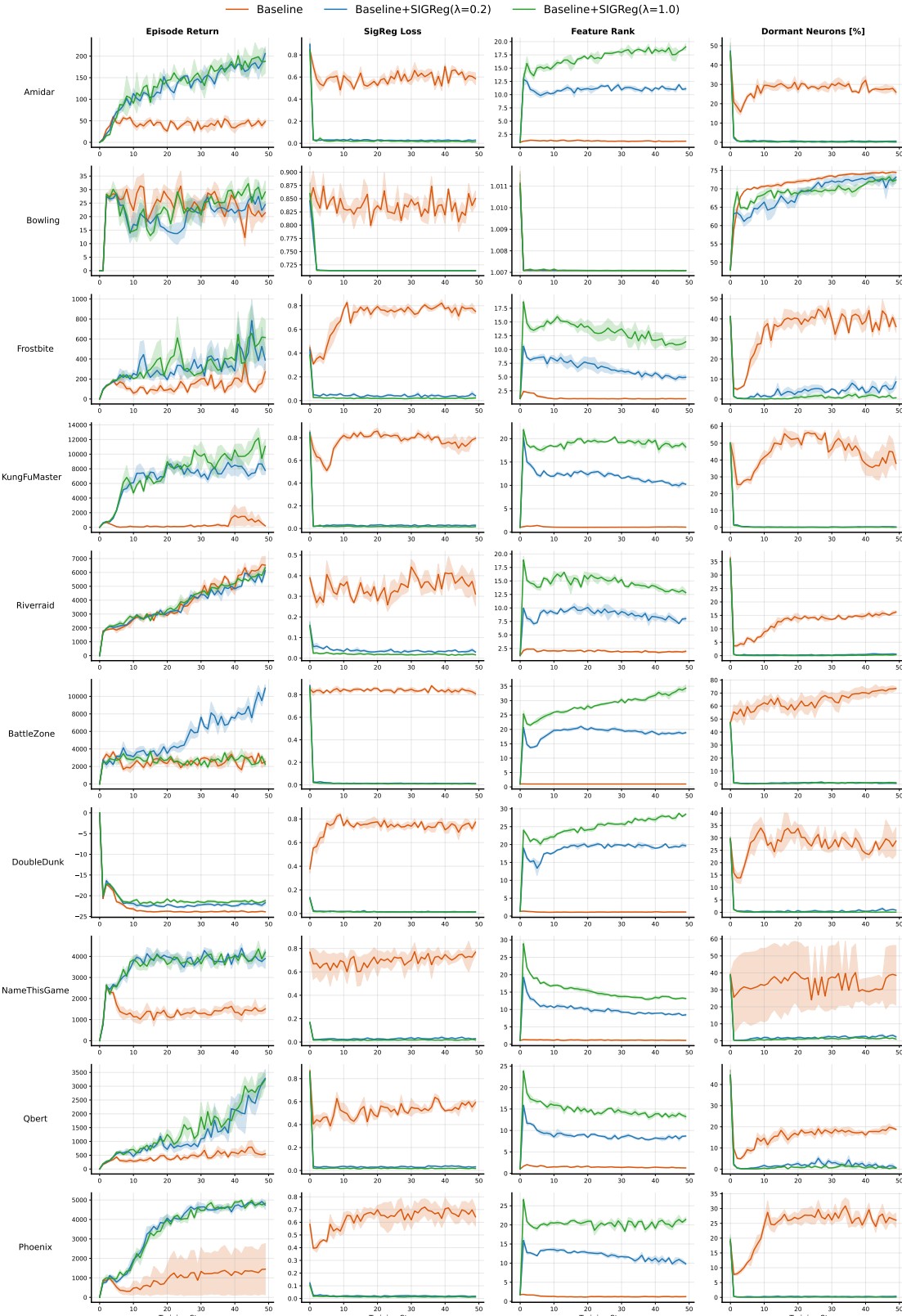

*Figure 13.* **Atari-10, PQN with Explicit SIGReg Loss Minimization.** The change of reward, SIGReg loss, feature rank, and percentage of dormant neurons through time. The orange line is the baseline without SIGReg loss minimization, and the other lines are SIGReg loss minimization with different strengths (Larger $\lambda$ means stronger regularization).

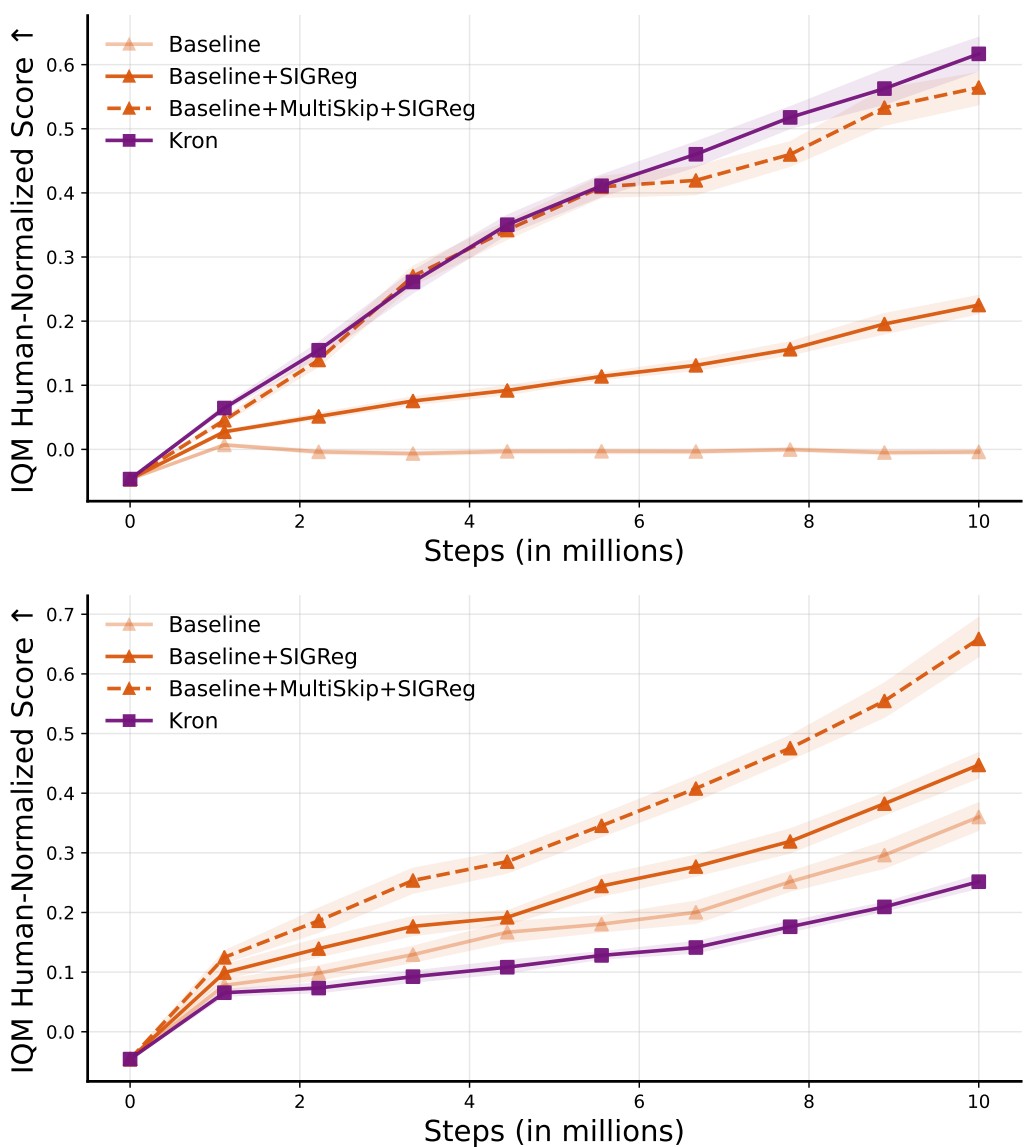

*Figure 14.* **IQM Human-Normalized Results for PQN (top) and PPO (bottom) Across the Atari-10 Benchmark Games.** Explicit SIGReg loss minimization encourages isotropic Gaussian embeddings, reducing the performance gap between first-order optimizers (RAdam) and second-order Kronecker-factored optimization. Adding multi-skip residual connections further improves performance.

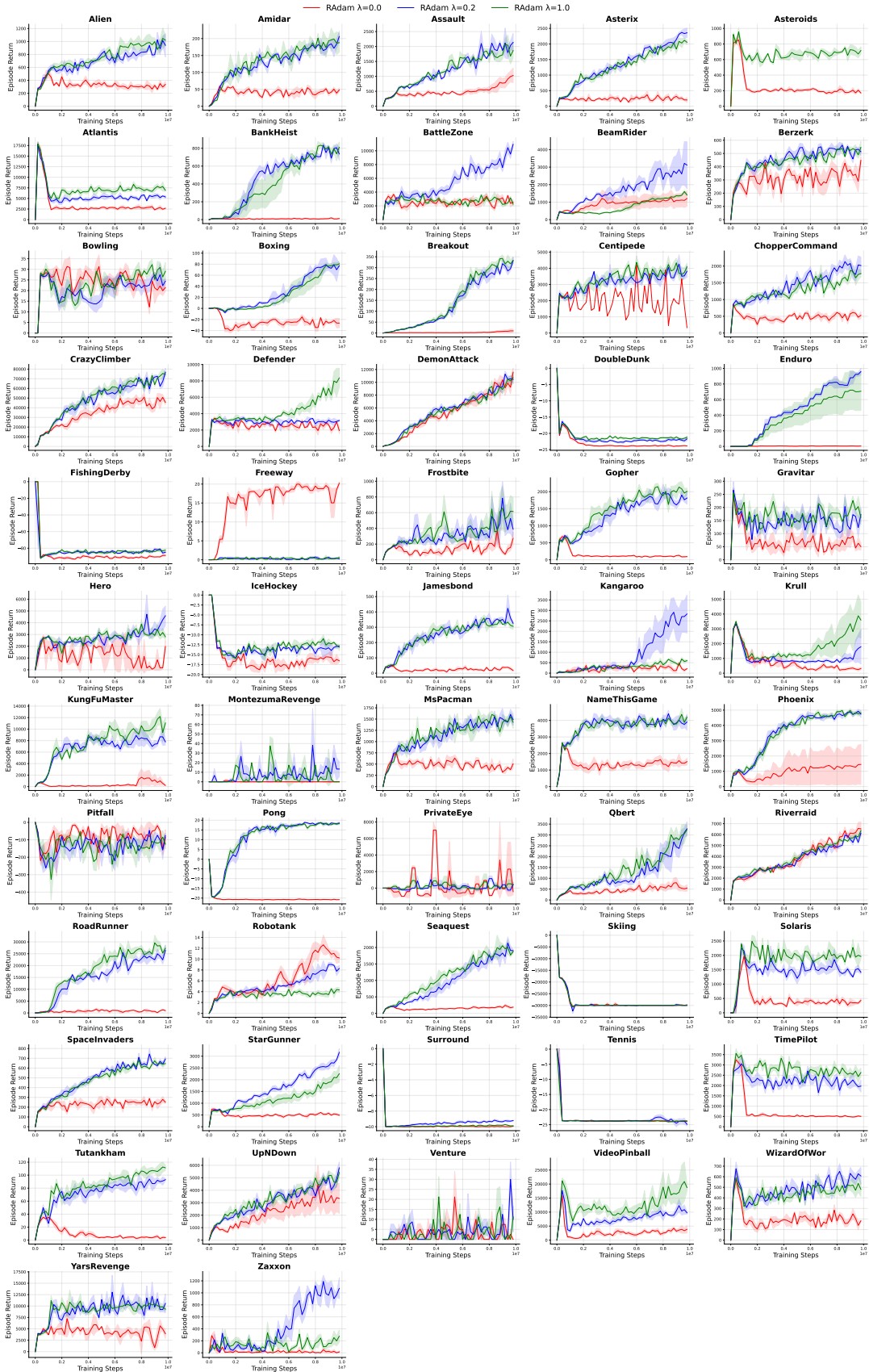

*Figure 15.* **PQN.** Reward over time for individual games with and without SIGReg loss minimization.

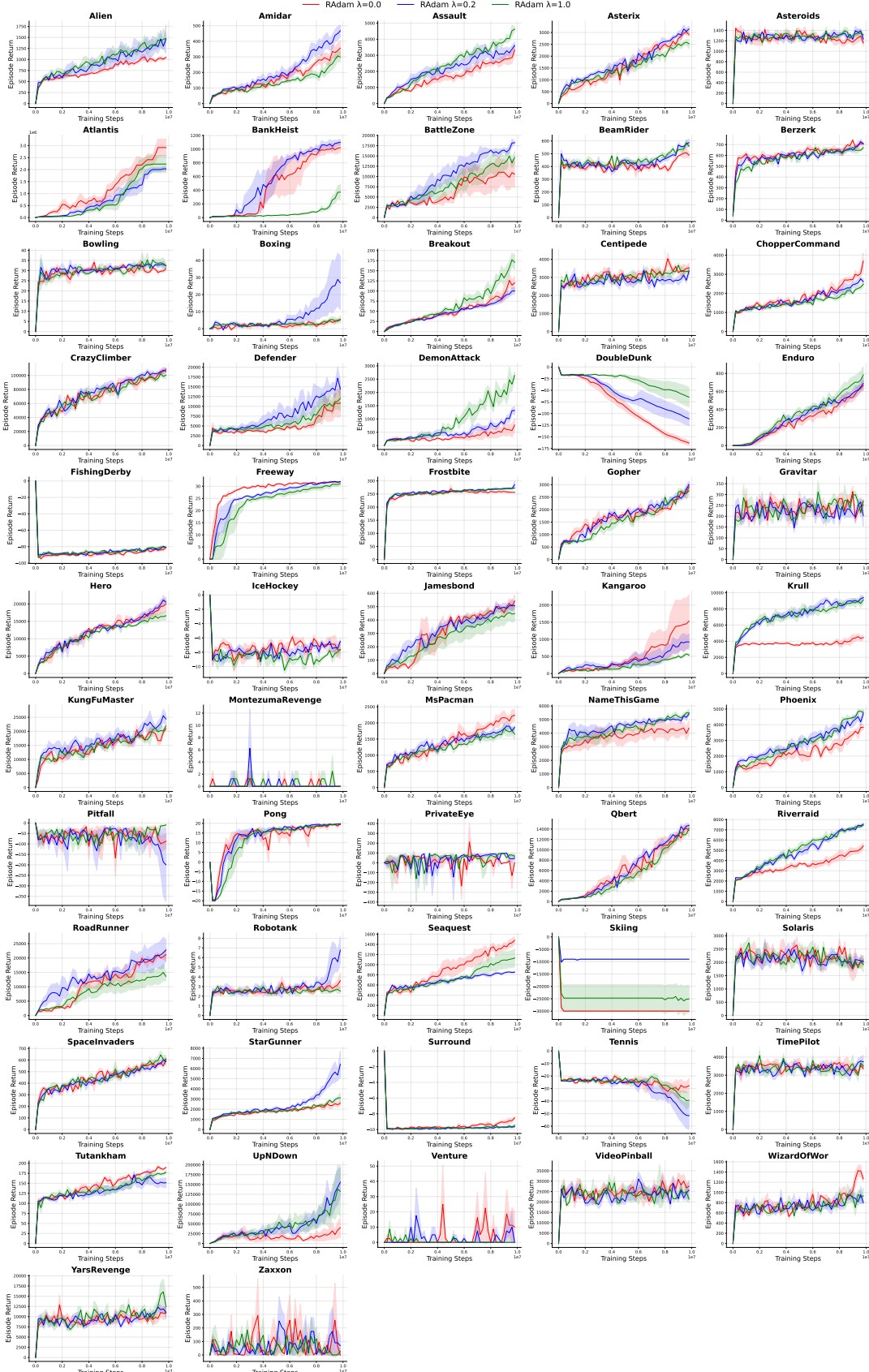

*Figure 16.* **PPO.** Reward over time for individual games with and without SIGReg loss minimization.

# D. Isaac Gym

Across Isaac Gym continuous-control tasks (Makoviychuk et al., 2021), enforcing isotropic Gaussian representations consistently improves learning dynamics and final performance. Compared to PPO, PPO + SIGReg achieves faster early learning, more stable training trajectories, and higher asymptotic returns across environments.

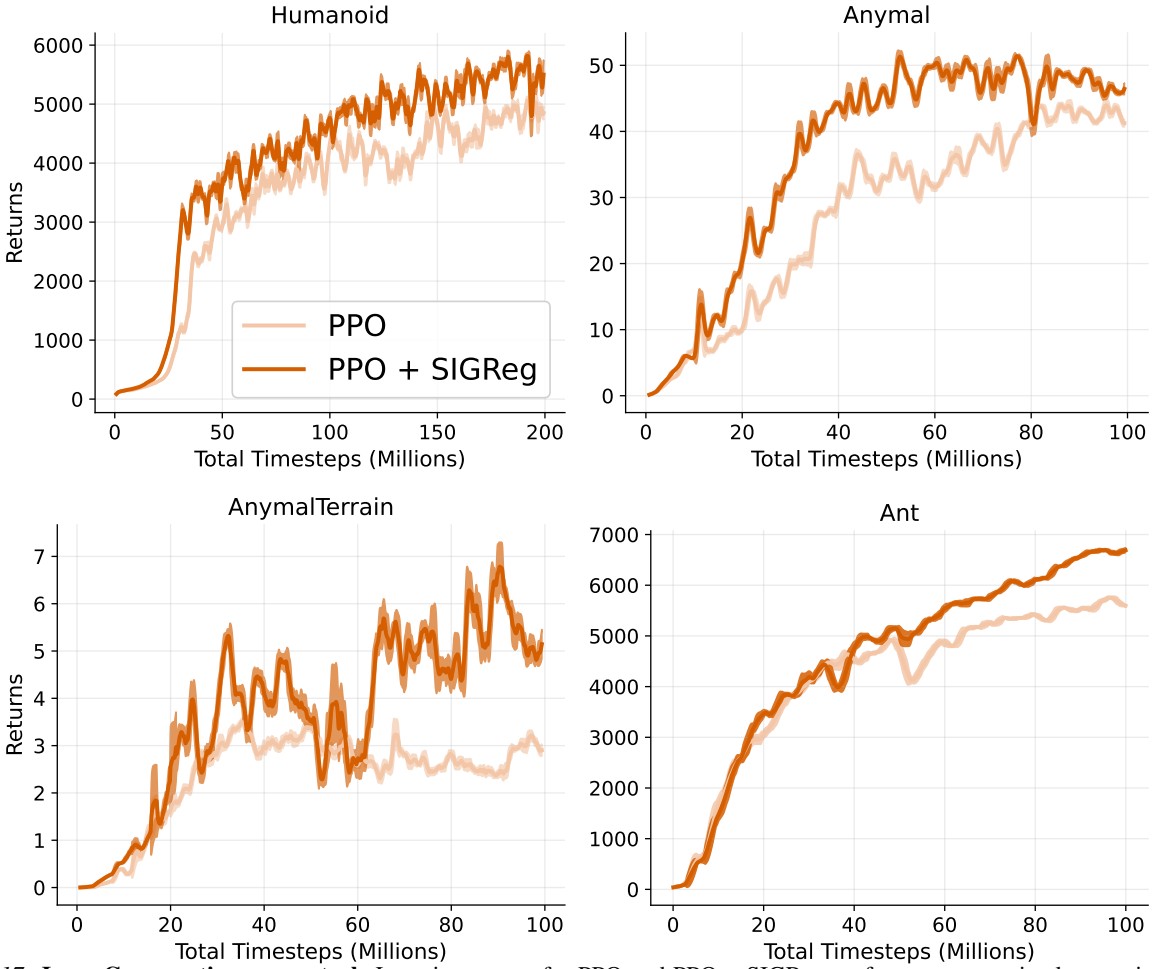

*Figure 17.* **Isaac Gym continuous control.** Learning curves for PPO and PPO + SIGReg on four representative locomotion tasks. Encouraging isotropic Gaussian representations improves learning stability and asymptotic performance across all environments. Curves show mean episode returns over five independent runs, with shaded regions indicating variability across runs.

# E. Hyperparameters

This section summarizes the hyperparameters used across all experiments and algorithms. Unless stated otherwise, we follow the configurations proposed in the corresponding original works and adopt the same settings used in Castanyer et al. (2026) for overlapping baselines.

For consistency and computational practicality, we use a single fixed set of hyperparameters for each baseline across all environments and experimental conditions. This choice isolates the effect of the proposed methods from confounding factors introduced by per-task tuning and ensures a fair comparison across algorithms. We note, however, that deep RL methods can be sensitive to hyperparameter choices (Ceron et al., 2024a). While performing an extensive hyperparameter search for each setting could potentially yield stronger absolute performance, such a procedure is computationally prohibitive at the scale considered in this work. Importantly, our conclusions focus on relative performance trends and representation behavior, which we found to be stable under reasonable variations of the default hyperparameters.

*Table 4.* PQN Hyperparameters

| Hyperparameter | Value / Description |
|---|---|
| Learning rate | 2.5e-4 |
| Anneal lr | False (no learning rate annealing) |
| Num envs | 128 (parallel environments) |
| Num steps | 32 (steps per rollout per environment) |
| Gamma | 0.99 (discount factor) |
| Num minibatches | 32 |
| Update epochs | 2 (policy update epochs) |
| Max grad norm | 10.0 (gradient clipping) |
| Start e | 1.0 (initial exploration rate) |
| End e | 0.005 (final exploration rate) |
| Exploration fraction | 0.10 (exploration annealing fraction) |
| Q lambda | 0.65 ($Q(\lambda)$ parameter) |
| Use ln | True (use layer normalization) |
| Activation fn | relu (activation function) |

*Table 5.* PPO Hyperparameters

| Hyperparameter | Value / Description |
|---|---|
| Learning rate | 2.5e-4 |
| Num envs | 8 |
| Num steps | 128 (steps per rollout per environment) |
| Anneal lr | True (learning rate annealing enabled) |
| Gamma | 0.99 (discount factor) |
| Gae lambda | 0.95 (GAE parameter) |
| Num minibatches | 4 |
| Update epochs | 4 |
| Norm adv | True (normalize advantages) |
| Clip coef | 0.1 (PPO clipping coefficient) |
| Clip vloss | True (clip value loss) |
| Ent coef | 0.01 (entropy regularization coefficient) |
| Vf coef | 0.5 (value function loss coefficient) |
| Max grad norm | 0.5 (gradient clipping threshold) |
| Use ln | False (no layer normalization) |
| Activation fn | relu (activation function) |
| Shared cnn | True (shared CNN between policy and value networks) |

*Table 6.* PPO Hyperparameters for IsaacGym

| Hyperparameter | Value / Description |
|---|---|
| Total timesteps | 30,000,000 |
| Learning rate | 0.0026 |
| Num envs | 4096 (parallel environments) |
| Num steps | 16 (steps per rollout) |
| Anneal lr | False (disable learning rate annealing) |
| Gamma | 0.99 (discount factor) |
| Gae lambda | 0.95 (GAE lambda) |
| Num minibatches | 2 |
| Update epochs | 4 (update epochs per PPO iteration) |
| Norm adv | True (normalize advantages) |
| Clip coef | 0.2 (policy clipping coefficient) |
| Clip vloss | False (disable value function clipping) |
| Ent coef | 0.0 (entropy coefficient) |
| Vf coef | 2.0 (value function loss coefficient) |
| Max grad norm | 1.0 (max gradient norm) |
| Use ln | False (no layer normalization) |
| Activation fn | relu (activation function) |

*Table 7.* Image Classification Hyperparameters (CIFAR-10)

| Hyperparameter | Value |
|---|---|
| Batch size | 256 |
| Epochs | 100 |
| Learning rate | 0.00025 |

*Table 8.* SIGReg Loss Hyperparameters.

| Hyperparameter | Value |
|---|---|
| Number of Random Projections | 16 |
| Number of Frequency Samples | 8 |
| Maximum Frequency | 5.0 |
| Regularization Factor (larger → stronger) | Best value between [1.0 and 0.2] (see Table 9) |

*Table 9.* Best regularization factor ($\lambda$) selected via AUC for PQN and PPO algorithms across Atari games (Larger means stronger regularization)

| Game | PQN | PPO | Game | PQN | PPO |
|---|---|---|---|---|---|
| Alien | 1.0 | 1.0 | Kangaroo | 0.2 | 0.2 |
| Amidar | 1.0 | 0.2 | Krull | 1.0 | 0.2 |
| Assault | 1.0 | 1.0 | KungFuMaster | 1.0 | 0.2 |
| Asterix | 0.2 | 0.2 | MontezumaRevenge | 0.2 | 1.0 |
| Asteroids | 1.0 | 1.0 | MsPacman | 1.0 | 0.2 |
| Atlantis | 1.0 | 1.0 | NameThisGame | 0.2 | 1.0 |
| BankHeist | 0.2 | 0.2 | Phoenix | 0.2 | 0.2 |
| BattleZone | 0.2 | 1.0 | Pitfall | 0.2 | 1.0 |
| BeamRider | 0.2 | 1.0 | Pong | 0.2 | 0.2 |
| Berzerk | 0.2 | 0.2 | PrivateEye | 1.0 | 1.0 |
| Bowling | 1.0 | 0.2 | Qbert | 1.0 | 0.2 |
| Boxing | 0.2 | 0.2 | Riverraid | 1.0 | 0.2 |
| Breakout | 1.0 | 1.0 | RoadRunner | 1.0 | 0.2 |
| Centipede | 1.0 | 1.0 | Robotank | 0.2 | 0.2 |
| ChopperCommand | 0.2 | 0.2 | Seaquest | 1.0 | 1.0 |
| CrazyClimber | 1.0 | 0.2 | Skiing | 0.2 | 0.2 |
| Defender | 1.0 | 0.2 | Solaris | 1.0 | 1.0 |
| DemonAttack | 0.2 | 1.0 | SpaceInvaders | 0.2 | 1.0 |
| DoubleDunk | 1.0 | 1.0 | StarGunner | 0.2 | 0.2 |
| Enduro | 0.2 | 1.0 | Surround | 0.2 | 1.0 |
| FishingDerby | 1.0 | 1.0 | Tennis | 0.2 | 1.0 |
| Freeway | 1.0 | 0.2 | TimePilot | 1.0 | 1.0 |
| Frostbite | 1.0 | 0.2 | Tutankham | 1.0 | 1.0 |
| Gopher | 1.0 | 0.2 | UpNDown | 1.0 | 1.0 |
| Gravitar | 1.0 | 1.0 | Venture | 0.2 | 0.2 |
| Hero | 1.0 | 0.2 | VideoPinball | 1.0 | 1.0 |
| IceHockey | 1.0 | 0.2 | WizardOfWor | 0.2 | 0.2 |
| Jamesbond | 0.2 | 0.2 | YarsRevenge | 1.0 | 1.0 |
| | | | Zaxxon | 0.2 | 0.2 |

# F. Additional Toy Experiments for Empirical Justification

We conduct additional toy experiments to further justify the choice of isotropic Gaussian as the embedding distribution. Specifically, we consider a linear regression model trained with SGD to track a time-varying target optimal weight $w_t^*$, consistent with the linear readout assumption used in our analysis. By varying the distribution of the input data used to train the linear regression model ($\phi$), we examine the effect of the data distribution on optimization dynamics and stability. Data samples are independently and identically distributed according to either a Gaussian or Laplace distribution, with covariance matrices whose condition number ($\kappa$) vary. A condition number of 1 corresponds to the isotropic case, while larger condition numbers correspond to increasingly anisotropic distributions.

## F.1. The Effect of Isotropy and the Type of Distribution

We first study how isotropy and the type of distribution affect tracking stability in a controlled linear regression setting trained with SGD. The linear model is being trained in a setting with a time-varying optimal weight vector $w_t^*$ under different feature distributions $\phi$, allowing us to isolate the role of geometry and tail behavior in the optimization dynamics. We compare four families of distributions: Isotropic Gaussian, Anisotropic Gaussian with varying condition numbers, Isotropic Laplace, and Anisotropic Laplace. The setup is similar to the non-stationary CIFAR-10 experiments discussed in Section 4.1, where the underlying target is periodically changed at fixed intervals. In this simplified setting, we directly evaluate which distributional regimes enable smoother contraction of the tracking error norm $\|w_t - w_t^*\|_2$ following each task switch, and how this behavior depends on isotropy, conditioning, and tail behavior.

**Gaussian Distribution.** Figures 18–20 summarize results for Gaussian features. The isotropic Gaussian case (Fig. 18) serves as a baseline and exhibits stable dynamics: after each task switch, the tracking error increases sharply due to the change in $w_t^*$ and then decreases rapidly and monotonically. In contrast, anisotropic Gaussian features degrade stability. As the condition number increases from $\kappa = 10$ (Fig. 19) to $\kappa = 100$ (Fig. 20), convergence becomes slower, with longer transient phases and higher sensitivity to task transitions. This reflects the growing imbalance in spread along different feature directions, which biases SGD updates toward dominant directions.

The corresponding arrow visualizations in Figs. 24–26 show the evolution of the error vector $e_t = w_t - w^*$ in parameter space through time. Each arrow represents a successive SGD update, forming the trajectory of $e_t$ over time. Direction and magnitude are encoded geometrically, while color indicates local change in error norm (red: increase, blue: decrease). This provides a direct view of transient divergence and contraction patterns, complementing the scalar norm plots.

**Laplace Distribution.** Figures 21–23 show results for Laplace features. The isotropic Laplace case (Fig. 21) converges to zero tracking error, but exhibits non-monotonic behavior with intermittent spikes in the norm of tracking error vector, highlighted by red dashed markers indicating steps where $\|e_{t+1}\|_2 > \|e_t\|_2$. Arrow plots in Figs. 27–29 provide a finer-grained view of these dynamics by visualizing $e_t$ in parameter space. Each arrow corresponds to a single SGD update, with color coding identical to the Gaussian case (red for norm increase, blue for decrease). Anisotropy further amplifies instability. In the anisotropic Laplace settings (Figs. 22 and 23), we observe more frequent norm increase, delayed convergence, and stronger transient deviations than both isotropic Laplace and Gaussian cases. These effects are consistently reflected in both norm curves and trajectory-level arrow visualizations, where repeated tracking error norm increase occur before eventual contraction.

Overall, the results indicate that both anisotropy and heavy-tailed feature distributions reduce stability, further supporting the use of isotropic Gaussian features as a favorable choice for the embedding distribution in non-stationary settings. Figure 30 provides a direct comparison between the best-performing case (isotropic Gaussian) and the worst-performing case (anisotropic Laplace), showing both error norm trajectories and arrow-based visualizations. As can be seen, the isotropic Gaussian case re-converges monotonically to zero tracking error after each task change, whereas the anisotropic Laplace case exhibits repeated increases in the error norm and fails to converge back to zero tracking error. To see whether these conclusions extend beyond 2-dimensional settings, we repeat the same experiment in higher dimensions. The results are shown in Figure 31, where the top panel corresponds to $d = 8$ and the bottom panel to $d = 32$. As can be seen, in both cases the isotropic Gaussian setting continues to exhibit monotonic re-convergence to zero tracking error, while anisotropy leads to non-convergent or unstable behavior. Similarly, Laplace distributions show frequent spikes in the tracking error norm.

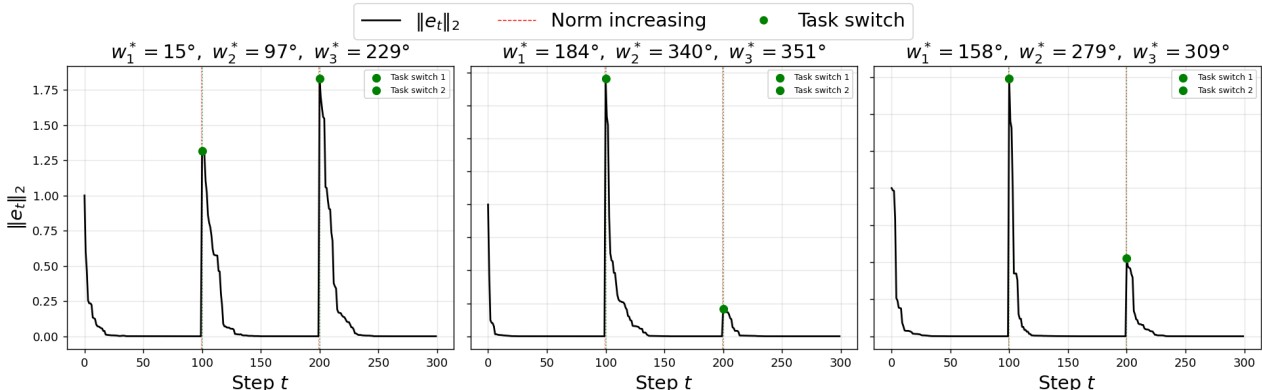

*Figure 18.* Tracking error norm $\|e_t\|_2 = \|w_t - w^*\|_2$ over all steps for **isotropic Gaussian distribution**, across three random seeds (columns). The learner tracks a sequence of three target weights, $w_1^* \overset{t=100}{\longrightarrow} w_2^* \overset{t=200}{\longrightarrow} w_3^*$, each drawn uniformly at random from the unit sphere. The corresponding target angles $(\theta_1, \theta_2, \theta_3)$ are reported in each column title. **Green dots** and dotted vertical lines mark the two task-switch points at $t \in \{100, 200\}$. **Dashed red vertical lines** indicate steps at which $\|e_{t+1}\|_2 > \|e_t\|_2$, highlighting transient divergence episodes. Each switch induces an abrupt spike in the tracking error norm due to the change in $w^*$, followed by reconvergence, whose rate and steady-state level depend on the geometry of the sample distribution.

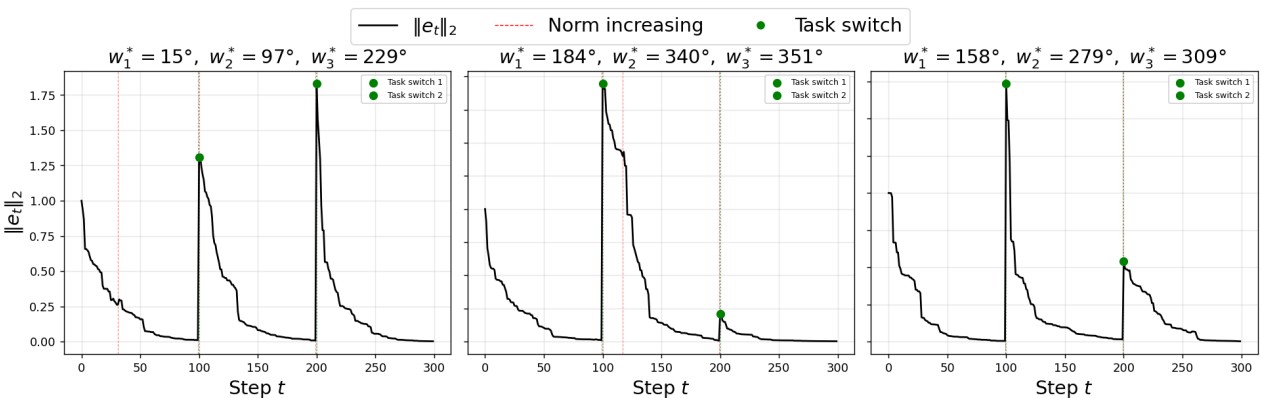

*Figure 19.* Tracking error norm $\|e_t\|_2 = \|w_t - w^*\|_2$ over all steps for **anisotropic Gaussian distribution with condition number** $\kappa = 10$, across three random seeds (columns). The learner tracks a sequence of three target weights, $w_1^* \overset{t=100}{\longrightarrow} w_2^* \overset{t=200}{\longrightarrow} w_3^*$, each drawn uniformly at random from the unit sphere. The corresponding target angles $(\theta_1, \theta_2, \theta_3)$ are reported in each column title. **Green dots** and dotted vertical lines mark the two task-switch points at $t \in \{100, 200\}$. **Dashed red vertical lines** indicate steps at which $\|e_{t+1}\|_2 > \|e_t\|_2$, highlighting transient divergence episodes. Each switch induces an abrupt spike in the tracking error norm due to the change in $w^*$, followed by reconvergence, whose rate and steady-state level depend on the geometry of the sample distribution.

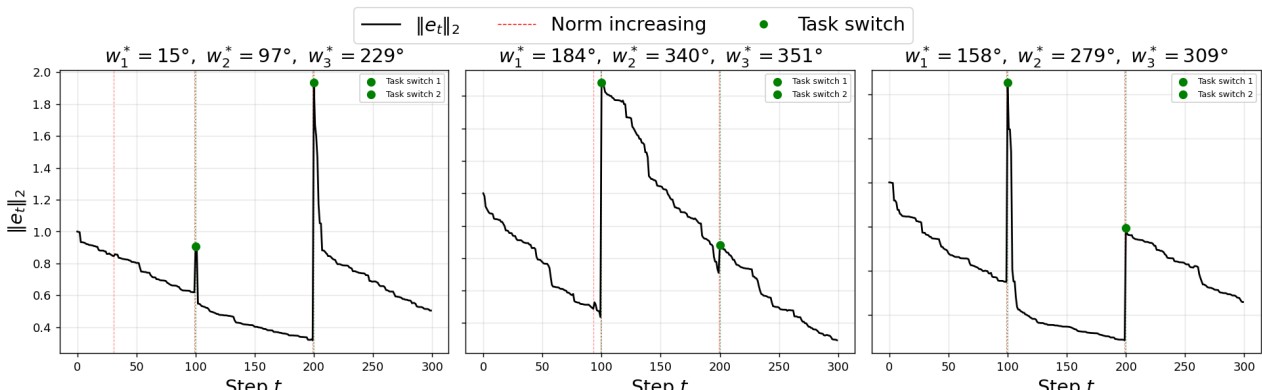

*Figure 20.* Tracking error norm $\|e_t\|_2 = \|w_t - w^*\|_2$ over all steps for **anisotropic Gaussian distribution with condition number** $\kappa = 100$, across three random seeds (columns). The learner tracks a sequence of three target weights, $w_1^* \overset{t=100}{\longrightarrow} w_2^* \overset{t=200}{\longrightarrow} w_3^*$, each drawn uniformly at random from the unit sphere. The corresponding target angles $(\theta_1, \theta_2, \theta_3)$ are reported in each column title. **Green dots** and dotted vertical lines mark the two task-switch points at $t \in \{100, 200\}$. **Dashed red vertical lines** indicate steps at which $\|e_{t+1}\|_2 > \|e_t\|_2$, highlighting transient divergence episodes. Each switch induces an abrupt spike in the tracking error norm due to the change in $w^*$, followed by reconvergence, whose rate and steady-state level depend on the geometry of the sample distribution.

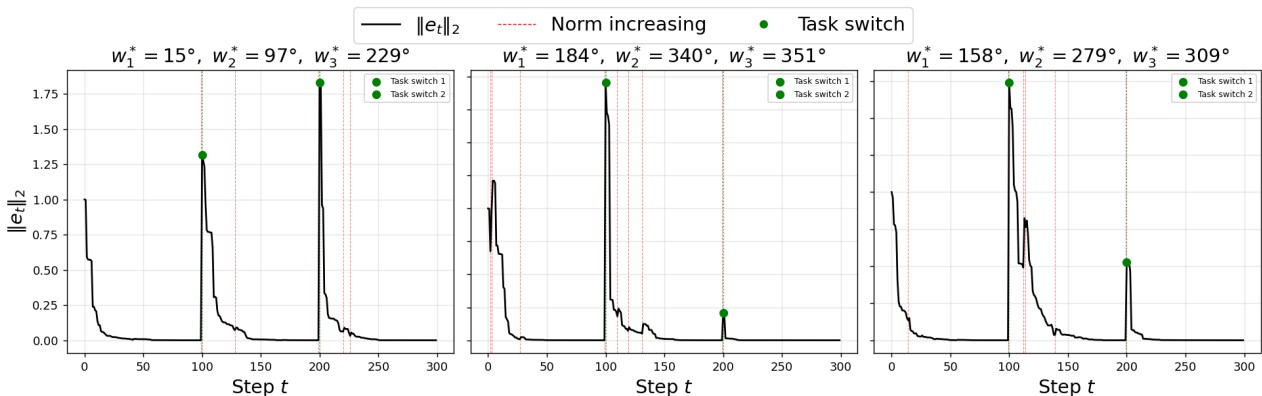

*Figure 21.* Tracking error norm $\|e_t\|_2 = \|w_t - w^*\|_2$ over all steps for **isotropic Laplace distribution**, across three random seeds (columns). The learner tracks a sequence of three target weights, $w_1^* \overset{t=100}{\longrightarrow} w_2^* \overset{t=200}{\longrightarrow} w_3^*$, each drawn uniformly at random from the unit sphere. The corresponding target angles $(\theta_1, \theta_2, \theta_3)$ are reported in each column title. **Green dots** and dotted vertical lines mark the two task-switch points at $t \in \{100, 200\}$. **Dashed red vertical lines** indicate steps at which $\|e_{t+1}\|_2 > \|e_t\|_2$, highlighting transient divergence episodes. Each switch induces an abrupt spike in the tracking error norm due to the change in $w^*$, followed by reconvergence, whose rate and steady-state level depend on the geometry of the sample distribution.

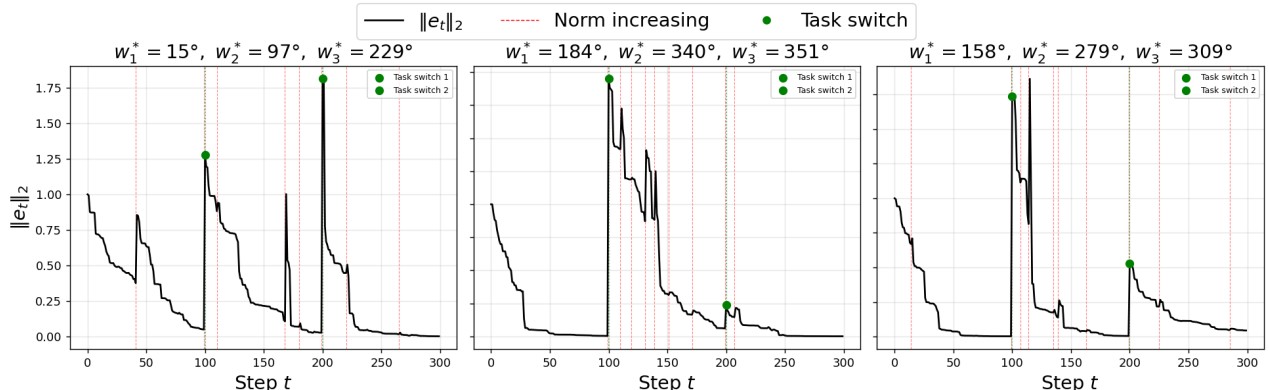

*Figure 22.* Tracking error norm $\|e_t\|_2 = \|w_t - w^*\|_2$ over all steps for **anisotropic Laplace distribution with condition number** $\kappa = 10$, across three random seeds (columns). The learner tracks a sequence of three target weights, $w_1^* \xrightarrow{t=100} w_2^* \xrightarrow{t=200} w_3^*$, each drawn uniformly at random from the unit sphere. The corresponding target angles $(\theta_1, \theta_2, \theta_3)$ are reported in each column title. **Green dots** and dotted vertical lines mark the two task-switch points at $t \in \{100, 200\}$. **Dashed red vertical lines** indicate steps at which $\|e_{t+1}\|_2 > \|e_t\|_2$, highlighting transient divergence episodes. Each switch induces an abrupt spike in the tracking error norm due to the change in $w^*$, followed by reconvergence, whose rate and steady-state level depend on the geometry of the sample distribution.

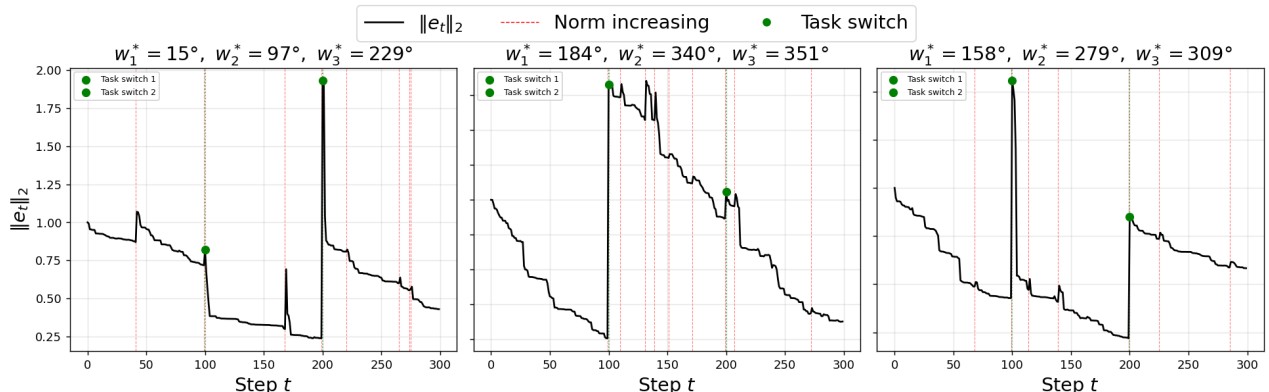

*Figure 23.* Tracking error norm $\|e_t\|_2 = \|w_t - w^*\|_2$ over all steps for **anisotropic Laplace distribution with condition number** $\kappa = 100$, across three random seeds (columns). The learner tracks a sequence of three target weights, $w_1^* \xrightarrow{t=100} w_2^* \xrightarrow{t=200} w_3^*$, each drawn uniformly at random from the unit sphere. The corresponding target angles $(\theta_1, \theta_2, \theta_3)$ are reported in each column title. **Green dots** and dotted vertical lines mark the two task-switch points at $t \in \{100, 200\}$. **Dashed red vertical lines** indicate steps at which $\|e_{t+1}\|_2 > \|e_t\|_2$, highlighting transient divergence episodes. Each switch induces an abrupt spike in the tracking error norm due to the change in $w^*$, followed by reconvergence, whose rate and steady-state level depend on the geometry of the sample distribution.

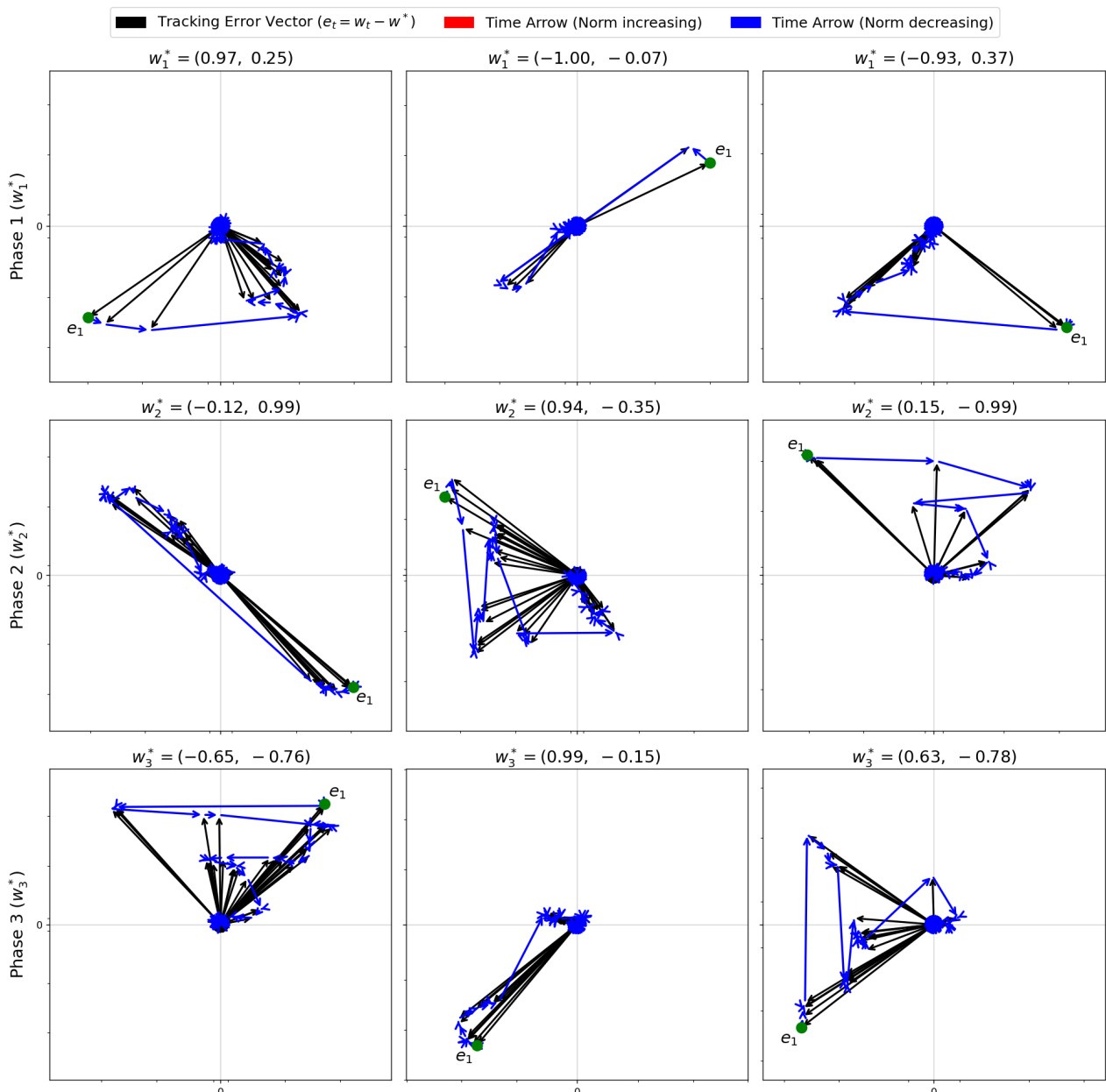

*Figure 24.* Tracking error vectors $e_t = w_t - w^*$ in the 2D weight space for **isotropic Gaussian distribution**. The learner is trained sequentially on three tasks defined by target weights $w_1^*, w_2^*, w_3^*$, each drawn uniformly at random from the unit circle, with task switches at steps $t = 100$ and $t = 200$. Each panel corresponds to one phase of training (rows) and one random seed (columns), and the title indicates the active target $w^*$ for that phase. Within each panel, **black** arrows originate at the origin and point to $e_t$, illustrating the instantaneous direction and magnitude of the tracking error. The **green** dot marks $e_1$, the first error vector immediately after each task switch. **Red** arrows connect consecutive error vectors when the error norm increases, while **blue** arrows indicate decreases in the error norm. Axes are shown on a symmetric logarithmic scale to simultaneously visualize transient spikes following task switches and steady-state residual errors.

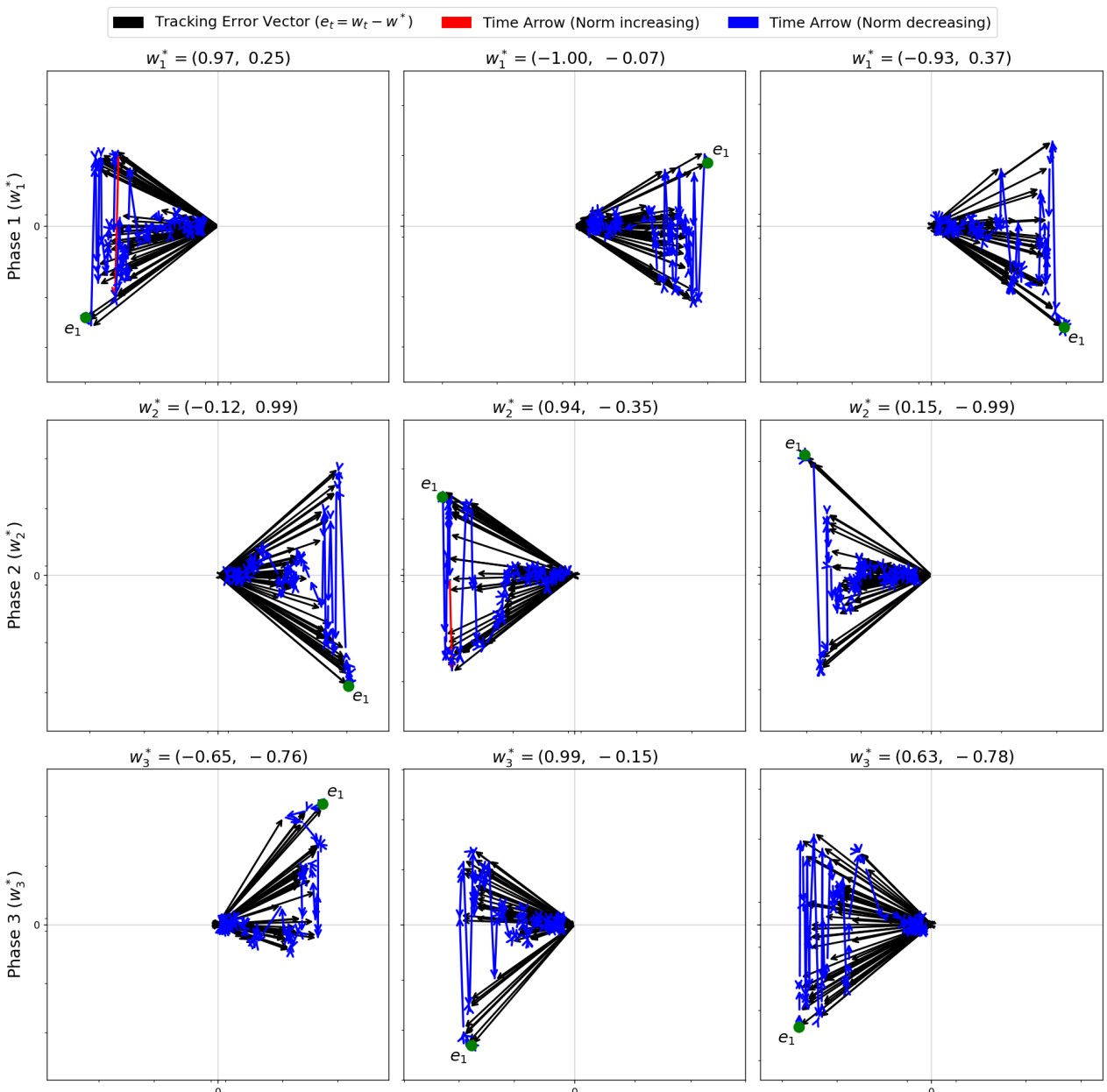

*Figure 25.* Tracking error vectors $e_t = w_t - w^*$ in the 2D weight space for **anisotropic Gaussian distribution with condition number** $\kappa = 10$. The learner is trained sequentially on three tasks defined by target weights $w_1^*, w_2^*, w_3^*$, each drawn uniformly at random from the unit circle, with task switches at steps $t = 100$ and $t = 200$. Each panel corresponds to one phase of training (rows) and one random seed (columns), and the title indicates the active target $w^*$ for that phase. Within each panel, **black** arrows originate at the origin and point to $e_t$, illustrating the instantaneous direction and magnitude of the tracking error. The **green** dot marks $e_1$, the first error vector immediately after each task switch. **Red** arrows connect consecutive error vectors when the error norm increases, while **blue** arrows indicate decreases in the error norm. Axes are shown on a symmetric logarithmic scale to simultaneously visualize transient spikes following task switches and steady-state residual errors.

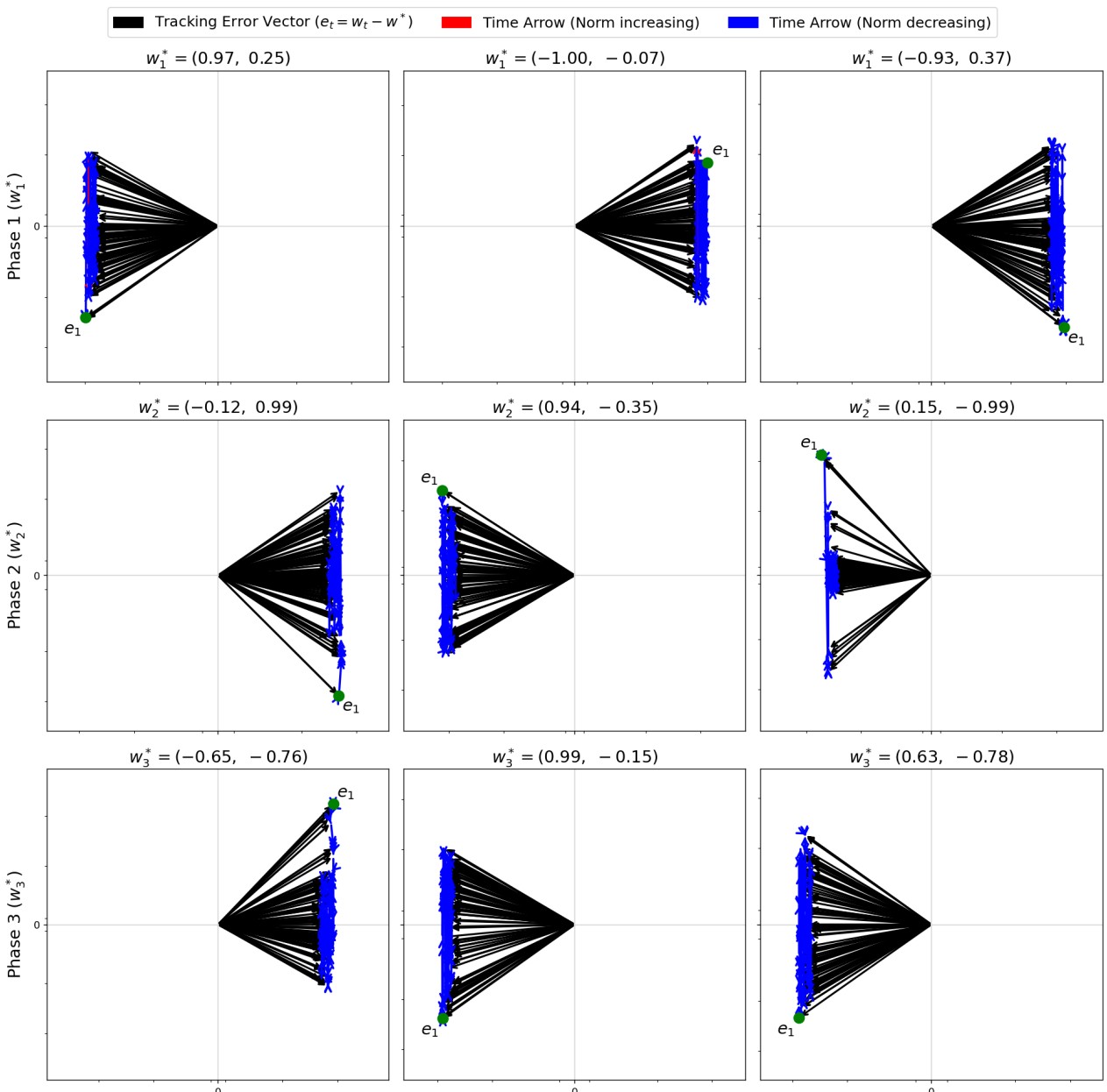

*Figure 26.* Tracking error vectors $e_t = w_t - w^*$ in the 2D weight space for **anisotropic Gaussian distribution with condition number** $\kappa = 100$. The learner is trained sequentially on three tasks defined by target weights $w_1^*, w_2^*, w_3^*$, each drawn uniformly at random from the unit circle, with task switches at steps $t = 100$ and $t = 200$. Each panel corresponds to one phase of training (rows) and one random seed (columns), and the title indicates the active target $w^*$ for that phase. Within each panel, **black** arrows originate at the origin and point to $e_t$, illustrating the instantaneous direction and magnitude of the tracking error. The **green** dot marks $e_1$, the first error vector immediately after each task switch. **Red** arrows connect consecutive error vectors when the error norm increases, while **blue** arrows indicate decreases in the error norm. Axes are shown on a symmetric logarithmic scale to simultaneously visualize transient spikes following task switches and steady-state residual errors.

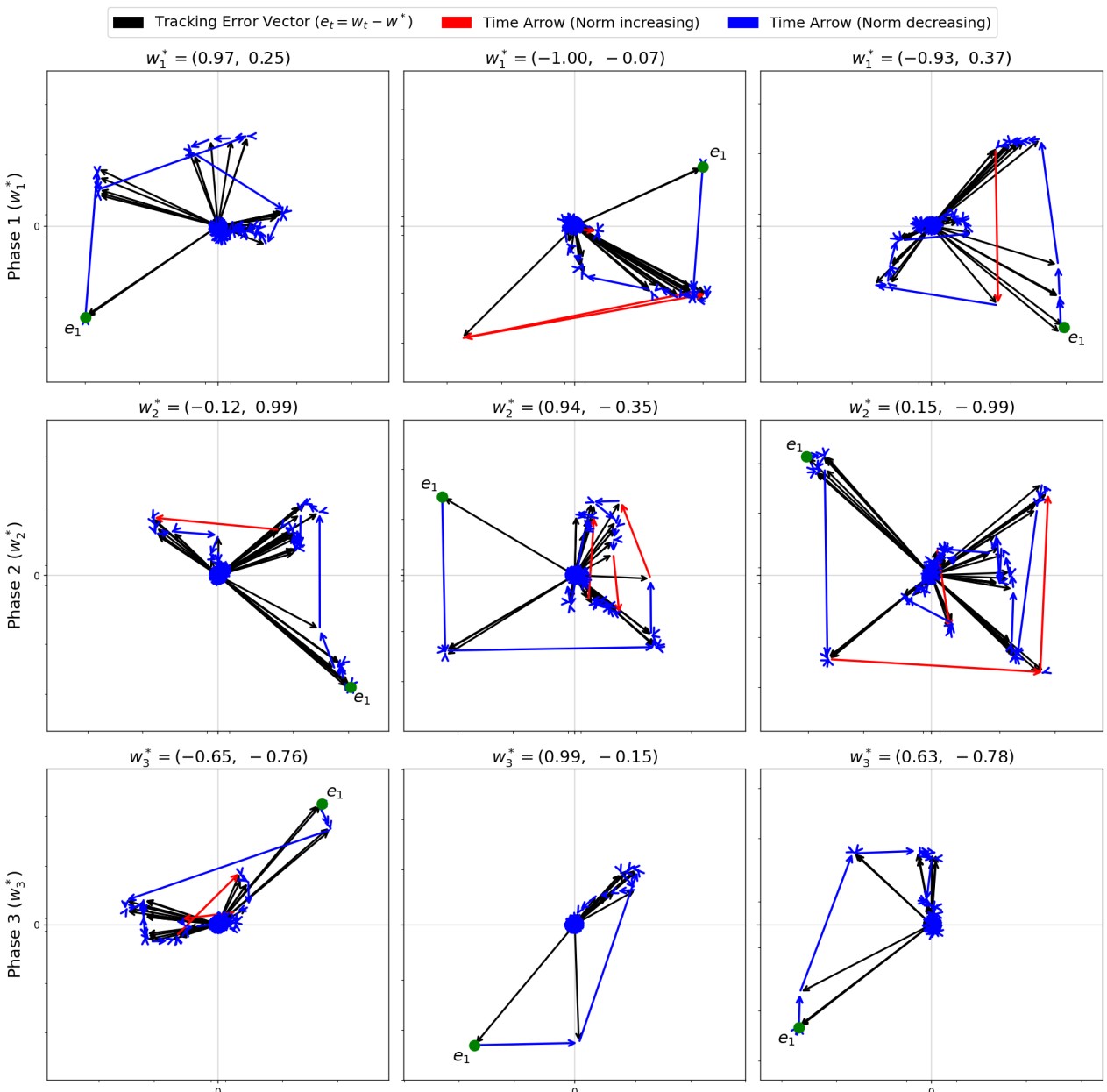

*Figure 27.* Tracking error vectors $e_t = w_t - w^*$ in the 2D weight space for **isotropic Laplace distribution**. The learner is trained sequentially on three tasks defined by target weights $w_1^*, w_2^*, w_3^*$, each drawn uniformly at random from the unit circle, with task switches at steps $t = 100$ and $t = 200$. Each panel corresponds to one phase of training (rows) and one random seed (columns), and the title indicates the active target $w^*$ for that phase. Within each panel, **black** arrows originate at the origin and point to $e_t$, illustrating the instantaneous direction and magnitude of the tracking error. The **green** dot marks $e_1$, the first error vector immediately after each task switch. **Red** arrows connect consecutive error vectors when the error norm increases, while **blue** arrows indicate decreases in the error norm. Axes are shown on a symmetric logarithmic scale to simultaneously visualize transient spikes following task switches and steady-state residual errors.

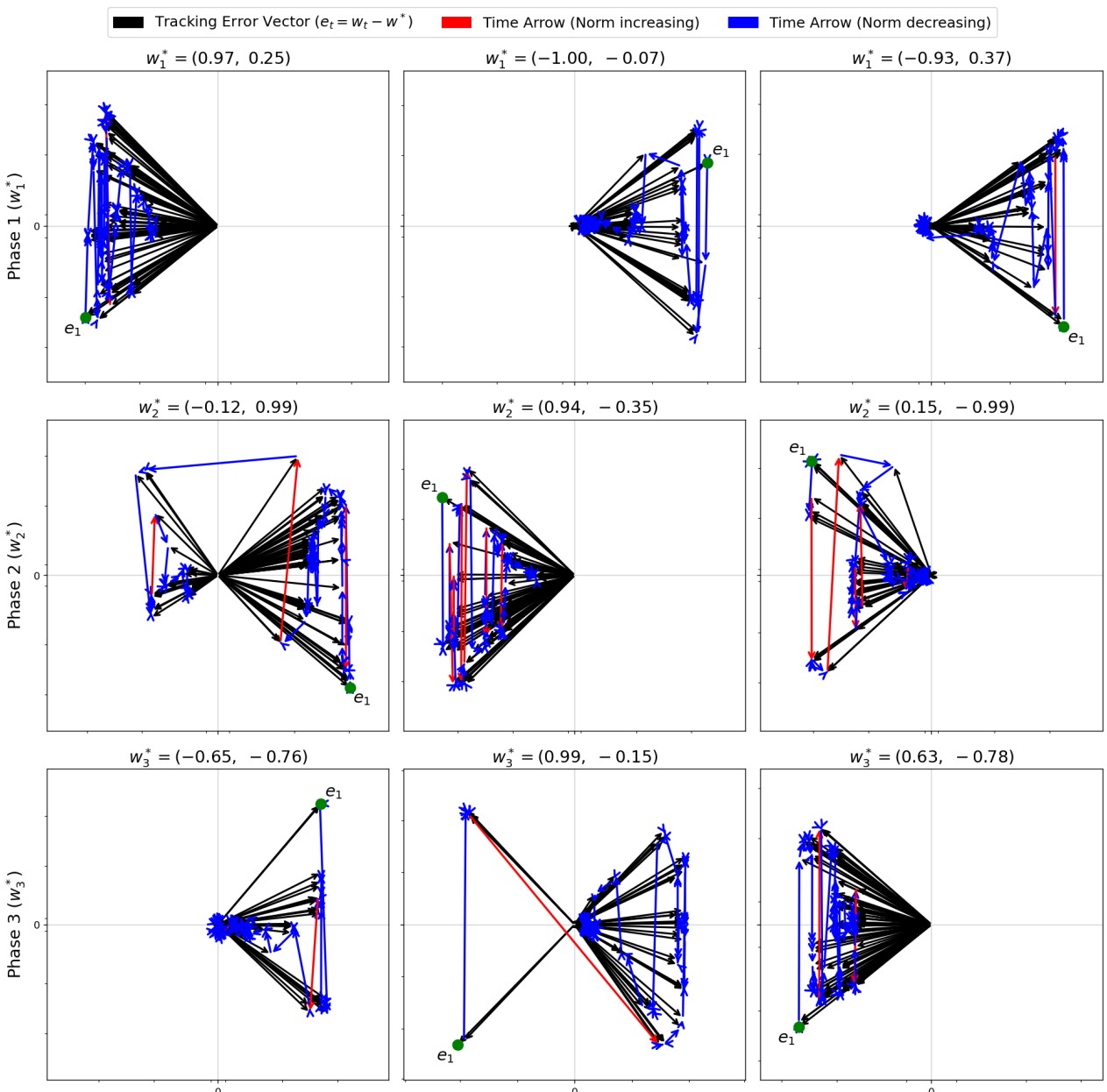

*Figure 28.* Tracking error vectors $e_t = w_t - w^*$ in the 2D weight space for **anisotropic Laplace distribution with condition number** $\kappa = 10$. The learner is trained sequentially on three tasks defined by target weights $w_1^*, w_2^*, w_3^*$, each drawn uniformly at random from the unit circle, with task switches at steps $t = 100$ and $t = 200$. Each panel corresponds to one phase of training (rows) and one random seed (columns), and the title indicates the active target $w^*$ for that phase. Within each panel, **black** arrows originate at the origin and point to $e_t$, illustrating the instantaneous direction and magnitude of the tracking error. The **green** dot marks $e_1$, the first error vector immediately after each task switch. **Red** arrows connect consecutive error vectors when the error norm increases, while **blue** arrows indicate decreases in the error norm. Axes are shown on a symmetric logarithmic scale to simultaneously visualize transient spikes following task switches and steady-state residual errors.

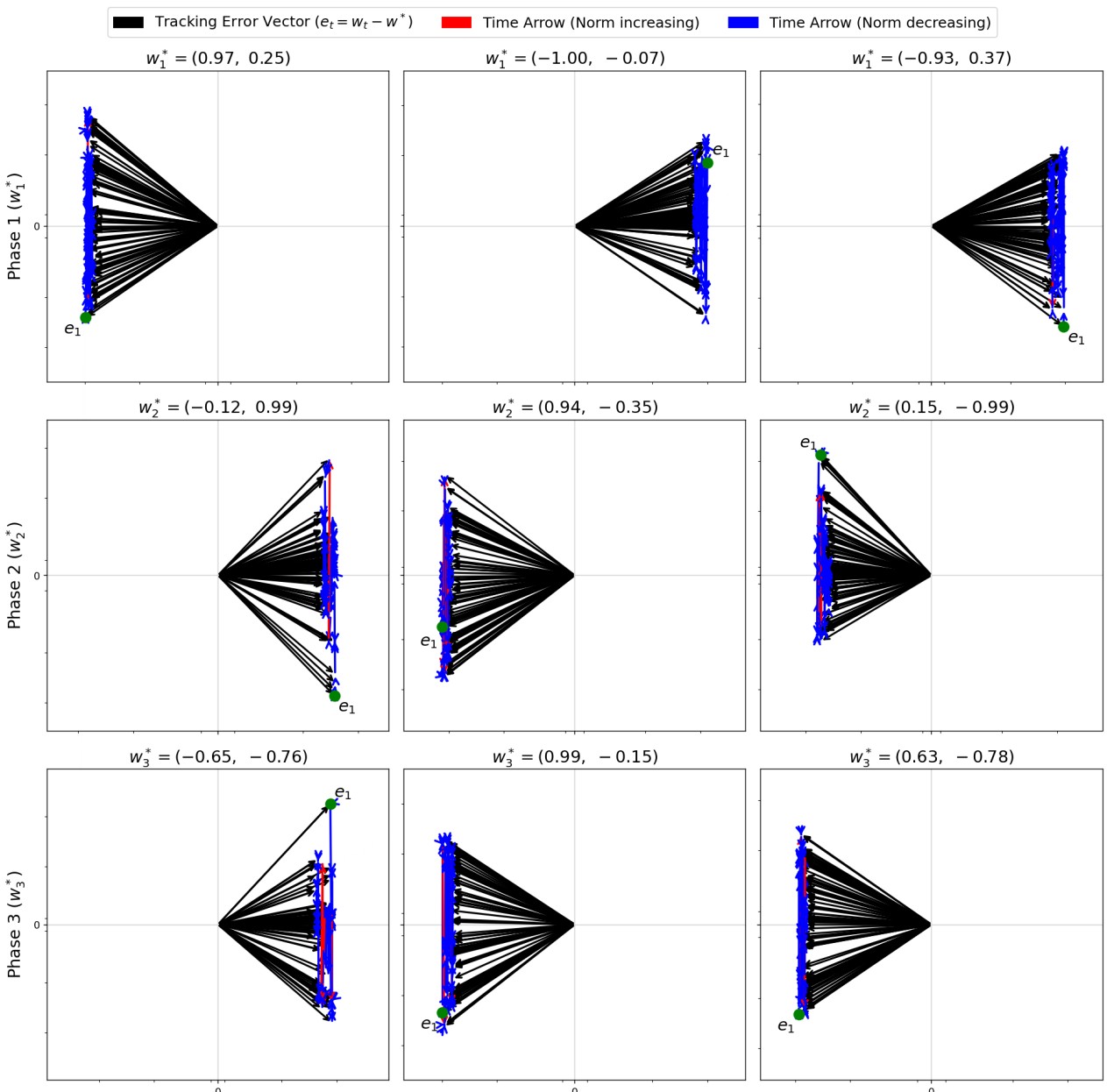

*Figure 29.* Tracking error vectors $e_t = w_t - w^*$ in the 2D weight space for **anisotropic Laplace distribution with condition number** $\kappa = 100$. The learner is trained sequentially on three tasks defined by target weights $w_1^*, w_2^*, w_3^*$, each drawn uniformly at random from the unit circle, with task switches at steps $t = 100$ and $t = 200$. Each panel corresponds to one phase of training (rows) and one random seed (columns), and the title indicates the active target $w^*$ for that phase. Within each panel, **black** arrows originate at the origin and point to $e_t$, illustrating the instantaneous direction and magnitude of the tracking error. The **green** dot marks $e_1$, the first error vector immediately after each task switch. **Red** arrows connect consecutive error vectors when the error norm increases, while **blue** arrows indicate decreases in the error norm. Axes are shown on a symmetric logarithmic scale to simultaneously visualize transient spikes following task switches and steady-state residual errors.

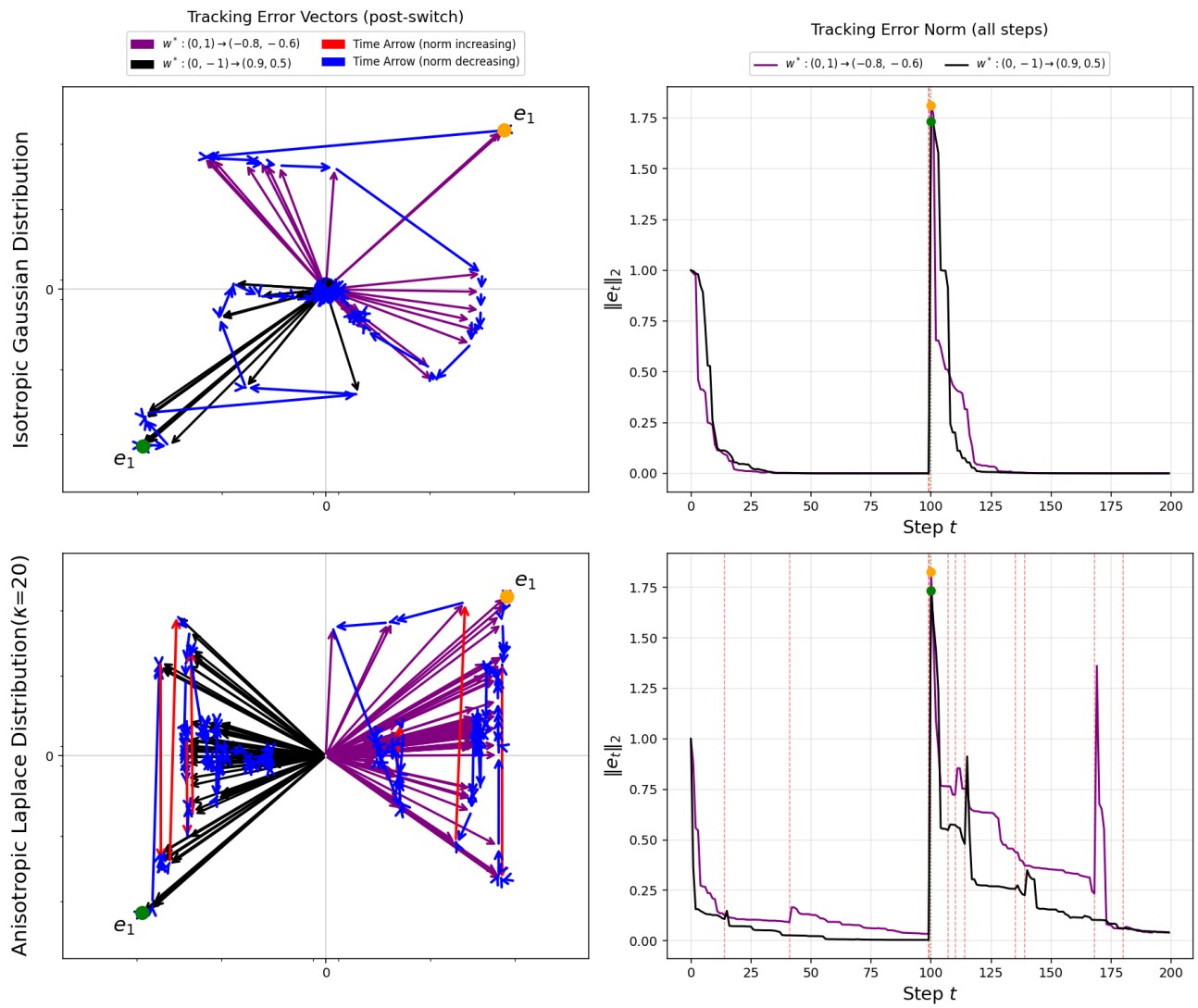

*Figure 30.* Tracking error dynamics under **isotropic Gaussian** and **anisotropic Laplace** feature distributions. The **left column** shows the evolution of the tracking error vector $e_t = w_t - w^*$ after the task switch, while the **right column** shows the corresponding error norm $\|e_t\|_2$ over time. The task switches at $t = T_{\text{switch}}$ from $w_1^* = (0, \pm 1)$ to $w_2^* = (\cdot, \cdot)$ (shown in the legend), with two scenarios overlaid in each panel. **Arrow plots (left).** Each arrow from the origin represents $e_t$ at a given step; purple and black arrows correspond to the two configurations. Colored arrows between successive tips indicate temporal evolution: red for an increase in $\|e_t\|_2$, blue for a decrease. The first post-switch error $e_1$ is highlighted by a color-coded dot. Axes use a symmetric log scale to capture small and large deviations. **Norm plots (right).** Solid curves show $\|e_t\|_2$ for each scenario; vertical dashed red lines mark norm increases and dotted lines indicate the task switch, with markers highlighting the error at switching. **Initialization.** $w_1^* = (0, \pm 1)$ aligns with the principal direction of the feature distribution (largest eigenvalue of covariance), which in the anisotropic case is the dominant eigenvector of $\Sigma$. **Anisotropic distribution.** For anisotropic Laplace, features are sampled independently with coordinate scales to match a diagonal covariance matrix with eigenvalues $\left(\frac{2}{1+\kappa}, \frac{2\kappa}{1+\kappa}\right)$ (trace 2, condition number $\kappa$), ensuring both distributions have the same total variance. **Observation.** While isotropic Gaussian features show near-monotone error contraction, anisotropic Laplace features lead to frequent increases and non-monotone trajectories that plateau before reaching zero, highlighting the impact of anisotropy and non-Gaussianity on tracking stability.

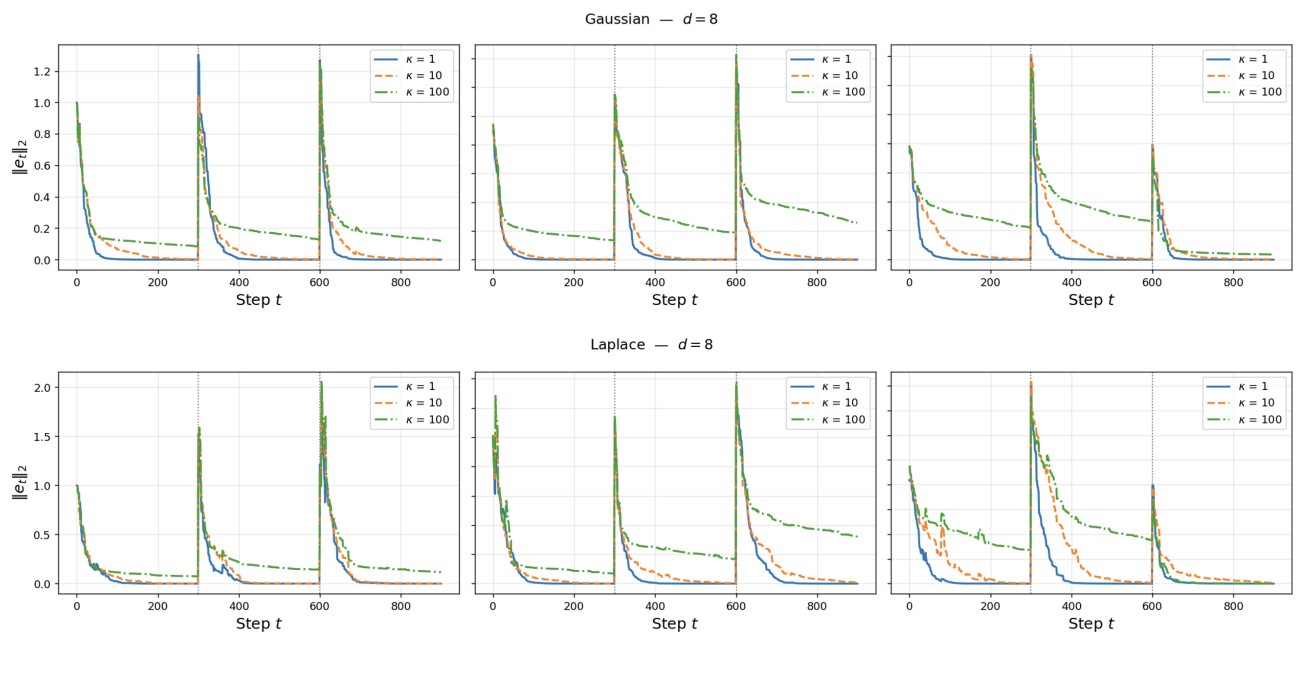

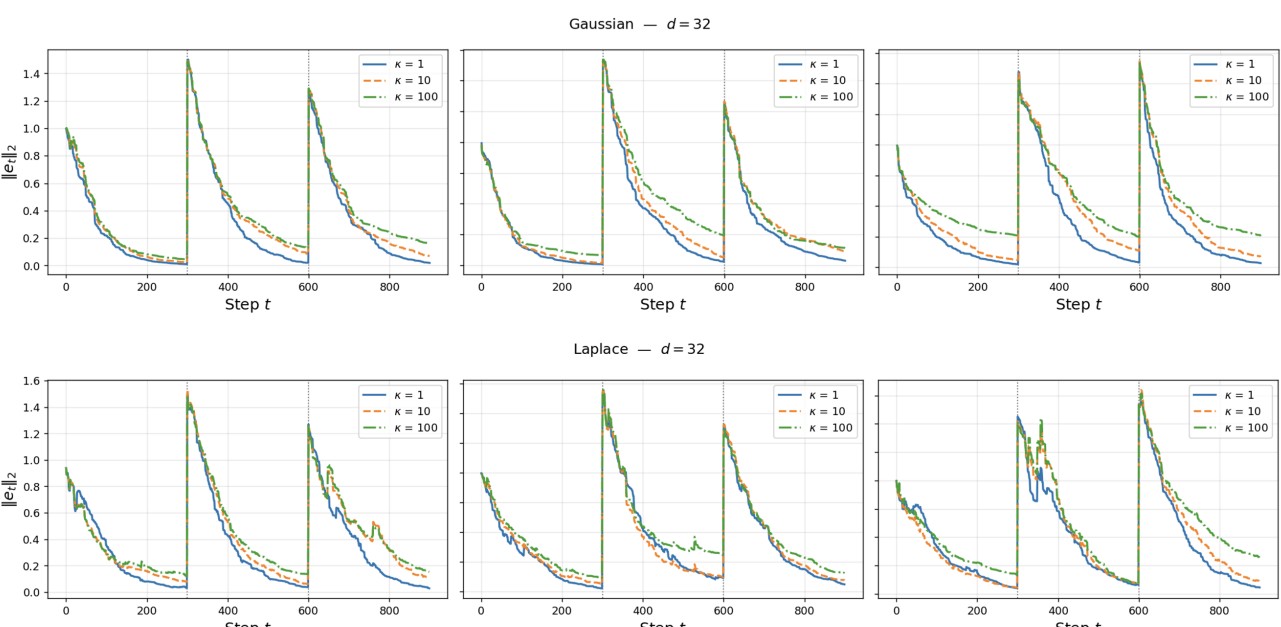

*Figure 31.* **Tracking error dynamics under different input distributions.** Evolution of the tracking error $\|e_t\|_2 = \|w_t - w^*\|_2$ over time for **Gaussian** and **Laplace** input distributions with varying condition numbers ($\kappa$), shown across three random seeds (columns). The top row corresponds to $d = 8$ and the bottom row to $d = 32$. The learner tracks a sequence of target weights, $w_1^* \xrightarrow{t=100} w_2^* \xrightarrow{t=200} w_3^*$, each sampled uniformly from the unit sphere. Each target switch induces a sharp increase in error, followed by reconvergence, whose rate and stability depend on the geometry of the input distribution. Across both dimensions and seeds, isotropic Gaussian inputs yield smooth, monotonic, and rapid convergence to zero error, whereas isotropic Laplace inputs also converge but exhibit pronounced transient spikes. Increasing the condition number (i.e., reduced isotropy) leads to slower convergence and, in some cases, failure to fully converge.

### F.2. Existence of Divergent Drift Directions

We further investigate the existence of divergent drift directions under anisotropic Gaussian feature distributions. In particular, we test the hypothesis that when the tracking error aligns with the eigenvector corresponding to the smallest eigenvalue of the input covariance in the reverse direction, i.e., $e_t = -v_d$, the dynamics may fail to converge to a zero steady-state error. To study this, we construct a controlled setting in which Gaussian inputs with different condition numbers are used and the underlying optimal weight is chosen at each step such that the error vector is aligned with $-\beta v_d$, where $v_d$ denotes the eigenvector associated with the smallest eigenvalue of the covariance matrix.

As shown in Fig. 32, the behavior of the system strongly depends on the conditioning of the feature distribution. For isotropic features ($\kappa = 1$), the dynamics remain stable and converge despite the imposed alignment. However, as the condition number increases, the system exhibits progressively stronger divergence, indicating the emergence of unstable drift directions along low-variance eigenmodes. We further observe that the magnitude parameter $\beta$ controls the severity of this effect: larger values of $\beta$ amplify divergence, while smaller values may still preserve stability under moderate conditioning (e.g., $\kappa = 5$ with $\beta = 2.5$). Overall, these results demonstrate that anisotropy induces specific directions in parameter space along which SGD becomes unstable, leading to persistent drift rather than convergence.

## G. Comparison with Other Regularization Methods

Beyond the methods included in Table 2, including covariance whitening (VICReg), various isotropic target distributions, and ablations that control tail behavior or symmetry through the SIGReg loss, we further evaluate additional regularization techniques such as L2 regularization (Kumar et al., 2023), simplicial embeddings (Obando-Ceron et al., 2026), and weight orthogonalization (Chung et al., 2024). These approaches have previously been shown to improve representation quality and performance by mitigating representation collapse, or shaping the geometry of the embedding space. Including them allows us to isolate the effect of isotropic Gaussianity from more general mechanisms such as anti-collapse behavior or generic feature spreading and rank maximization.

The results on the Atari-10 benchmark using the PQN algorithm are summarized in Table 10. While several of these methods yield measurable improvements over the baseline, explicitly enforcing isotropic Gaussian embeddings through the full SIGReg loss achieves the strongest overall performance. This suggests that the gains are not solely due to improved representation spreading, but are specifically driven by matching the geometry of an isotropic Gaussian distribution. Among methods that do not use SIGReg, weight orthogonalization and covariance whitening improve a similar percentage of games. However, the average improvement is higher when using the full SIGReg loss. Overall, the findings in Table 10 further support the conclusion that isotropic Gaussian structure provides a stronger and more consistent inductive bias than alternative regularization strategies. Embedding norm regularization performs the worst among the baselines, while simplicial embeddings improve over L2 normalization but still exhibit a substantial gap relative to isotropic Gaussian embeddings. Weight orthogonalization achieves a comparable number of improved games to isotropic Gaussian methods but yields lower average gains, in addition to requiring computation of the full Gram matrix of weights, which is computationally expensive. VICReg, which encourages isotropy without explicitly enforcing Gaussianity, improves a similar fraction of games but does not match the average performance of isotropic Gaussian embeddings.

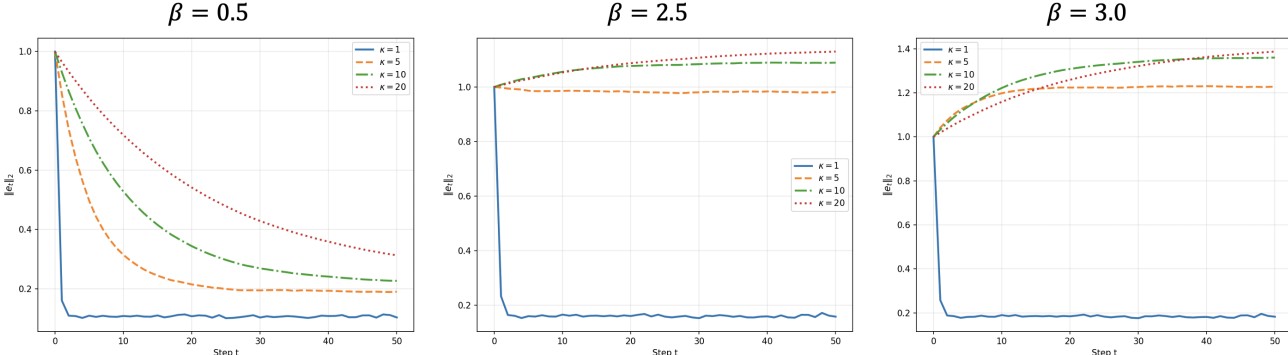

*Figure 32.* **Existence of divergent directions under anisotropic Gaussian distributions.** We test the hypothesis that when the tracking error aligns with the eigenvector corresponding to the smallest eigenvalue of the input covariance in the reverse direction ($e_t = -v_d$), the dynamics fail to converge to zero steady-state error. Specifically, for Gaussian inputs, we initialize the task such that the error vector is aligned with $-v_d$, where $v_d$ is the eigenvector associated with the smallest eigenvalue, and subsequently update the target weights so that $e_t = -\beta v_d$ at each step. Results show that larger condition numbers ($\kappa$) lead to increased divergence, whereas the isotropic case ($\kappa = 1$) remains convergent. Moreover, increasing $\beta$ amplifies divergence; however, for moderate conditioning (e.g., $\kappa = 5$), smaller values of $\beta$ (e.g., $\beta = 2.4$) do not induce divergence.

| Method | % of Games Improved (↑) | Avg. Improvement (↑) |
|---|---|---|
| Gaussian (Real and Imaginary parts) | 90% | 305% |
| Laplacian | 90% | 234% |
| Logistic | 90% | 209% |
| Real Part Only | 100% | 151% |
| Imaginary Part Only | 70% | 56% |
| Covariance Whitening (VICReg) | 90% | 144% |
| Embedding Norm Regularization | 60% | -1% |
| Simplicial Embedding | 70% | 17% |
| Weight Orthogonalziation | 90% | 219% |

*Table 10.* **Comparison with alternative regularization methods.** We report the percentage of games improved over the baseline and the average improvement across Atari-10 using the PQN algorithm. **Top:** different isotropic target distributions. **Middle:** minimizing different components. **Bottom:** comparison with alternative regularization methods. Among the baselines, embedding norm regularization performs the worst. Simplicial embedding improves over L2 normalization, but a substantial gap remains compared to isotropic Gaussian embeddings. Weight orthogonalization improves a similar number of games as isotropic Gaussian, but yields lower average improvement; moreover, it requires computing the Gram matrix of all weights, which is computationally expensive. VICReg, which promotes isotropy without enforcing Gaussianity, improves a similar number of games but does not match the average improvement of isotropic Gaussian embeddings.

# H. Spectral Analysis

To analyze the effect of the SIGReg loss on the geometry of learned representations, we examine the distribution of eigenvalues of the embedding covariance matrix using two metrics. First, we measure the entropy of the eigenvalue spectrum (Table 11). Higher entropy indicates a more uniform distribution of eigenvalues across directions, reflecting reduced representation collapse and improved isotropy. Second, we report the ratio of condition numbers between the baseline and SIGReg embeddings (Table 12), defined as $\kappa_{\text{baseline}}/\kappa_{\text{SIGReg}}$. Higher values of this ratio indicate a stronger reduction in the condition number under SIGReg. From a theoretical perspective, a lower condition number implies a more balanced curvature across directions, leading to more uniform convergence rates and better controlled drift dynamics in representation learning.

The results in Table 11 show that SIGReg consistently increases eigenvalue entropy compared to the baseline, indicating a more evenly spread spectrum. Furthermore, increasing the regularization strength $\lambda$ further amplifies this effect, leading to progressively higher entropy, as expected from the objective's tendency to equalize eigenvalues. When comparing across algorithms, PPO generally exhibits higher final entropy than PQN, suggesting that PPO induces less severe representation collapse under the same training conditions. However, the effect of SIGReg is not uniformly beneficial across all environments: while games such as Amidar and Qbert show consistent improvements in both PQN and PPO, environments such as Freeway, Pitfall, Atlantis, and Venture exhibit cases where SIGReg reduces performance relative to the baseline, despite increased entropy.

Table 12 further supports these findings by showing that SIGReg substantially reduces the condition number relative to the baseline across settings. In particular, $\lambda = 1$ achieves a significantly lower condition number, consistent with the theoretical expectation that SIGReg loss minimization leads to more isotropic representations and smaller condition number as the result. Together, these results indicate that SIGReg not only promotes a more uniform eigenvalue distribution but also improves the conditioning of the representation space, which is closely linked to improved optimization and stability properties.

| | PQN | | | | | PPO | | | |
|---|---|---|---|---|---|---|---|---|---|
| Game | $\lambda$ | Start | Mid | End | Game | $\lambda$ | Start | Mid | End |
| Amidar | 0.0 | 5.02 | 1.26 | 1.29 | Amidar | 0.0 | 20.96 | 4.01 | 3.45 |
| | 0.2 | 5.02 | 10.63 | 10.45 | | 0.2 | 20.96 | 11.93 | 14.08 |
| | 1.0 | 5.02 | 15.66 | **17.93** | | 1.0 | 20.96 | 15.19 | **25.75** |
| Qbert | 0.0 | 4.32 | 1.46 | 1.39 | Qbert | 0.0 | 19.60 | 2.85 | 3.24 |
| | 0.2 | 4.32 | 8.00 | 7.52 | | 0.2 | 19.60 | 12.78 | 15.20 |
| | 1.0 | 4.32 | 14.61 | **14.52** | | 1.0 | 19.60 | 15.83 | **31.57** |
| Freeway | 0.0 | 22.72 | 1.10 | 1.37 | Atlantis | 0.0 | 34.05 | 1.94 | 2.86 |
| | 0.2 | 22.72 | 27.20 | 34.30 | | 0.2 | 34.05 | 16.62 | 20.96 |
| | 1.0 | 22.72 | 26.47 | **35.26** | | 1.0 | 34.05 | 38.45 | **43.99** |
| Pitfall | 0.0 | 3.56 | 1.10 | 1.04 | Venture | 0.0 | 33.74 | 4.59 | 1.82 |
| | 0.2 | 3.56 | 7.22 | 7.56 | | 0.2 | 33.74 | 10.16 | 14.50 |
| | 1.0 | 3.56 | 20.44 | **28.47** | | 1.0 | 33.74 | 21.59 | **20.70** |

*Table 11.* **Entropy of eigenvalues.** Comparison of eigenvalue entropy at the start, middle, and end of training for PQN (left) and PPO (right) across different games and $\lambda$ values. Amidar and Qbert are selected as representative cases where SIGReg improves performance in both PQN and PPO. Freeway, Pitfall, Atlantis, and Venture are cases where adding the SIGReg loss degrades performance relative to the baseline. Comparing the same games and baseline settings, PPO consistently exhibits higher final entropy, indicating less severe representation collapse. For a fixed game, increasing $\lambda$ leads to higher entropy, as expected, since the SIGReg loss promotes equalization of eigenvalues.

| | PQN | | Game | PPO | |
|---|---|---|---|---|---|
| Game | $\lambda$ | Avg. Condition Number Ratio | Game | $\lambda$ | Avg. Condition Number Ratio |
| Amidar | 0.2 | $8.8 \times 10^3$ | Amidar | 0.2 | $8.3 \times 10^1$ |
| | 1.0 | $2.1 \times 10^4$ | | 1.0 | $1.8 \times 10^2$ |
| Qbert | 0.2 | $1.2 \times 10^1$ | Qbert | 0.2 | $1.3 \times 10^2$ |
| | 1.0 | $6.7 \times 10^3$ | | 1.0 | $4.3 \times 10^2$ |
| Freeway | 0.2 | $8.5 \times 10^5$ | Atlantis | 0.2 | 3.9 |
| | 1.0 | $3.9 \times 10^6$ | | 1.0 | $5.7 \times 10^2$ |
| Pitfall | 0.2 | $1.9 \times 10^1$ | Venture | 0.2 | 0.4 |
| | 1.0 | $6.0 \times 10^1$ | | 1.0 | $4.9 \times 10^5$ |

*Table 12.* **Condition number ratio averaged over training.** We report the time-averaged ratio of condition numbers between the baseline and SIGReg embeddings, $\frac{\kappa_{\text{baseline}}}{\kappa_{\text{SIGReg}}}$. According to our theoretical analysis, a lower condition number is desirable, as it indicates more uniform convergence rates across directions and more controlled drift term. As shown, $\lambda = 1$ yields a substantially lower condition number than the baseline, consistent with the effect of SIGReg in spreading eigenvalues and promoting isotropy.

