# OpenReview forum: "Stable Deep Reinforcement Learning via Isotropic Gaussian Representations"
_ICML.cc/2026/Conference — ICML 2026 spotlight_

### Official Review · Reviewer_b8zb · 2026-03-11

**Soundness:** 3
**Presentation:** 3
**Significance:** 2
**Originality:** 3
**Overall Recommendation:** 5
**Confidence:** 4

**Summary:**

This paper argues that isotropic Gaussian embeddings are suitable for deep RL, just as it does in LeJEPA paper for self-supervised learning. It proposes a theoretical rationale for why it should be true under non-stationary targets. It also tests this idea in a broad deep RL settings.

**Compliance With Llm Reviewing Policy:**

Affirmed.

**Final Justification:**

The response fully resolves my concerns.

**Key Questions For Authors:**

Why are the gains much stronger in PQN than in PPO?
How much of the gain comes from isotropic Gaussianity versus other generic feature spreading or anti-collapse regularization?

**Limitations:**

The paper is technically sound and clearly written, but its novelty is limited by substantial conceptual overlap with LeJEPA. The empirical evidence would be significantly stronger with comparisons to alternative representation regularizers.

**Strengths And Weaknesses:**

Soundness: The submission is technically sound and the claims are well supported by both theoretical analysis and experimental results with appropriate methods.The empirical scope is good for an RL paper.
Presentation: The submission is clearly written, well-structured, and easy to follow. The work was inspired by a prior work in self-supervised learning and applied it to deep RL.
Significance: The paper addresses the problem of stability in RL. The representation regularizer is lightweight and improves training across multiple RL settings. However, the paper did not introduce the core representation principle but imported it from a prior work.
Originality: This paper was based on another paper in self-supervised learning. It introduces a new theoretical analysis, "linear tracking under non-stationary targets," to explain why it should work in RL and presents the empirical results of SIGReg in RL.

---

> ### Author Rebuttal · Authors · 2026-03-31
>
> We thank the reviewer for their positive and thoughtful feedback. We are pleased that the reviewer found the paper “technically sound” and that “the claims are well supported by both theoretical analysis and experimental results.” We also appreciate the recognition that the submission is “clearly written, well-structured, and easy to follow,” as well as the effectiveness of the regularizer in “improving training across multiple RL settings”. We respond to their main concerns below.
>
>
> > Why are the gains much stronger in PQN than in PPO?
>
> We observe larger gains in PQN primarily due to its off-policy nature and greater sensitivity to representation quality. PQN relies on bootstrapped value estimates which amplifies non-stationarity and makes learning more susceptible to issues such as representation collapse, and degraded gradient flow [1]. In contrast, PPO operates in an on-policy regime with fresher data and more tightly coupled updates between the policy and value function. This reduces distribution mismatch and mitigates some of the instability induced by bootstrapping, resulting in less severe degradation of representation geometry. As a result, improvements from better-conditioned representations (via isotropy and Gaussianity) are more pronounced in PQN, where stabilizing the feature covariance and controlling higher-order variability directly improves tracking under non-stationary targets.
>
> [1.] Castanyer  et al., Stable Gradients for Stable Learning at Scale in Deep Reinforcement Learning, NeurIPS ‘25
>
>
> > How much of the gain comes from isotropic Gaussianity versus other generic feature spreading or anti-collapse regularization?
>
> Our results indicate that both factors matter, but isotropic Gaussianity provides additional benefits beyond generic feature spreading. Beyond the methods included in the main paper (VICReg/covariance whitening, various isotropic distributions, and ablations controlling tails or symmetry via the SIGReg loss), we ran new experiments with alternative regularizers (L2 [1], SEM [2], and weight orthogonalization-based method  [3]). These methods have previously been shown to improve representation quality by promoting feature spreading, reducing collapse, or shaping the geometry of the representation space in deep RL. Including them allows us to disentangle the effect of isotropic Gaussianity from more general anti-collapse or feature-shaping mechanisms. The numerical results are shown in the following table (also available here: https://anonymous.4open.science/r/anonymous-icml-2026-2771/regularizers_comparison.png with more detailed explanation). While several of these methods provide measurable gains, imposing isotropic Gaussian distribution via full SIGReg loss consistently achieve stronger improvements, indicating that the benefits go beyond standard regularization effects.
> | Method                               | % of Games Improved (↑) | Avg. Improvement (↑) |
> |--------------------------------------|--------------------------|----------------------|
> | Gaussian (Real and Imaginary parts)  | 90%                      | 305%                 |
> | Laplacian                            | 90%                      | 234%                 |
> | Logistic                             | 90%                      | 209%                 |
> | Real Part Only                       | 100%                     | 151%                 |
> | Imaginary Part Only                  | 70%                      | 56%                  |
> | Vicreg (Covariance Whitening)   | 90%                      | 144%                 |
> | L2 [1]                               | 60%                      | -1%                  |
> | SEM [2]                           | 70%                      | 17%                  |
> | Weight Ortho [3]                 | 90%                      | 219%                 |
>
> [1.] Kumar et al., Offline Q-learning on Diverse Multi-Task Data Both Scales And Generalizes, ICLR ‘23
>
> [2.] Obando-Ceron et al., Simplicial Embeddings Improve Sample Efficiency in Actor-Critic Agents, ICLR ‘26
>
> [3.] Chung et al., Parseval regularization for continual reinforcement learning, NeurIPS ‘24

---

> > ### Author Rebuttal · Reviewer_b8zb · 2026-04-04
> >
> > Thank you for the detailed rebuttal. Your response fully resolves my concerns.

---

> > > ### Author Response · Authors · 2026-04-08
> > >
> > > Thank you for your positive feedback and for taking the time to engage with our rebuttal. We’re glad the response addressed your concerns. Thanks!
> > >
> > > * Authors

---

### Official Review · Reviewer_DkR6 · 2026-03-13

**Soundness:** 4
**Presentation:** 4
**Significance:** 3
**Originality:** 3
**Overall Recommendation:** 5
**Confidence:** 4

**Summary:**

Deep reinforcement learning has a deserved reputation for being unstable to
train. This paper studies one cause of this instability: "representational
collapse" in which the learned representations concentrate their variance in
a few directions and capacity in the neural network is effectively lost. This
kind of ill-conditioning seems to arise from the intrinsic non-stationarity
of the RL problem and it seems to make gradient-based learning difficult.

This paper proposes a regularizer to encourage the representations to be roughly
distributed according to a spherical Gaussian distribution. This prevents
collapse to the ill-conditioned situation.  The regularization is achieved by
performing random projections and then matching the empirical characteristic
function.

The paper provides a theoretical arguments based on a simplified linear value
function model and study the case as a tracking problem.  The analysis shows
that the tracking dynamics depend on the spectrum of the feature covariance
matrix and isotropic features reduce the condition number and encourage better
tracking.  The Gaussian specifically is motivated via an argument that one
should want light tails.

The paper studies the problem empirically and shows that standard RL training
leads to increasingly anisotropic representations (measured by feature rank and
neuron dormancy), and that the proposed regularizer maintains higher effective
rank and improves learning performance across several RL benchmarks.

**Compliance With Llm Reviewing Policy:**

Affirmed.

**Key Questions For Authors:**

Table 2 shows that VICReg-style whitening alone is insufficient and that
Gaussian targets outperform heavier-tailed alternatives.  Does this gap persist
across all environments, or are there regimes (e.g., sparse-reward games)
where the simpler whitening penalty closes the gap?

**Limitations:**

yes

**Strengths And Weaknesses:**

Strengths

I like framing value-function learning as a tracking problem under drifting
targets.  This is a natural way to think about deep RL instability and it
leads to a clean analysis: the contraction rate of the tracking dynamics
depends on the spectrum of the feature covariance, and isotropic features
equalize contraction across directions.  The connection between representation
conditioning and learning stability makes sense to me.

The method is seems easy to implement and computationally light.  Using random
projections avoids explicit covariance estimation and seems likely to scale well.  The
experiments support the claim that standard RL training leads to increasingly anisotropic
representations, and that the regularizer maintains higher effective rank,
and improves performance across several benchmarks.
The ablations (Gaussian vs Laplacian vs Logistic targets, symmetry vs tail
control, VICReg whitening) are thoughtful.

I think this is a strong paper, so these are fairly minor considerations:

The theoretical analysis motivates well-conditioned feature covariance, but the
additional claim that embeddings should specifically follow a Gaussian is less
well supported.  The ablation in Table 2 shows Gaussian targets outperform
Laplacian and Logistic, and that VICReg-style whitening alone is insufficient,
which is good evidence.  But any sub-Gaussian distribution with identity
covariance would satisfy the moment-concentration argument in Section 3.3, so
I am not fully convinced that Gaussianity per se is doing the work rather than
light tails more generally.

The empirical evaluation tracks effective rank and neuron dormancy but does not
directly measure the condition number of the feature covariance matrix, which
is important in the theoretical argument (Theorem 3.1).  Plotting
the eigenvalue spectrum over training would be interesting and add support to
the theoretical claims.

The analysis assumes a linear value function with fixed feature covariance,
while the proposed method directly modifies the
representation itself.  It is a bit unclear how directly the theoretical
results transfer to the full nonlinear deep RL setting where both $w$ and
$\phi$ are changing simultaneously.

---

> ### Author Rebuttal · Authors · 2026-03-30
>
> We thank the reviewer for the positive and careful assessment. We are motivated that the reviewer describes our manuscript as “a strong paper” and finds the experiments “thoughtful”. Below, we respond to each of the reviewer’s questions and considerations.
>
> ### Additional Toy Experiments for Empirical Justification
> We conduct extra toy experiments to examine how distribution influences learning stability. Specifically, we consider linear regression trained with SGD to track a time-varying target $w_t^*$, mirroring the linear readout assumption in our analysis. By varying the distribution of $\phi$, we evaluate its effect on optimization dynamics and stability.
> 1. **Isotropy and distribution type:**
> Similar to CIFAR-10 experiments, we test the setup in the case of task change at regular intervals. In 2D, isotropic Gaussian representations exhibit monotonic convergence to zero, while anisotropic settings show unstable or slower behavior (see Figures 18-29 in folder Toy 2D here: https://bit.ly/4tnBiee). Also, we have obtained a similar figure to Figure 1 from this setup (see Figure 30 here: https://bit.ly/4s7iC1g). In higher dimensions, isotropic Gaussian representations consistently yield stable tracking dynamics(see Figure 31 here: https://bit.ly/4bHkLf5). These experiments provide empirical support for the role of isotropy and Gaussian distribution in avoiding poorly conditioned directions and controlling drift.
> 2. **Existence of divergent drift directions:**
> In a high-dimensional setting $d=512$, at each optimization step we construct a worst-case scenario by aligning $e_t$ with $-v_d$, where $v_d$ is the eigenvector corresponding to the smallest eigenvalue. This alignment minimizes the contraction term while maximizing the drift term. The target drift is scaled by a factor $\beta$ to normalize the relative magnitude of both terms. In Figure 32 (https://bit.ly/3OcwiKe), we observe that when the condition number $\kappa = 1$ (isotropic case), the error norm decreases without divergence. For larger $\kappa$, divergence may occur, while reducing $\beta$ mitigates this effect.
>
> > But any sub-Gaussian...
>
> We appreciate this insightful comment. We agree that sub-Gaussian distributions share similar tail properties; however, Gaussianity provides additional structure beyond light tails. First, the Gaussian is the maximum-entropy distribution under a fixed variance constraint, encouraging efficient use of representational capacity. Second, for Gaussian distributions, all higher-order moments are determined by the covariance (via Isserlis’s theorem), so the covariance fully characterizes the distribution. In contrast, for general sub-Gaussian distributions, higher-order structure is not determined by the covariance and may influence behavior beyond second-order statistics. Finally, the Gaussian provides a canonical and tractable target for distribution matching within our framework.
>
> > Plotting the eigenvalue...
>
> Thanks for the suggestion! To provide additional insight beyond the PCA figures included in the paper, we compute complementary metrics, including the entropy of eigenvalues (higher indicates more isotropy) and the average condition number ratio (baseline / SIGReg). These results are shown in Tables 11 (https://bit.ly/4sfsaHC) and 12 (https://bit.ly/4bY2Q2H), respectively. The results indicate that SIGReg increases spectral entropy and significantly reduces the condition number, further supporting our theoretical claims.
>
> > The analysis assumes...
>
> While our analysis is based on a linear setting, this is standard practice in RL literature, as adequate tools for non-linear analysis in highly non-stationary regimes are still limited. Our goal is to isolate a key mechanism linking representation geometry to stability under non-stationarity. In practice, the method operates in a fully non-linear deep RL setting where both representations and targets evolve. Nevertheless, empirical results across multiple algorithms (PQN, PPO) and domains (Atari, Isaac Gym) suggest that the identified mechanism remains relevant. We will include this discussion in the limitations section.
>
> ### Q: Table 2 shows that VICReg-style whitening...
> We observe that enforcing isotropic Gaussian representations consistently outperforms VICReg-style whitening in most environments. However, there are a few games (e.g., KungFuMaster, BattleZone, Phoenix) where whitening performs comparably. In these cases, enforcing well-conditioned second-order statistics (i.e., covariance) appears sufficient to stabilize learning. In contrast, in the majority of the environments, Gaussianity provides additional gains by controlling higher-order variability (i.e., tail behavior), which is not captured by whitening alone. This is consistent with our analysis in Sec. 3.3, where Gaussian representations reduce the variance of the drift term under non-stationarity.We will include a per-environment analysis in the revised version to further clarify this behavior.

---

> > ### Author Rebuttal · Reviewer_DkR6 · 2026-04-03
> >
> > Thank you or the response.

---

> > > ### Author Response · Authors · 2026-04-08
> > >
> > > Thank you for your positive feedback and for taking the time to engage with our rebuttal. We’re glad the response addressed your concerns. Thanks!
> > >
> > > * Authors

---

### Official Review · Reviewer_e56C · 2026-03-14

**Soundness:** 3
**Presentation:** 3
**Significance:** 3
**Originality:** 3
**Overall Recommendation:** 5
**Confidence:** 3

**Summary:**

**Problem**

Deep RL suffers from inherent non-stationarity where data distributions shift as training progresses, leading to training instability, neuron dormancy, and representation collapse, which significantly degrade learning efficiency and limit scalability.

**Proposed Method**

The paper proposes using isotropic Gaussian embeddings to mitigate the distribution shift problem. Theorem 3.1 provides theoretical justification by analyzing tracking error dynamics, showing that isotropy equalizes contraction across all directions and Gaussianity minimizes drift variance. SIGReg is adopted as the practical implementation — it projects embeddings onto random unit vectors and performs univariate distribution matching, bypassing expensive high-dimensional distribution matching with minimal computational cost.

**Results**

Validated on CIFAR-10 under distribution shift. On ALE (discrete control), 51 out of 57 Atari games (89.5%) show improvement with mean AUC gain of 889% and median of 138%. On IsaacGym (continuous control), improved training stability and reduced variance across seeds. The method also improves stability in PPO, though with smaller gains as PPO is inherently more stable.

**Compliance With Llm Reviewing Policy:**

Affirmed.

**Final Justification:**

The authors addressed my concerns satisfactorily in the rebuttal. I maintain my score.

**Key Questions For Authors:**

The method generally performs well across ALE environments, but there are environments where performance degrades. What are the underlying causes? Is there any additional analysis available?

**Limitations:**

- The theoretical analysis is based on the linear readout assumption; extensions to nonlinear heads, off-policy actor-critic algorithms, and continual RL settings remain unverified.
- The method shapes only the marginal distribution of representations without enforcing task-specific structure or semantic alignment, which may be suboptimal for tasks requiring highly structured features.
- Balancing isotropy with task-adaptive biases remains an open challenge.

**Strengths And Weaknesses:**

**Strengths**
- The paper provides a solid theoretical foundation for why isotropic Gaussian embeddings are optimal under non-stationarity. Theorem 3.1 decomposes tracking error dynamics into contraction and drift terms, showing that isotropy equalizes contraction across all directions (eliminating weak/blind-spot directions), while Gaussianity minimizes the variance of the drift term, suppressing extreme perturbations.
- The diagnosis that anisotropic representations create directions with weak gradient signals — where weight adaptation is slow and tracking fails when target drift occurs along those directions — is convincing and well-articulated.
- The adoption of SIGReg as a practical implementation is well-motivated. By projecting embeddings onto random unit vectors and performing univariate distribution matching to N(0, σ²), it avoids high-dimensional distribution matching with negligible computational overhead and no additional learnable parameters.
- The method is broadly applicable as a plug-in auxiliary loss that does not modify the underlying RL algorithm structure, demonstrated across both PQN and PPO.

**Weaknesses**
- Lack of analysis on environments where performance degrades. In PQN, 6 out of 57 games show performance drops, and in PPO the proportion appears even higher. Some of these drops are substantial (on the order of -10¹ to -10² on the log scale in Figure 6). However, the paper only highlights "89.5% improvement" without any discussion or analysis of the failure cases. Understanding the conditions under which enforcing isotropy is harmful is important for determining the method's applicability, and at minimum a qualitative analysis of these failure cases is needed.

---

> ### Author Rebuttal · Authors · 2026-03-31
>
> We thank the reviewer for the positive and careful assessment. We are pleased that the reviewer highlighted that “the adoption of SIGReg as a practical implementation is well motivated,” and that “the method is broadly applicable as a plug-in auxiliary loss that does not modify the underlying RL algorithm structure”. We respond to their main concerns below.
>
> ### Q:
> > The method generally performs well across ALE environments, but there are environments where performance degrades. What are the underlying causes? Is there any additional analysis available?
>
> Although aggregate performance comparisons often result in one algorithm significantly outperforming a comparable baseline, it is rarely the case that it does so across all environments in a suite [1,2,3]. Uncovering the underlying causes for this variance in performance remains an open problem, and lies beyond the scope of this paper. Nonetheless, we provide per-game training curves for all our experiments to enable the community to continue in this investigation.
>
> To further illustrate this variance, we highlight Freeway as a representative case where SigReg degrades performance for both PQN and PPO. Freeway has sparse and weak reward signals, and the additional regularization may introduce frequent but uninformative optimization signals that interfere with learning. The task also appears to require limited representational complexity, as reflected by its low effective rank.  Although SigReg increases effective rank and reduces dormancy, indicating improved representation utilization, these changes do not translate into better performance, and in PQN lead to complete collapse. This gap suggests that the inductive bias introduced by SigReg may be misaligned with the task, and its impact on optimization dynamics can outweigh representational benefits. A similar pattern is observed in Pitfall and Private Eye, which have sparse rewards, as well as Robotank and Bowling, which exhibit low visual and state complexity. In these environments, SigReg also leads to degraded PQN performance, suggesting that the regularization may be misaligned with learning dynamics when reward signals are weak or the representational demands are limited. We will include this discussion in the final version.
>
> [1.] Castanyer  et al., Stable Gradients for Stable Learning at Scale in Deep Reinforcement Learning, NeurIPS ‘25
>
> [2.] Castro et all., MICo: Improved representations via sampling-based state similarity for Markov decision processes, NeurIPS’21.
>
> [3.] Schwarzer, Bigger, Better, Faster: Human-level Atari with human-level efficiency, ICML’23
>
> > extensions to nonlinear heads, off-policy actor-critic algorithms, and continual RL settings remain unverified
>
> While our analysis is based on a linear setting, this is standard practice in the RL literature, as we do not yet have the adequate tools for non-linear analyses in these highly non-statationary settings. Our goal is to isolate a tractable mechanism linking representation geometry to stability under non-stationarity. Empirically, the method remains effective in broader settings, including nonlinear networks, PPO, and continuous control. Extending the theory to richer function classes and continual RL is an important direction for future work. We will make sure to include this in the limitations section of the revised manuscript.
>
>
> > The method shapes only the marginal distribution of representations without enforcing task-specific structure or semantic alignment, which may be suboptimal for tasks requiring highly structured features. Balancing isotropy with task-adaptive biases remains an open challenge.
>
> We agree that SIGReg shapes marginal statistics rather than enforcing task-specific structure. Our goal is to evaluate  a generic regularizer that improves conditioning and stability without restricting the representation space. In practice, isotropy complements, rather than replaces, task-specific learning, but balancing isotropy with adaptive structure is an important direction for future work. We will include a discussion on this in the revised manuscript.

---

> > ### Author Rebuttal · Reviewer_e56C · 2026-04-06
> >
> > The authors have addressed my concerns satisfactorily. No further questions.

---

> > > ### Author Response · Authors · 2026-04-08
> > >
> > > Thank you for your positive feedback and for taking the time to engage with our rebuttal. We’re glad the response addressed your concerns. Thanks!
> > >
> > > * Authors

---

### Official Review · Reviewer_11UV · 2026-03-19

**Soundness:** 3
**Presentation:** 2
**Significance:** 2
**Originality:** 2
**Overall Recommendation:** 4
**Confidence:** 3

**Summary:**

The paper focuses on one of the key problems in Deep RL, which is the training stability. Isotropic Gaussian embedding, due to their attractive characteristics, can be used as guidance for the internal representations. These characteristics are discussed as minimum sensitivity to drift, maximize entropy under variance constraint and balanced utilization of representational dimension. The main source of inspiration is from Balestriero and LeCun, 2025.

Section 3 provides theoretical support for the empirical robustness of isotropic Gaussian.  The considered problem in Section 3 is ultimately a least square estimation problem under a linear model assumption. The derivation of the minimizer and gradient flow equation is quite standard (Appendix B).  Theorem 1 derives a closed form expression for the error norm rate given in (1). Two terms appear in Theorem 1: one related to contraction controlled by the spectrum of embedding covariance matrix $\Sigma_\phi$, and the other one related to the drift $b_t$.

Sketched Isotropic Gaussian Regularization form Balestriero & LeCun, 2025 is used (details in page 3 and 4) to promote isotropic Gaussian property.  The main experiments of the paper: CIFAR-10 with distribution shift (simulating non-stationarity using periodic label permutation), Deep RL tested on Atari-10. A similar analysis is done for PPO in 4.6 and for continuous space in the same section.

**Compliance With Llm Reviewing Policy:**

Affirmed.

**Final Justification:**

Although the theoretical results were limited to a simple setup, the authors managed to clarify further my concerns about some of the derivations, and added new experiments justifying the choice of Gaussian further.

**Key Questions For Authors:**

* Please precisely specify the probability spaces when defining $\Sigma_\phi(t)$ and $b_t$. What are the sources inducing randomness in these two terms. Then it becomes clearer why $b_t$ and $\phi$ are correlated, as well as clarifying many other theoretical arguments of paper.
* Regarding "Equalizing the contraction across all directions therefore requires distributing the total variance uniformly across dimensions, which implies that all eigenvalues of $\Sigma_\phi$ are equal":  To be precise, $\Sigma_\phi$ as the covariance matrix of a random vector needs to be a diagonal matrix with equal diagonal entries in order to have equal eigenvalues. In other words, the random vector needs to be with i.i.d. entries. However, $\Sigma_\phi$ seems to be used a dense non-diagonal matrix throughout the text. Can authors clarify further?
* The statements like "it achieves the least-structured, most unbiased representation compatible  with isotropy, balancing expressiveness with stability" is difficult to parse as the key technical terms are not precisely used.  Example what is "unbiased representation", as the term bias has a clear definition in estimation theory.  What is "balancing expressiveness" as the expressivity is captured usually within the framework of approximation theory. And so on.
* Regarding the effect of isotropy:  The authors claim "under a fixed total variance budget, anisotropic representations increase the likelihood that some directions exhibit weak contraction and large drift." Can authors provide a more rigorous justification for this claim? The current formulation is bit too informal and schematic. For instance, the error $e(t)$ is itself a random variable, impacting $b_t$ and its derivation. Therefore, the overall drift term (in red) can have more complex dynamics (for example, maybe the dynamics of $e(t)$ and $b_t$ counter each other?). In any case, the existence of directions with weak contraction and large drift does not seem trivial to me.
* The arguments of section 3.3 are quite informal. For instance, why Isserlis's theorem matters at all for controlling the drift and contract, largely determined by the spectrum of the covariance matrix? The authors indicate this in B.1.5, however I still feel it is not satisfactory. See my comments on Section B.1.5.
* Has Figure 1 been derived based on the simulation or it is just a drawing? If so, I encourage the authors to replace it with an actual simulation which makes the message stronger and more technically solid.
* The claim of usefulness of Gaussian is justified informally using intuitive justification (for instance the claim 3  of page 3 in the yellow box). In my opinion, a proper justification of such hypothesis (empirically and theoretically) can make the findings more compelling.
* It is not clear which characteristics of Gaussian is essential for the claim: maximum entropy feature or tail decay? Regarding the tail decay, many other distributions provide such behavior. The sub-gaussian distributions satisfy similar property, and besides, they can be used for similar isometric embedding and measurements as witnessed by the plethora of works in compressed sensing field.
* It is not clear why maximum entropy feature is needed.
* Suggestion: regarding  the existence of directions with weak contraction and large drift, the authors can setup toy experiments to verify that for various cases.
* The universality of Gaussian embedding has been discussed in some prior works going back to Johnson-Lindenstrauss lemma. Can the authors comment on the connection further?
* Regarding SIGReg application: does it get applied only on the last representation layer, or applied layer wise?
* In section 4.3, the authors mention "minimizing the imaginary" and "minimizing the real", the meaning of which is clarified in the appendix, line 1146. I suggest adding it to the main paper, where SIGReg loss is discussed.
* How is the alternative distributions (Laplacian and Logistic) are imposed? Note that if SIGReg method is used (modified with the new characteristic function), then this is wrong. The projections do not need to have the same distribution.
* Regarding the choice of Atari-10: if I understand correctly, the non-stationarity comes from the algorithm component, and the environment can be considered stationary. Is that correct?
* Appendix B, line 843: "Our goal is to analyze under what conditions $||e||_2^2 = 0$ is contractive" --> Not clear what it means.
* Line 1012 in Appendix B.1.5: is the reference to Eq. 56 correct?
* Line 1042: "the first term in Eq. 55 is independent of the choice of distribution.": Can you explain this statement further?  I understand it as " the first term will always be there regardless of the choice of the distribution", not that the value is not dependent. Is  that correct?
*  Line 1044-1045: The statement "the variance of the whole term is reduced" is only true if the first term has no dependence on the choice of the distribution of $\phi$.
*  Following the argument of B.1.5, I cannot see how it is mathematically proven that Gaussian distribution under total variance (trace?) constraint minimizes the drift term. The argument seems to be heuristic (see for instance the treatments from eq. 56 to eq 58 with rough approximations).

**Limitations:**

See my comments above.

**Strengths And Weaknesses:**

**Strengths**

* The problem of optimization in non-stationary setting is an important one in context of reinforcement learning. The authors take a step toward solving this problem by proposing a regularization on the latent space. Such approaches hold the promise of being applicable to variety of the problems.

* The various ablation experiments of the paper (in the RL section) are well thought and provide good insights.

**Weakness**

* The theoretical rigor of the paper can be improved. Particularly, some justifications for using Gaussian distributions as sketched in Appendix B are not as strong as it is claimed, and rely on various approximations.
* The theoretical analysis of the paper is not too general or novel/challenging. There are many works on stability analysis of such linear time variant systems. For instance, there are  many works on Lyapunov stability analysis of Kalman-Bucy filters, see for instance:  Del Moral, Pierre, and Julian Tugaut. "On the stability and the uniform propagation of chaos properties of ensemble Kalman–Bucy filters." The Annals of Applied Probability 28.2 (2018): 790-850.
* Ignoring the theoretical section, the paper seems to apply SIGReg method to the training pipeline. In this sense, the contribution of the paper is not very strong. Atari-10 is the only RL experiment, and although it is good starting point, other benchmarks could have been added.

---

> ### Author Rebuttal · Authors · 2026-03-31
>
> Thanks for the careful review. Due to the char limit, we provide concise responses below and will incorporate all suggestions and expand the discussions in the final version.
> # W1
> Please see Sects 4.3 & B.1.5 for theoretical justification for Gaussians, & responses to Qs 5,8,9,18,19,20.
> # W2
> Our analysis is complementary to those, and novel in deep RL. We show how properties of embedding distrs (e.g., isotropy & Gaussianity) influence contraction and drift, and thus the tracking behavior of the learning dynamics. Nonetheless, we’ll add a discussion of the works mentioned.
> # W3
> Our evaluation is not limited to Atari-10. We also run on full suite (Figs 6,15,16) & Isaac Gym (Figs 7,17), as well as CIFAR-10 experiments (Figs 3,8) which allows us to control for non-stationarity.
> # Q1
> Source of randomness for $\Sigma_{\phi}$ is state visitation distribution, & for $b_t$ it’s trajectory distribution, where trajectories are sampled from the joint distribution induced by the initial state distribution, policy, and dynamics. We will provide a formal definition of probability space in revised version.
> # Q2
> Any matrix of the form $\Sigma_\phi = Q(\sigma^2 I)Q^\top $ for any orthogonal matrix $Q$ satisfies this condition. The condition means covariance is isotropic (i.e., rotation invariant): exactly the condition enforced by SIGReg.This condition doesn’t require components of $\phi$ to be indep, only that distr has no preferred directions in standard basis. While i.i.d. entries are sufficient to ensure isotropy, they aren't necessary.
> # Q4
> Since $b_t$ isn't controllable, we consider worst-case scenario: when 1st term is small (i.e., $e_t$ aligns with direction of smallest eigenvalue) and drift term is large and positive. The magnitude of the 2nd term scales with condition number. Hence, optimal case is when the condition number is 1, which avoids degenerate directions for $e_t$ and prevents amplification of 2nd term. For toy setup showing existence of such direction, see answer to Reviewer DkR6.
> # Q5
> Isserlis’s theorem explains why Gaussian distrs are preferable among isotropic ones. While different distributions may share the same covariance, non-Gaussian ones can differ in higher-order moments, which are not controlled by eigenvalues of the cov matrix & may induce rare but large fluctuations in drift. In contrast, for Gaussian distrs, all higher-order moments are fully determined by the covariance, eliminating these additional degrees of freedom.
> # Q6
> Thank you for suggestion! Please see answer to Reviewer DkR6.
> # Q7
> For theoretical justification, see responses to Qs 5,8,9,18-20. For empirical, we've added toy experiments please see answer to Reviewer DkR6 (Figures: https://bit.ly/4tmvsK0). Table 2 compares isotropic Gaussian representations against Laplacian, Logistic, and covariance-whitened baselines.
> # Q8
> Sub-Gaussian families share tail decay but form a broad class. Gaussian is a canonical choice due to its rotational invariance & fully specified higher-order moments, which support stable representations. We will clarify this motivation and acknowledge sub-Gaussian alternatives.
> # Q9
> Maximizing entropy encourages full use of representational capacity as entropy shows the information content of a random variable.
> # Q10
> Please see answer to Reviewer DkR6.
> # Q11
> Gaussian random projections approx preserve pairwise structure (as in JL). In SIGReg, projections are used as a sketching mechanism to enforce distributional constraints along multiple directions. The link is thus geometric rather than distributional, as projected features are not necessarily Gaussian.
> # Q12
> Only on last layer.
> # Q14
> Our goal is to impose directional statistical constraints through projections as a form of regularization, rather than to match a specific joint distribution. This idea is related to classical projection-based results such as Cramér–Wold [1] type arguments, where properties of projections reflect important aspects of the multivariate distribution, even if they do not fully determine it.
>
> [1.] Cramér et al., Some theorems on distribution functions, Journal of the London Mathematical Society'1936.
> # Q15
> Yes, env is stationary.
> # Q16
> We mean that update dynamics make the stationary point an attracting fixed point such that trajectories move monotonically toward it.
> # Q17
> We thank the reviewer for catching this, 56 should be 50.
> # Q18-20
> Under linear readout assumption, $Q_{\theta^-}$ can be written as an inner product between weights & embedding. When differentiating wrt $\phi$, 1st term depends only on task and is independent of embedding distribution, while 2nd term depends on distribution & contributes to variance when residual $r$ is non-zero. In Gaussian case, residual term vanishes(Eq.46), removing this contribution & reducing total variance. Thus, distributional dependence in Eq.55 enters only through residual term, which is zero iff the distribution is Gaussian. Eqs.56-58 illustrate how non-Gaussian contributions scale.

---

> > ### Author Rebuttal · Reviewer_11UV · 2026-04-03
> >
> > I would like to thank the authors for their answer. I did go through their response to other reviewers and considered those reviews as well. It is encouraging to see all reviewers were also mentioning the importance of the considered problem, namely stability under nonstationary settings, and the solution based on shaping the representation space with SIGReg.
> >
> > The new experiments in the anonymized link definitely strengthen the relevance of the used approach and Gaussian assumption, and many other answers are very helpful. The answers to Q18-20 clarify well the concern about the theoretical derivations, although I appreciate if the authors add some of these discussions to the final version.
> >
> > I plan to raise my score accordingly.

---

> > > ### Author Response · Authors · 2026-04-08
> > >
> > > Thank you for the thoughtful follow-up and for taking the time to engage with both our responses and the additional experiments. We’re glad the new results helped clarify the benefits of SIGReg and its effectiveness in addressing stability under nonstationary settings. We also appreciate your suggestion regarding Q18–20 and will incorporate these clarifications into the final version to further strengthen the theoretical discussion.
> > >
> > > We are encouraged by your positive assessment of the problem’s importance and the approach, and we thank you again for your careful and constructive evaluation.
> > >
> > > * Authors

---

### Decision · Program_Chairs · 2026-04-30

**Decision:**

Accept (spotlight)

**Comment:**

The problem of dealing with non-stationarity in reinforcement learning is an important one and this paper makes a significant contribution to addressing this problem. The authors introduce a relatively simple and computationally lightweight approach to regularize training. Multiple reviewers liked the carefully done ablations in the experiments. Initially, there were some concerns around novelty and comparing to alternative regularization methods which were resolved during the rebuttal.